# Towards Robust and Expressive Whole-body Human Pose and Shape Estimation

**Hui En Pang[1]**,    **Zhongang Cai[1,2]**,    **Lei Yang[2]**,    **Qingyi Tao[2]**,    **Zhonghua Wu[2]**,
**Tianwei Zhang[1]** ✉,    **Ziwei Liu[1]**

[1]S-Lab, Nanyang Technological University    [2]SenseTime Research
{huien001, tianwei.zhang, ziwei.liu}@ntu.edu.sg
{caizhongang, yanglei, taoqingyi, wuzhonghua}@sensetime.com

## Abstract

Whole-body pose and shape estimation aims to jointly predict different behaviors (e.g., pose, hand gesture, facial expression) of the entire human body from a monocular image. Existing methods often exhibit degraded performance under the complexity of in-the-wild scenarios. We argue that the accuracy and reliability of these models are significantly affected by the quality of the predicted *bounding box*, e.g., the scale and alignment of body parts. The natural discrepancy between the ideal bounding box annotations and model detection results is particularly detrimental to the performance of whole-body pose and shape estimation. In this paper, we propose a novel framework RoboSMPLX to enhance the robustness of whole-body pose and shape estimation. RoboSMPLX incorporates three new modules to address the above challenges from three perspectives: **1) Localization Module** enhances the model's awareness of the subject's location and semantics within the image space. **2) Contrastive Feature Extraction Module** encourages the model to be invariant to robust augmentations by incorporating contrastive loss with dedicated positive samples. **3) Pixel Alignment Module** ensures the reprojected mesh from the predicted camera and body model parameters are accurate and pixel-aligned. We perform comprehensive experiments to demonstrate the effectiveness of RoboSMPLX on body, hands, face and whole-body benchmarks. Codebase is available at https://github.com/robosmplx/robosmplx.

## 1  Introduction

Human pose and shape estimation tries to build human body models from monocular RGB images or videos. It has gained widespread attention owing to its extensive applications in various fields, including robotics, computer graphics, and augmented/virtual reality. Early works use various statistical models (e.g., SMPL [31], MANO [43], FLAME [26]) to individually reconstruct different parts, including human body [17, 22, 4, 16, 24, 10, 21, 20], face [9, 8, 12], and hand [28, 3, 63]. Recently, there is a growing interest in whole-body estimation [11, 6, 61, 44, 58], which jointly estimates the pose, hand gestures and facial expressions of the entire human body from the input. Commonly these methods first employ separate sub-networks to extract the features of body, hands and face. These features are then used to predict whole-body 3D joint rotations and other parameters (e.g., body shape, facial expression), which are further combined to generate the whole-body 3D mesh. This is a crucial step towards modeling human behaviors in an efficient and practical manner.

However, achieving accurate and robust whole-body estimation is particularly challenging as it requires precise estimation of each body part and the correct connectivity between them. In particular, due to the smaller sizes of hand and face images, they are typically localized, cropped and resized to higher resolutions before being processed by the relevant sub-network. To tackle the absence

37th Conference on Neural Information Processing Systems (NeurIPS 2023).

of ground-truth bounding boxes in the real-world scenarios, existing whole-body methods utilize various detection techniques to obtain the crops. The accuracy of the whole-body estimation is highly sensitive to the quality of input crops. Our experiment results in Section 3 show that even minor fluctuations in the scale and alignment of input crops can significantly affect the model performance, indicating a limited ability to localize and extract meaningful features about the subject in the image.

The lack of robustness in existing whole-body pose and shape estimation methods highlights three critical aspects that can be improved upon: 1) accurate localization of the subject and its parts, 2) accurate extraction of useful features, and 3) accurate pixel alignment of outputs. Inspired by these findings, we propose three novel modules, each specifically designed to address a particular goal:

- **Localization Module**. This module implements sparse and dense prediction branches to ensure the model is aware of the location and semantics of the subject's parts in the image. The learned location of the joint positions are helpful in recovering the relative rotations.

- **Contrastive Feature Extraction Module**. This module incorporates a pose- and shape-aware contrastive loss, along with positive samples, to promote better feature extraction under robust augmentations. By minimizing the contrastive loss, the model can produce consistent representations for the same subject, even when presented with different augmentations, making it robust to various transformations and capable of extracting meaningful invariant features.

- **Pixel Alignment Module**. This module applies differentiable rendering to ensure a more precise pixel alignment of the projected mesh, and learn more accurate pose, shape and camera parameters.

By integrating these three modules, we build a more robust and reliable whole-body pose and shape estimation framework, `RoboSMPLX`. Comprehensive evaluations demonstrate its effectiveness on body, face, hands and whole-body benchmarks.

## 2  Related Works

**Whole-body Mesh Recovery.** Despite significant progress in 3D body-specific [23, 22, 4, 16, 24, 10, 21, 20], hand-specific [28, 3], and face-specific [9] mesh recovery methods, there have been limited attempts to simultaneously recover all those parts. Early studies on whole-body pose and shape estimation primarily fit a 3D human model to 2D or 3D evidence [15, 53, 41, 54], which can be slow and susceptible to noise. Recent studies utilized neural networks to regress the SMPL-X parameters for a whole-body 3D human mesh. The model is composed of separate sub-networks to process body, hand and face, respectively. *One-stage* methods, e.g., OS-X [27], have the benefit of reduced computational costs and improved communication within part modules for more natural mesh articulation. However, the omission of hand and face experts makes it difficult for the model to leverage the widely available part-specific datasets, thus decreasing the hand and face performance. *Multi-stage* methods, e.g., ExPose [6], FrankMocap [44], PIXIE [11] and Hand4Whole [35], use different techniques to localize part crops.

Expose [41] and PIXIE [11] localize hand and part crops from the body mesh, making them dependent on the accuracy of body poses. Minor rotation errors accumulated along the kinematic chain may result in deviations in joint locations and thus inaccurate part crops. In contrast, Hand4Whole [35] predicts hand and face bounding boxes using a network leveraging image features and 3D joint heatmaps, but the resulting crops have low resolution. PyMAF-X [11] relies on an off-the-shelf whole-body pose estimation model to obtain crops, which, while more accurate, incurs extra computation. More detailed comparison with PyMAF-X are in Appendix C.

**Robustness in vision tasks.** Efforts to tackle robustness in vision tasks have utilized diverse strategies such as data augmentation, architectural innovations, and training methodologies [25, 56, 42, 30, 51, 60, 2]. AdvMix [51] employs adversarial augmentation and knowledge distillation, challenging models with corrupted images to foster learning from complex samples. Architectural modifications, such as novel heatmap regression [60], have been introduced to mitigate the impact of minor perturbations. HuMoR [42] utilizes a conditional variational autoencoder to capture the dynamics of human movement, thereby achieving generalization across diverse motions and body shapes. Additionally, PoseExaminer [30] employs a multi-agent reinforcement learning system to uncover failure modes inherent in human pose estimation models, highlighting model limitations in real-world scenarios. Complementing these efforts, Robo3D [25] provides a comprehensive benchmark for assessing the robustness of 3D detectors and segmentors in out-of-distribution scenarios.

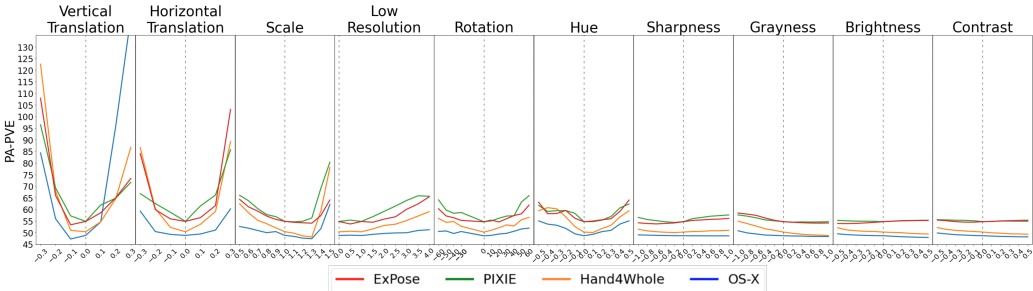

Figure 1: **Wholebody PA-PVE errors under different augmentations (sorted in descending order).** The dashed line indicates baseline performance without augmentation.

Furthermore, [56] utilize a confidence-guided framework to improve the accuracies of propagated labels. Contrastive learning, as demonstrated by CoKe [2], has also been employed to enhance robustness in keypoint detection, especially in occlusion-prone scenarios.

**Contrastive Learning.** Recently contrastive learning has demonstrated state-of-the-art performance among self-supervised learning (SSL) approaches. This strategy has been applied to 3D hand pose and shape estimation [46, 63]. Sanyal et al. [45] incorporate a novel shape consistency loss for 3D face shape and pose estimation that encourages the face shape parameters to be similar when the identity is the same and different for different people. Choi et al. [5] were the first to apply contrastive learning for 3D human pose and shape estimation. They found that SSL is not useful for this task, as the learned representations could be challenging to embed with high-level human-related information. Khosla et al. [19] proposed supervised contrastive learning for image classification tasks, which incorporates label information during training. Currently there is not attempt to apply this strategy to human pose and shape estimation, where the definition of positive samples is unclear, and data lie in a continuous space. We are the first to overcome these challenges and integrate supervised contrastive learning with whole-body pose and shape estimation.

**Pixel Alignment in Pose and Shape Estimation.** Many studies have been done to learn the subject's location in an image. Some works implicitly supervise the location. They primarily utilize projected meshes by supervising 2D joints regressed from the mesh [17, 23, 22, 4, 16, 24, 10, 21, 20]. Further supervision, such as dense body landmarks, silhouettes, and body part segmentation, is also employed to better align the predictions with the image [55, 37, 40, 49, 59, 57, 10]. Some other works explicitly learn the subject's location. Moon et al. [35] explicitly predict the keypoint locations in the image. Semantic body part segmentation is used as an explicit intermediate representation [41, 37]. PARE [41] employs a renderer to project the ground-truth mesh to the image space, and supervise the predicted part silhouette mask. However, dense part segmentation and differentiable rendering have not been employed in whole-body pose and shape estimation, which will be achieved in our framework.

## 3 Motivation

As discussed in Section 1, existing whole-body pose and shape estimation approaches suffer from the robustness issue, due to the models' sensitivity to the quality of input crops. To investigate the reasons and disclose the influence factors, we conduct a comprehensive evaluation of four state-of-the-art methods: ExPose [6], PIXIE [11], Hand4Whole [35] and OS-X [27]. We opt for a set of ten commonly encountered augmentations and vary their scales within a realistic range (see Appendix A for more details). The augmentations can be classed into three categories (1) *image-variant* augmentations: they affect the image without altering the objects' 3D poses or positions, such as color jittering; (2) *location-variant* augmentations: they modify the subject's location without changing its pose, involving operations like translation and scaling; (3) *pose-variant* augmentations: they simultaneously alter both the 3D pose and location, including rotation.

**Impact of subject localization**. We first reveal that existing models demonstrate high sensitivity to the subject's position, indicating potential difficulties in subject localization. Figure 1 reports the PA-PVE errors of the whole body under different augmentations. We observe that image-variant augmentations (contrast, sharpness, brightness, hue and grayscale) lead to an acceptable range of error rates (approximately in the 50s) and minimal fluctuation (around ±2). In contrast, location-

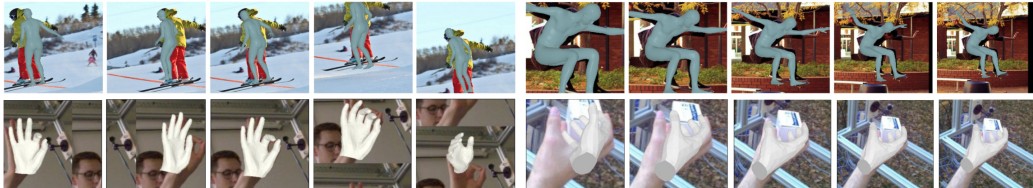

Figure 3: **Sensitivity of existing body and hand models to different alignments (left) and scales (right).**

variant augmentations altering the subject's position within the frame, such as rotation, scaling, and horizontal or vertical translation, result in substantially higher error magnitudes. This demonstrates the heightened sensitivity of existing models to changes in the subject's position. In Appendix, we provide the results of other metrics and benchmarks in Figures 22 – 23, and visualizations of whole-body estimation under different settings in Figures 25 – 26.

Such position-altering augmentations are common in real-world scenarios, where the subject in the image is often localized using external detection models and control over the quality of crops is less feasible. In practice, to guarantee the visibility of the subject, crops are often made broader, This can lead to significant performance degradation, as errors increase with smaller augmentation scale factors (<1.0) (Figure 1). Besides, horizontal and vertical translations, which correspond to scenarios where the subject is not perfectly centralized or entirely visible within the frame, can further decrease

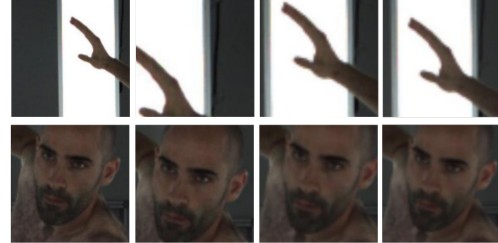

Figure 2: **Crops from (a) ExPose [6] (b) PIXIE [11], (c) Hand4Whole [35] (d)** `RoboSMPLX`**.**

the performance. Similarly, the alignment and scale of these crops also influence the pose and shape estimation systems targeting body, face and hands (Figure 3, more quantitative and qualitative evidence in Appendix M). Whole-body methods bear the additional responsibility of accurately localizing body parts such as hands and face. Inaccurate part crops (Figure 2) can adversely affect the performance of part subnetworks, and further the whole-body estimation.

**Impact of feature extraction**. The deterioration of performance in the face of such variations suggests that the model struggles to extract meaningful features. Under alterations in translation or scale, the subject remains within the image frame, though the proportion of background content may vary. It is difficult for existing methods to effectively disregard irrelevant background elements and extract relevant features related to the subject of interest. To enhance the model's robustness, it is critical to produce consistent features irrespective of various augmentations applied to the image.

**Impact of output pixel alignment**. Pixel alignment is a critical aspect of high model performance. In certain instances, despite having precise subject localization, the model fails to produce properly aligned results (Figure 25 in Appendix). This is often caused by the suboptimal camera parameter estimation. To address this issue, we need to accurately estimate the camera parameters, ensuring the projected mesh is precisely aligned with the ground-truth at the pixel level. Such precision would enhance the effectiveness of the model in producing accurate pose, shape and camera parameter predictions, improving the overall accuracy and reliability of the estimation process.

## 4 `RoboSMPLX` **Framework**

We design `RoboSMPLX` to enhance the robustness of whole-body pose and shape estimation. It provides three specialized modules to address each challenge in Section 3: 1) **Localization Module** (Section 4.2): explicitly learning the location information of the subject and incorporating it into model estimations for pose, shape and camera ; 2) **Contrastive Feature Extraction Module** (Section 4.3): reliably extracting pertinent features under various augmentations, thereby improving the model's generalization ability and robustness to a broader range of real-world scenarios; 3) **Pixel Alignment Module** (Section 4.4): ensuring that the outputs are pixel aligned.

We start with the description of `RoboSMPLX` architecture with Body, Hand and Face subnetworks (Section 4.1). Each subnetwork is integrated with the **Localization Module** and **Pixel Alignment**

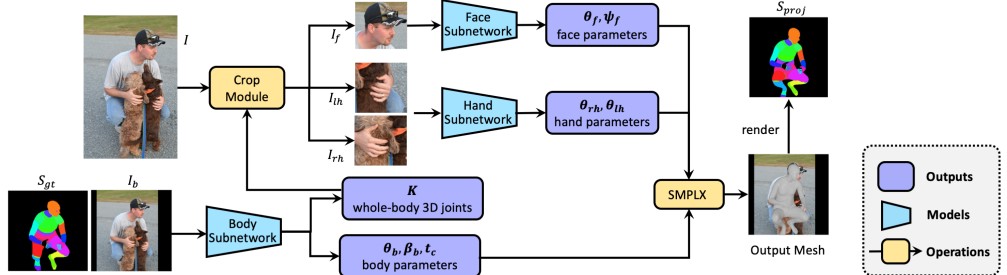

Figure 4: **Pipeline of our** `RoboSMPLX` **framework consisting of Body, Hand and Face subnetworks.**

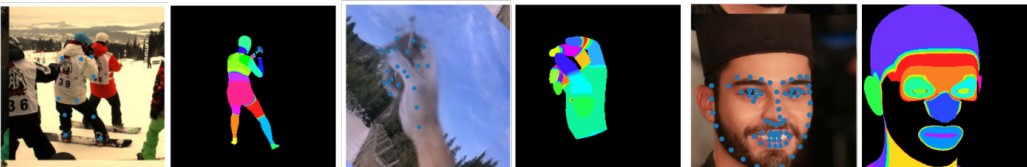

Figure 5: **Examples of keypoint and part segmentation supervision for Body, Hand and Face subnetworks.**

**Module**, and applies the **Contrastive Feature Extraction Module** for learning more robust features. Figure 6 shows the Hand subnetwork architecture. The other two subnetworks have the same designs.

## 4.1 Architecture and Training Details

Figure 4 shows the overall pipeline of `RoboSMPLX` for whole-body 3D human pose and mesh estimation. The Body subnetwork outputs 3D body joint rotations $\theta_b \in \mathbb{R}^{21\times3}$, global orientation $\theta_{bg} \in \mathbb{R}^3$, shape parameters $\beta_b \in \mathbb{R}^{10}$, camera parameters $\pi_b \in \mathbb{R}^3$, and whole-body joints $K \in \mathbb{R}^{137\times3}$. Joints corresponding to the hand and face are used to derive bounding boxes. Subsequently, hand and face images are cropped from a high-resolution image to preserve details. The Hand subnetwork predicts left and right hand 3D finger rotations $\theta_h \in \mathbb{R}^{15\times3}$. Simultaneously, the Face subnetwork generates 3D jaw rotation $\theta_f \in \mathbb{R}^3$ and expression $\psi_f \in \mathbb{R}^{10}$. When training Hand and Face subnetworks with part-specific datasets, additional parameters such as global orientation $\theta_{fg} \in \mathbb{R}^3$, shape $\beta_f \in \mathbb{R}^{50}$, and camera $\pi_f \in \mathbb{R}^3$ are estimated. These branches are discarded during whole-body estimation and training. Additional information concerning each subnetwork can be found in Appendix B. Further details regarding the training and inference durations are elaborated upon in Appendix K.

Subnetworks are trained separately, then integrated in a multi-stage manner. Initial whole-body training runs for 20 epochs. The hand and face modules are substituted with the trained Hand and Face subnetworks, followed by 20 epochs of fine-tuning to better unify the knowledge from the Hand and Face subnetworks into the whole-body understanding. Each subnetwork is trained by minimizing the following loss function $L$:

$$L = \lambda_{3D}L_{3D} + \lambda_{2D}L_{2D} + \lambda_{BM}L_{BM} + \lambda_{proj}L_{proj} + \lambda_{segm}L_{segm} + \lambda_{con}L_{con} \qquad (1)$$

Here $L_{BM}$ is the L1 distance between the predicted and ground-truth body model parameters. $L_{3D}$ denotes the L1 distance between 3D keypoints and joints regressed from the body model. $L_{2D}$ signifies the L1 distance of the ground-truth 2D keypoints to predicted and projected 2D joints. The latter are obtained by projecting the regressed 3D coordinates from the 3D mesh to the image space using the perspective projection [17]. The part segmentation loss $L_{segm}$ is the cross-entropy loss between $P_{h,w}$ after softmax and $P_{h,w}$ averaged over H×W elements, following [20]. $L_{proj}$ refers to the projected segmentation loss, which is the sigmoid loss between the projected mesh and the ground-truth segmentation map. $L_{con}$ is the contrastive loss described in Section 4.3. For wholebody training, $L_{box}$ is added to measure the L1 distance between the predicted and actual center and scale of the hands' and face's boxes.

## 4.2 Localization Module

This module focuses on subject localization by explicitly learning both sparse and dense predictions of the subject within the image. Figure 5 shows an example of the supervision used for each subnetwork.

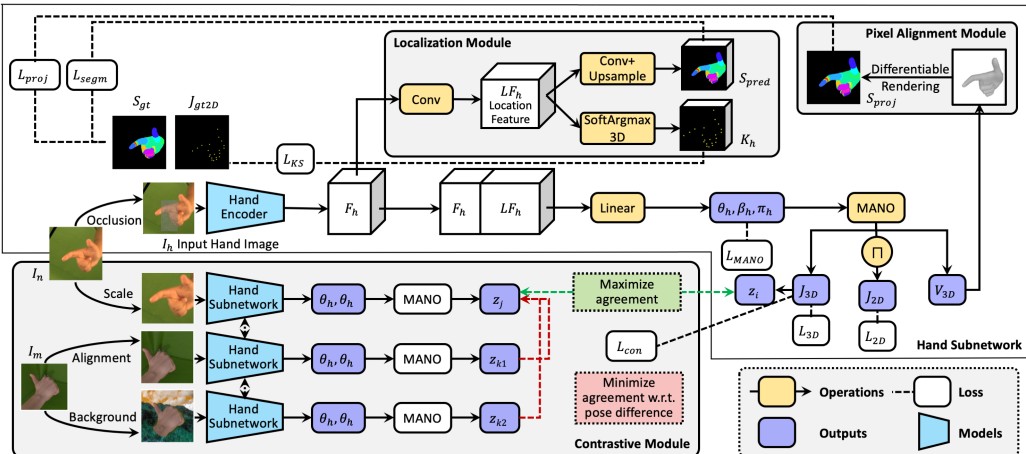

**Figure 6: Subnetwork Architecture with three modules.** We use the Hand subnetwork as an example. $z$ represents normalized $J_{3D}$ while $p$ corresponds to the ground-truth of $z$. Green and red dashed lines refers to contrastive loss for positive and negative samples respectively.

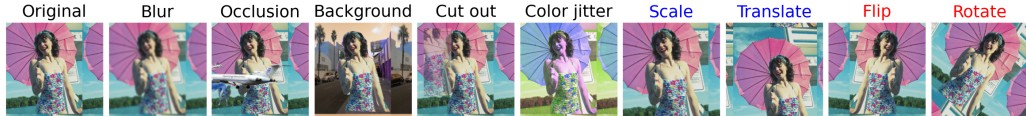

**Figure 7: Augmentations for the Body subnetwork.** Black, blue and red labels represent image-variant, location-variant and pose-variant augmentations, respectively.

In contrast to prior methods that directly output pose rotations from backbone features, this module aims to make the model explicitly conscious of the subject's location and semantics while predicting pose, shape and camera parameters. It can reduce the model's sensitivity to the variations of the subject's position, caused by minor shifts in the scale and alignment of the bounding box.

As shown in Figure 6, given an image, a convolutional backbone is utilized to extract its feature map $F \in \mathbb{R}^{512 \times 32 \times 32}$. Following [35], a $1 \times 1$ convolutional layer is then used to predict 3D feature maps $LF \in \mathbb{R}^{32J \times 32 \times 32}$ from $F$, where $J$ represents the number of predicted joints with a feature map depth of 32. $LF$ contains valuable information about the mesh's position in the image and semantics of various parts. It is concatenated with the backbone feature map $F$ to predict pose $\theta \in \mathbb{R}^P$, shape $\beta \in \mathbb{R}^{10}$ and camera translation $\pi \in \mathbb{R}^3$, where $P$ is the number of body parts. Meanwhile, $LF$ is also used to obtain extra information with two branches: (1) 3D joint coordinates $K \in \mathbb{R}^{J \times 3}$ are obtained from $LF$ using the soft-argmax operation [47] in a differentiable manner. (2) 2D part segmentation maps $S \in \mathbb{R}^{P+1 \times 64 \times 64}$ are extracted from $LF$ with several convolution layers, which model $P$ part segmentation and 1 background mask. Here, 64 represents the height and width of the feature volume, and each pixel $(h, w)$ stores the likelihood of belonging to a body part $P$.

Note that learning part segmentation maps and 3D joint coordinates is complementary, as 3D joint coordinates encode depth information that may inform part ordering in segmentation maps. Additionally, joints often reside at the boundaries of part segmentation maps, serving as separators for distinct parts. The Body subnetwork utilizes 24 parts $P$ and 137 joints $J$, the Hand subnetwork employs 16 parts $P$ and 21 joints $J$, while the Face subnetwork employs 15 parts $P$ and 73 joints $J$.

## 4.3 Contrastive Feature Extraction Module

This module incorporates a pose- and shape-aware contrastive loss, along with positive samples. By minimizing this loss, the model can produce consistent representations for the same subject, even when presented with different augmentations, thus fostering the extraction of meaningful features.

Conventional contrastive learning methods based on SSL (e.g., SimCLR) face challenges in unifying similar pose embeddings and distancing dissimilar ones in human pose and shape estimation tasks. Without labels for guidance, images with similar poses could be misidentified as negative samples and contrasted away, complicating the self-organization of the embeddings in pose space. Figures 9 to 12 in Appendix show their ineffectiveness for the 3D human pose and shape task [5] by visualizing the retrieved samples from the embeddings. The supervised contrastive learning approach by Khosla

et al. [19], though effective for image classification, might not extend well to human pose and shape estimation, which is a high-dimensional regression problem and poses exist in a continuous space rather than well-defined classes.

Our module overcomes the aforementioned issues with two innovations. First, we experiment with three human pose representations $z$ and the corresponding distance functions: (1) A concatenated form of the global orientation and rotational pose; (2) global orientation and rotational pose as separate entities (3) 3D root-aligned joints regressed from the body model, derived from pose and shape inputs. For (1) and (2), we explore relative rotations in two forms: 6D vector and rotation matrix representation. For (3), L1, Smooth L1, and Mean Squared Error (MSE) was used (Table 9).

Second, we investigate ten data augmentations, and classify them into three categories (see Figure 7 for the Body subnetwork, and Figure 29 in Appendix for Hand subnetwork): (1) *image-variant* augmentations such as color jittering, blur, occlusion and background swapping; (2) *location-variant* augmentations involving translation and scaling; (3) *pose-variant* augmentations including rotation and horizontal flipping. Our ablation study in Table 10 shows that augmentations with varied global orientation are detrimental to the model performance. Consequently, we exclude such modifications when constructing positive pairs. Instead, each positive sample is constructed utilizing a random combination of location-variant and color-variant augmentations.

Formally, for a batch of $N$ images, we construct another $N$ images by applying augmentation to each sample. For each anchor $i$, let $j$ be the corresponding augmented sample. Then $i$ is contrasted against $2N - 1$ terms (1 positive and $2N - 2$ negatives). The loss takes the following form:

$$
\mathcal{L}_{con} = \sum_{i=1}^{N} \left( \tau_{pos} \left( \left| \mathrm{d}\left(\boldsymbol{p}_i, \boldsymbol{p}_j\right) - \mathrm{d}\left(\boldsymbol{z}_i, \boldsymbol{z}_j\right) \right| \right) + \tau_{neg} \sum_{k=1}^{2N} \mathbb{1}_{[k \neq i,j]} \left( \left| \mathrm{d}\left(\boldsymbol{p}_i, \boldsymbol{p}_k\right) - \mathrm{d}\left(\boldsymbol{z}_i, \boldsymbol{z}_k\right) \right| \right) \right)
$$
(2)

where $\boldsymbol{z}_i$, $\boldsymbol{z}_j$ and $\boldsymbol{z}_k$ denote the predicted pose representations, and $\boldsymbol{p}_i$, $\boldsymbol{p}_j$ and $\boldsymbol{p}_k$ denote the ground-truth pose representations for the anchor, positive and negative samples in the batch. The objective of this loss function is to minimize the distance between the positive pairs and maximize the distance between the negative pairs, in alignment with the pose similarity. Note that unlike traditional approaches where the distance is the same for all negative samples, the pairwise distance $\mathrm{d}(\boldsymbol{p}_i, \boldsymbol{p}_k)$ varies depending on the pose similarity.

### 4.4 Pixel Alignment Module

This module employs differentiable rendering to ensure that the projected mesh aligns precisely at the pixel level. The alignment is supervised by the projected mask loss. Attaining a proper alignment between the ground-truth part segmentation and rendered mesh requires the accurate prediction of pose, shape, and camera parameters, which subsequently leads to a more precise estimation process.

## 5 Experiments

**Datasets.** For whole-body training, we employ Human3.6M (H36M) [13], COCO-Wholebody [14] (the whole-body version of MSCOCO [29]) and MPII [1]. The 3D pseudo-ground truths for training are acquired using NeuralAnnot [36]. For hand-specific training, we use FreiHAND [62], Interhand [34] and COCO-Wholebody Hands [14]. For face-specific training, we use FFHQ [18], BUPT [52] and AffectNet [32]. For evaluations specific to 3D body, 3D hand, and 3D face, we utilize 3DPW [50], FreiHAND [62], and Stirling [11], respectively. For the 3D whole-body evaluation, we use EHF [41] and AGORA [39]. Additionally, we present qualitative results on the MSCOCO validation set.

**Metrics.** Mean Per Joint Position Error (MPJPE) and Mean Per-Vertex Position Error (MPVPE) are employed to evaluate the positions of 3D joint and mesh vertices, respectively. Each metric calculates the average 3D joint distance (in *mm*) and 3D mesh vertex distance (in *mm*) between the predicted and ground-truth values after aligning the root joint translation. The pelvis serves as the root joint for whole-body and body, whereas the wrists and neck are utilized as root joints for hands and face. Procrustes Aligned (PA) variants of these metrics, PA-MPJPE and PA-MPVPE, further align with rotation and scale. We report the average errors for the left and right hands as the 3D hand error.

Table 1: **Evaluation of the Hand subnetwork**.

| Method | PA-PVE ↓ | PA-MPJPE ↓ | F-Scores ↑ |
|---|---|---|---|
| **\* Hand-only** | | | |
| FreiHAND [62] | 10.7 | - | 0.529/0.935 |
| Pose2Mesh [4] | 7.8 | 7.7 | 0.674/0.969 |
| I2L-MeshNet [33] | 7.6 | 7.4 | 0.681/0.973 |
| METRO (HR64) [28] | **6.7** | **6.8** | 0.717/0.981 |
| **\* Whole-body** | | | |
| ExPose [41] | 11.8 | 12.2 | 0.484/0.918 |
| Zhou et al. [61] | - | 15.7 | -/- |
| FrankMocap [44] | 11.6 | 9.2 | 0.553/0.951 |
| PIXIE [11] | 12.1 | 12 | 0.468/0.919 |
| Hand4Whole † [35] | 7.7 | 7.7 | 0.664/0.971 |
| HMR (Baseline) [17] | 8.6 | 8.9 | 0.605/0.963 |
| PyMAF [58] | 8.1 | 8.4 | 0.638/0.969 |
| PyMAF † [58] | 7.5 | 7.7 | 0.671/0.974 |
| RoboSMPLX | **7.3** | **7.5** | 0.683/0.976 |
| RoboSMPLX † | **7.1** | **7.4** | 0.688/0.978 |
| RoboSMPLX (**HR64**) | **6.7** | **6.9** | 0.715/0.981 |

Table 2: **Evaluation of the Body subnetwork**.

| Method | PA-MPJPE ↓ | MPJPE ↓ | PVE ↓ |
|---|---|---|---|
| HMR (Res50) [17] | 76.7 | 130 | - |
| GraphCMR (Res50) [24] | 70.2 | - | - |
| SPIN (Res50) [22] | 59.2 | 96.9 | 116.4 |
| HMR-EFT (Res50) [16] | 54.3 | - | - |
| ROMP (Res50) | 53.5 | 89.3 | 105.6 |
| PARE (Res50) [20] | 52.3 | 82.9 | 99.7 |
| PARE (HR32) [20] | 50.9 | 82 | 97.9 |
| PyMAF (Res50)[58] | 49.0 | 79.7 | 94.4 |
| PyMAF (HR48) [58] | **47.1** | **78.0** | 91.3 |
| Baseline (Res50) | 52.4 | 85.2 | 103.6 |
| RoboSMPLX (Res50) | 49.8 | 80.8 | 96.7 |
| Baseline (HR48) | 50.3 | 84.5 | 101.5 |
| RoboSMPLX (HR48) | 48.5 | 80.1 | 95.2 |

Table 3: **Evaluation of the Face subnetwork**.

| Method | LQ Mean(mm) ↓ | HQ Mean(mm) ↓ |
|---|---|---|
| ExPose [6] | 2.27 | 2.42 |
| ExPose † | 2.46 | 2.38 |
| HMR | 2.18 | 2.11 |
| HMR † | 2.31 | 2.27 |
| HMR \* | 2.02 | 2.04 |
| PyMAF \* | 1.97 | 1.92 |
| RoboSMPLX | **2.12** | **2.08** |
| RoboSMPLX † | **2.12** | **2.10** |

Table 4: **PA-PVE/PVE errors of the Hand subnetwork under different positional augmentations.**

| | Normal | Transx +0.2x | Transx -0.2x | Transy +0.2y | Transy -0.2y | Scale 1.3x | Scale 0.7x |
|---|---|---|---|---|---|---|---|
| Hand4Whole [35] | 7.47/ 15.70 | 8.51/ 21.58 | 8.38/ 20.36 | 8.74/ 22.51 | 8.48/ 19.85 | 7.73/ 16.44 | 7.78/ 17.00 |
| RoboSMPLX | **7.24/ 15.23** | **7.27/ 15.62** | **7.36/ 15.59** | **7.28/ 15.50** | **7.34/ 15.50** | **7.49/ 15.90** | **7.45/ 16.51** |

## 5.1 Benchmarking Results

**Hand Subnetwork**. Table 1 compares the performance of the Hand subnetwork with different hand-only and whole-body methods. Our method outperforms that of our whole-body counterparts when trained with only the FreiHAND dataset (i.e. PIXIE, Hand4Whole, PyMAF) or under mixed datasets (i.e. Hand4Whole †, PyMAF †)[1] using an identical backbone. Prior research [33, 48] demonstrated that whole-body methods generally employ a parametric representation of the hand mesh, and are numerically inferior to the non-parametric representation used in recent hand-only methods [33, 28]. Despite such reported gap, RoboSMPLX manages to outperform mesh-based techniques, and achieve comparable results as the state-of-the-art METRO when using the same backbone (HRNet-64). Table 4 compares the estimation errors of the Hand subnetwork in Hand4Whole (current whole-body method with SOTA on hands) and RoboSMPLX under different positional augmentations on the FreiHAND test set. It is clear that RoboSMPLX exhibits much better robustness than Hand4Whole. More visualizations are provided in Figure 20 in Appendix.

**Body Subnetwork.** Table 2 compares the performance of the Body subnetwork across different methods on the 3DPW test set. We observe the competitiveness of RoboSMPLX in relation to other SMPL-based approaches. Besides, since the performance of various methods may significantly differ based on their backbone initialization, datasets and training strategies [38], we establish a baseline to evaluate the effectiveness of our added modules in Table 12 in Appendix. RoboSMPLX achieves a substantial improvement compared to the baseline.

**Face Subnetwork.** Table 3 compares the performance of the Face subnetwork for different methods on the Stirling3D test set. When training with the same dataset, RoboSMPLX outperforms ExPose. The performance of ExPose declines when training on multiple datasets, while RoboSMPLX can still keep low and consistent errors. Figure 8 in Appendix shows some qualitative results for the in-the-wild scenarios, which demonstrates the high generalization of RoboSMPLX. Table 5 compares the robustness of ExPose and RoboSMPLX under different positional augmentations. We also observe that RoboSMPLX has lower errors with different translation and scaling operations. More visualizations are provided in Figure 21 in Appendix.

**Whole-body Network.** We further provide results of the whole-body network on two benchmarks: EHF val set and AGORA test set in Table 6. On EHF, RoboSMPLX outperforms other full-body approaches, particularly in hand and face performance evaluations, and under different positional augmentations (Table 7). It gives subpar performance on AGORA as the predominant source of

---

[1]† denotes training with extra datasets in the following evaluation and tables.

Table 5: **3DRMSE errors of the Face subnetwork under different positional augmentations.**

| | Normal | Transx +0.2x | Transx -0.2x | Transy +0.2y | Transy -0.2y | Scale 1.3x | Scale 0.7x |
|---|---|---|---|---|---|---|---|
| ExPose [6] | 2.27 | 2.38 | 2.29 | 2.46 | 2.30 | 2.46 | 2.27 |
| RoboSMPLX | **2.12** | **2.20** | **2.17** | **2.13** | **2.18** | **2.24** | **2.10** |

Table 6: **Evaluation of wholebody network on EHF and AGORA test set.**

| Method | EHF | | | | | | AGORA | | | | | |
|---|---|---|---|---|---|---|---|---|---|---|---|---|
| | PVE ↓ | | | PA-PVE ↓ | | | PVE ↓ | | | | N-PVE ↓ | |
| | WB | H | F | WB | H | F | WB | B | F | LH/RH | WB | B |
| ExPose [6] | 77.1 | 51.6 | 35 | 54.5 | 12.8 | 5.8 | 217.3 | 151.5 | 51.1 | 74.9/71.3 | 265 | 184.8 |
| PIXIE [11] | 89.2 | 42.8 | 32.7 | 55 | 11.1 | **4.6** | 191.8 | 142.2 | 50.2 | 49.5/49.0 | 233.9 | 173.4 |
| Hand4Whole [35] | 76.8 | 39.8 | 26.1 | 50.3 | 10.8 | 5.8 | 135.5 | 90.2 | 41.6 | 46.3/48.1 | 144.1 | 96.0 |
| OSX (ViT-L) | 70.8 | 53.7 | 26.4 | **48.7** | 15.9 | 6.0 | **122.8** | **80.2** | **36.2** | 45.4/46.1 | **130.6** | **85.3** |
| PyMAF-X (HR48) | **64.9** | **29.7** | 19.7 | 50.2 | 10.2 | 5.5 | 125.7 | 84 | 35 | **44.6/45.6** | 141.2 | 94.4 |
| Ours | 73.7 | 34.9 | **17.8** | 49.7 | **10.0** | **4.6** | 132.3 | 85 | 39.4 | 45.3/46.1 | 138.2 | 91.5 |

error is the misidentification of individuals under intense person-person occlusion. We give detailed investigation in Appendix D.

## 5.2 Ablation Studies

**Contrastive loss**. We validate prior contrastive SSL methods [63, 46, 5] are not particularly adept at learning useful embeddings for human pose and shape estimation. Figures 9 – 12 in Appendix visualize the retrieved images based on the top-5 embedding similarity. They show that without labels, the model primarily extracts features based on background information instead of pose information. Table 8 shows the estimation errors of top-1 retrieved pose (COCO-train) and query pose (COCO-test) with different methods and contrastive loss functions. We observe that SimCLR has higher mean errors than the supervised training method HMR. These results are aligned with [5] that the representations learned through SSL are not transferable for human pose and shape estimation tasks. RoboSMPLX incorporates contrastive loss and positive samples ("HMR + $L_{con}$, +ve"), which can produce similar representations under varied augmentations, enhancing its robustness.

Table 9 shows the estimation errors when applying contrastive loss with different representations in Section 4.3: "pose" (a concatenated form of global orientation and rotational pose), "go+pose" (global orientation and rotational pose as separate entities), "keypoint" (3D joints regressed from the body model). We observe that regressed 3D joints are the most effective representation, as they encode both shape and pose information in a normalized space. In contrast, the representation of pose as relative rotation has a detrimental impact on the model performance. Incorporating positive samples ("pose, +ve" and "keypoint, +ve") bolsters contrastive learning, encouraging the model to generate similar representations under varied augmentations. Table 10 compares the model performance with different augmentations. Prior methods [63, 46] employed pose-variant augmentations (e.g., rotation and flipping), which can adversely affect the learning by altering the global orientation, and lead to increased errors ("pose") compared to "baseline". Conversely, color-variant, location-variant and their combination provide an improvement over the baseline, showing these augmentations are helpful.

**Location features.** Table 11 shows the ablation of different modules on the Hand subnetwork (ablation for the Body subnetwork is in Table 12 in Appendix). The baseline model is trained that randomly augments images with a scale factor of 0.2 and bounding box jitter of 0.2. We observe that training using *strongaug* with a larger scale and jitter factor harms the baseline performance. This is likely due to a domain shift. Hand4Whole [35] employs sampled features from positional pose-guided pooling (PPP) to predict

Table 11: **Ablation of different modules on Hand subnetwork. Results are trained and evaluated on FreiHAND.**

| | Supervision | PA-↓ | MPJPE↓ | PA-↓ | PVE↓ |
|---|---|---|---|---|---|
| Base (R50) | | 8.06 | 16.78 | 7.85 | 16.71 |
| Base (R50) + Strongaug | | 8.47 | 17.01 | 8.11 | 16.17 |
| Base (DR54) | | 7.8 | 15.57 | 7.67 | 15.72 |
| Base (DR54) | $L_{KS}$ | 7.68 | 15.8 | 7.62 | 16.29 |
| PPP [35] | $L_{KS}$ | 7.65 | 15.93 | 7.56 | 16.37 |
| LF | $L_{KS}$ | 7.52 | 15.84 | 7.56 | 16.15 |
| joints | $L_{KS}$ | 7.86 | 15.92 | 7.75 | 16.24 |
| LF (all) | $L_{KS}$ | 7.49 | 15.51 | 7.46 | 15.59 |
| LF (all) + $L_{con}$ | $L_{KS}$ | 7.48 | 15.01 | 7.32 | 15.29 |
| LF (all) + $L_{con}$, +ve | $L_{KS}$ | 7.42 | 14.88 | 7.16 | 14.57 |
| LF (all) | $L_{KS}, L_{segm}$ | 7.44 | 14.92 | 7.58 | 15.30 |
| LF (all) | $L_{KS}, L_{segm}, L_{proj}$ | 7.36 | **14.38** | 7.53 | 15.05 |
| **LF (all)+ $L_{con}$, +ve** | $L_{KS}, L_{segm}, L_{proj}$ | **7.33** | 14.59 | **7.02** | **14.11** |

pose parameters while shape and camera parameters only utilize backbone features. Our method focuses on explicitly learning the location and part silhouettes, utilizing sparse and dense supervision methods. This proves advantageous as the location information ("LF") improve the performance of pose and shape estimations, with the reduced joint and vertex errors of the regressed mesh. Moreover,

Table 7: **Wholebody, Hand and Face PA-PVE errors under different positional augmentations.**

| | Method | Normal | Transx +0.2x | Transx -0.2x | Transy +0.2y | Transy -0.2y | Scale 1.3x | Scale 0.7x |
|---|---|---|---|---|---|---|---|---|
| Hands | ExPose [6] | 14.39 | 17.36 | 17.86 | 14.93 | 17.21 | 14.15 | 14.56 |
| | PIXIE [11] | 14.68 | 15.05 | 16.11 | 15.32 | 15.85 | 14.52 | 14.79 |
| | Hand4Whole [35] | 10.83 | 11.15 | 11.34 | 10.50 | 13.70 | 10.77 | 11.25 |
| | OSX [27] | 15.97 | 16.42 | 16.55 | 16.94 | 17.86 | 15.91 | 17.24 |
| | RoboSMPLX | **10.00** | **10.37** | **10.21** | **10.16** | **12.49** | **9.98** | **10.19** |
| Face | ExPose [6] | 6.34 | 10.28 | 6.71 | 8.17 | 6.43 | 6.24 | 6.24 |
| | PIXIE [11] | 5.63 | 6.67 | 6.94 | 6.53 | 6.94 | 5.84 | 5.84 |
| | Hand4Whole [35] | 5.81 | 5.88 | 5.91 | 5.74 | 5.93 | 5.76 | 5.76 |
| | OSX [27] | 6.09 | 6.03 | 6.09 | 5.83 | 5.96 | 5.92 | 5.92 |
| | RoboSMPLX | **4.65** | **5.10** | **5.38** | **4.75** | **5.30** | **4.77** | **5.22** |
| Wholebody | ExPose [6] | 54.82 | 61.64 | 65.98 | 65.03 | 65.98 | 54.03 | 59.23 |
| | PIXIE [11] | 54.85 | 66.16 | 69.26 | 64.83 | 69.26 | 56.28 | 60.31 |
| | Hand4Whole [35] | 50.37 | 59.10 | 67.85 | 64.64 | 67.85 | 48.10 | 55.28 |
| | OSX [27] | **48.79** | **51.09** | 55.96 | 95.97 | **55.96** | **47.35** | **50.89** |
| | RoboSMPLX | 49.79 | 52.46 | **53.62** | 61.65 | 63.99 | 47.90 | 51.39 |

Table 8: **Ablation of contrastive learning methods and loss.**

| Scale factor | Mean ↓ | Std ↓ |
|---|---|---|
| SimCLR | 0.227 | 0.0915 |
| SimCLR (+ pose-variant aug.) | 0.230 | 0.0911 |
| SimCLR (+ background aug.) | 0.222 | 0.0959 |
| SimCLR (+ $L_{con}$) | 0.164 | 0.0772 |
| HMR | 0.140 | 0.0823 |
| HMR (+ $L_{con}$) | 0.124 | **0.0624** |
| HMR (+ $L_{con}$, +ve samples) | **0.119** | 0.0679 |

Table 9: **Ablation of different representation for contrastive loss.**

| Representation | PA-↓ | MPJPE↓ | PA ↓ | PVE↓ |
|---|---|---|---|---|
| baseline | 7.49 | 15.51 | 7.46 | 15.59 |
| pose | 8.11 | 15.81 | 7.67 | 16.08 |
| go + pose | 7.71 | 14.98 | 7.54 | 14.91 |
| keypoint | 7.48 | 15.01 | 7.32 | 15.29 |
| pose, +ve | 7.45 | 14.94 | 7.20 | **14.77** |
| keypoint, +ve | **7.31** | **14.62** | **7.18** | 15.01 |

Table 10: **Ablation of augmentation +ve samples, using pose rotation as representation.**

| Augmentation | PA-↓ | MPJPE↓ | PA -↓ | PVE↓ |
|---|---|---|---|---|
| baseline (no +ve) | 8.11 | 15.81 | 7.67 | 16.08 |
| color | **7.42** | 15.01 | **7.18** | 14.94 |
| pose | 8.59 | 16.96 | 8.15 | 17.21 |
| location | 7.80 | 15.98 | 7.46 | 15.56 |
| color + location | 7.45 | **14.94** | 7.20 | **14.77** |

we find that using location features "LF (all)" for predicting shape and camera parameters is also beneficial.

**Pixel alignment.** Tables 11 also shows that incorporating differential rendering and using projected segmentation loss ($L_{proj}$) for the mesh in RoboSMPLX helps to achieve lower PVE and MPJPE errors. It facilitates the learning of more precise body model and camera parameters to improve the alignment between the rendered 3D model and 2D image. Notably, metrics such as PVE and MPJPE errors is calculated after root alignment and may not sufficiently reflect the quality of mesh projection onto the image space. To offer a more precise analysis, we evaluate the discrepancies between the projected 2D vertices of the ground-truth and projected meshes. More quantitative and qualitative comparisons can be found in Appendix F.

## 6 Conclusion

In this paper, we introduce a new framework RoboSMPLX to advance the field of whole-body pose and shape estimation. It enhances the whole-body pipeline by learning more precise localization for part crops while ensuring that part subnetworks are robust enough to handle suboptimal part crops and produce reliable outputs. It achieves this goal with three innovations: accurate subject localization by explicitly learning both sparse and dense predictions of the subject, robust feature extraction with supervised contrastive learning, and accurate pixel alignment of outputs with differentiable rendering. Nevertheless, it is important to acknowledge that there are instances in which our framework exhibits limitations, such as (1) inaccurate beta estimation due to out-of-distribution data (children), (2) challenges posed by severe object-occlusion, (3) difficulties arising from person-person occlusion, and (4) the potential for prediction errors in multi-person scenarios, as exemplified by the cases detailed in Appendix G. These challenges represent important avenues for future refinement of our approach.

There are several potential avenues for future research. First, the current approach does not deliberately select negative samples during training. Future work could explore if hard mining by intentionally selecting similar poses in a batch could enhance learning. Second, the careful selection of augmentations is essential. While augmentations that modify the global orientation, such as flipping and rotation, have proven detrimental and are not employed, the effects of individual augmentations and their combinations are not examined. Future research could explore the potential for automatically determining the optimal selection of augmentations to achieve improved performance. Additionally, simplifying the complex framework without sacrificing performance is a beneficial direction for future work. Lastly, considering that videos are a prevalent input format, the integration of video-based estimation can contribute to bolstering model robustness can enhance model robustness, alleviate depth ambiguity, and improve temporal consistency.

## Acknowledgements

This research/project is supported by the National Research Foundation, Singapore under its AI Singapore Programme (AISG Award No: AISG3-PhD-2023-08-049T). This study is also supported by the Ministry of Education, Singapore, under its MOE AcRF Tier 2 (MOE-T2EP20221-0012), NTU NAP, and under the RIE2020 Industry Alignment Fund – Industry Collaboration Projects (IAF-ICP) Funding Initiative, as well as cash and in-kind contribution from the industry partner(s). We sincerely thank the anonymous reviewers for their valuable comments on this paper.

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
