## Overview

This supplementary material presents more details and additional results not included in the main paper due to the page limitation. The list of items included are:

- Description of augmentation settings for robustness benchmarking in Section A
- More experiment setup and details in Section B
- Comparison with PyMAF-X in Section C
- Analysis of the subpar performance on AGORA test set in Section D
- Ablation of different modules on the Body subnetwork in Section E
- Quantitative and qualitative and comparisons for pixel alignment in Section F
- Examples of failure cases in Section G
- Analysis of embedding similarity in Section H
- Discussion on pose (rotation) versus keypoint representation in Section I
- Extra comparisons against SOTA body networks in Section J
- Training and inference time in Section K
- Accuracy of derived part bounding boxes in Section L
- Qualitative comparisons of `RoboSMPLX`'s Hand, Face and Body subnetworks under augmentations in Section M
- Quantitative and qualitative comparisons of `RoboSMPLX`'s wholebody model in Section N

## A   Augmentation Settings for Robustness Benchmarking

In the selection of augmentations, we opted for a set of ten commonly encountered augmentations that could be benchmarked in a controlled setting. We also ensure that the selected values for manipulation fall within a realistic range. We used the following augmentations:

1. Vertical translation: We shifted the image by factors relative to the image size. For instance, a +0.1 shift corresponds to a 10% upward movement, while a -0.1 shift represents a 10% downward movement. Our boundaries were set at $\pm 0.3$ to ensure that majority of the subject remains visible within the image frame.

2. Horizontal translation: We manipulated the image by factors relative to the image size. A shift of +0.1 denotes a 10% move to the right, while -0.1 indicates a 10% shift to the left. We imposed a $\pm 0.3$ limit to keep the majority of the subject within the image.

3. Scale: We adjusted the person's crop using factors relative to the bounding box size. For example, a factor of +0.1 leads to a 10% size reduction, resulting in a tighter crop, while a -0.1 factor enlarges the crop size by 10%. A $\pm 0.5$ boundary was set to maintain visibility of the majority of the person within the image.

4. Low Resolution: The resolution of the cropped image was modified by factors related to the image size. A 2.0 factor signifies that the image was downsampled to half its original size before being upsampled back, reducing the resolution by a factor of 2.0.

5. Rotation: The image was manipulated by various rotations up to degrees of $\pm 60$.

6. Hue: The image hue was altered by converting the image to HSV format, cyclically shifting intensities in the hue channel (H), and converting back to the original image mode. Hue adjustments were limited to $\pm 0.5$.

7. Sharpness: Sharpness was controlled by introducing an enhancement factor. A factor of -1.0 leads to a blurred image, while +1.0 results in a sharpened image, with 0.0 leaving the image unaltered. This effect is achieved by blending the source image with the degraded mean image.

8. Grayness: The degree of grayness was adjusted by introducing an enhancement factor. A factor of -1.0 results in a completely grayed image, while +1.0 leads to a whitened image, with 0.0 leaving the image unaltered. This effect is achieved by blending the source image with its gray counterpart. The limit was set to $\pm 0.5$, as the subject becomes unidentifiable at extremes of $\pm 1.0$.

9. Contrast: This was controlled by introducing an enhancement factor. A factor of -1.0 leads to a completely grayed image, while +1.0 results in a whitened image, with 0.0 leaving the image unaltered. This effect is achieved by blending the source image with the degraded mean image. The limit was set to $\pm 0.5$, as the subject becomes unidentifiable at extremes of $\pm 1.0$.

10. Brightness: The brightness of the image was adjusted by introducing an enhancement factor. A factor of -1.0 results in a black image, while +1.0 leads to a white image, with 0.0 leaving the image unaltered. This effect is achieved by blending the source image with the degraded black image. The limit was set to $\pm 0.5$, as the subject becomes unidentifiable at extremes of $\pm 1.0$.

## B    More Experiment Setup

This section includes extra description of each submodule and implementation details.

**Body subnetwork.** The body image is downsampled from the original image to reduce the computational cost, resulting in $I_b \in R^{3 \times 256 \times 256}$. The Body subnetwork outputs 3D body joint rotations $\theta_b \in R^{21 \times 3}$, global orientation $\theta_{bg} \in R^3$, shape parameters $\beta_b \in R^{10}$, camera parameters $\pi_b \in R^3$, and whole-body joints $K \in R^{137 \times 3}$. Hand and face bounding boxes are then derived from the face and hand keypoints. Width and height are determined from the x-y range of the keypoints, and the center is the aggregated mean of the keypoints. High resolution crops are used for hand and face inputs following ExPose and PIXIE. In line with ExPose [6] and PIXIE [11], hand and face input images are obtained from high resolution crops to utilize the information available from the original image instead of the downsampled image.

**Hand subnetwork.** After obtaining the cropped hand images $I_h \in R^{3 \times 256 \times 256}$, the left hand images are flipped to match the orientation of the right hands before being input to the Hand subnetwork. After predicting the 3D finger rotations $\theta_h \in R^{15 \times 3}$, the outputs of the flipped left hands are reverted to their original orientation. The 3D finger rotations of the left and right hands are denoted as $\theta_{rh}$ and $\theta_{lh}$ respectively. When training the full version on hand datasets, we also output the global orientation $\theta_{hg} \in R^3$, shape $\beta_h \in R^{10}$ and camera $\pi_h \in R^3$. However, these branches are discarded during whole-body estimation and training.

**Face subnetwork.** This subnetwork generates the 3D jaw rotation $\theta_f \in R^3$ and expression $\psi_f \in R^{10}$ from the cropped face image $I_f \in R^{3 \times 256 \times 256}$. When training the full version on face datasets, additional outputs include the global orientation $\theta_{fg} \in R^3$, shape $\beta_f \in R^{50}$, expression $\psi_f \in R^{50}$ and camera $\pi_f \in R^3$. These branches are also discarded during whole-body estimation and training.

**Implementation details.** The training and evaluation of our model builds upon the MMHuman3D framework [7]. For model initialization, we pre-train the ResNet backbone on the MSCOCO 2D whole-body human pose dataset. During training, we use the Adam optimizer with a mini-batch size of 32 and apply data augmentations, e.g., scaling, rotation, random horizontal flip, and color jittering. The initial learning rate is set to $10e-4$, decayed by a factor of 10 at the later epoch. We use the SMPL, MANO, FLAME and SMPL-X body models for the training of body, hand, face and wholebody respectively. Further details will be provided in our code.

## C    Comparison with PyMAF-X

Below we provide detailed discussions and comparisons with PyMAF-X [58].

1. Acquisition of part bounding boxes: PyMAF-X relies on an off-the-shelf whole-body pose estimation model (OpenPifpaf) to obtain whole body 2D keypoints of the person in the image, from which part crops are derived. During the EHF evaluation, PyMAF-X employs hand and face bounding boxes derived from OpenPose keypoints. In contrast, our method and other works (ExPose [6], PIXIE [11], Hand4Whole [35] and OS-X [27].) encompass a self-integrated module designed to extract hand and face bounding boxes directly from the image.

2. Operational efficiency: Openpifpaf imposes extra computation during inference, making PyMAF-X less efficient than our method. Please refer to Section K in Appendix.

3. Network architecture: Due to the diverse backbone and dataset combinations utilized, it is challenging for us to make whole-body network comparisons. In Table 1, we focus

on contrasting RoboSMPLX's Hand subnetwork with PyMAF's Hand subnetwork. Both networks are trained and evaluated on the same backbone and dataset, FreiHAND. In this context, our method surpasses PyMAF.

4. Performance: On the EHF metrics, our performance lags behind PyMAF-X. This could potentially arise from variations in the training datasets employed. While the training pipeline of the body network for PyMAF-X has been disclosed, the training specifics for hands and face and the methodology to integrate hand, face, and body module PyMAF-X, remains undisclosed. We intend to replicate with similar training datasets in the future.

## D   Analysis of performance on AGORA test set

Figure 13 visualise samples with significant errors during training. AGORA contains extensive person-to-person occlusion, frequently leading to substantial overlap between the target individual (marked with red vertices) and another person. In cases that experienced large errors, the model often incorrectly identified the target individual as the person situated in the forefront (model predictions marked with green vertices), thereby introducing instability throughout the training process due to the model's challenge in accurately discerning the intended subject.

We also added qualitative comparisons of RoboSMPLX under varying scales and alignments as shown in Figures 14. We demonstrate that RoboSMPLX produces better pixel alignment of the body, and more accurate hand and face predictions where the target person has been accurately identified.

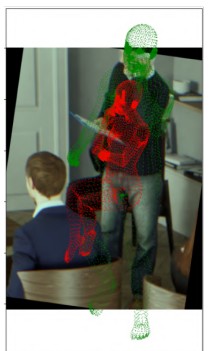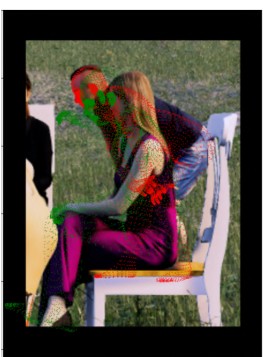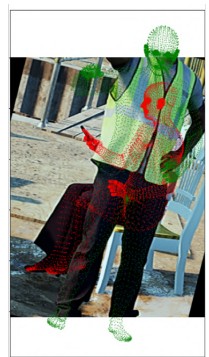

Figure 13: **Visualisation of samples with high errors at train time.** Red vertices indicates the target person while green vertices are the model's predictions.

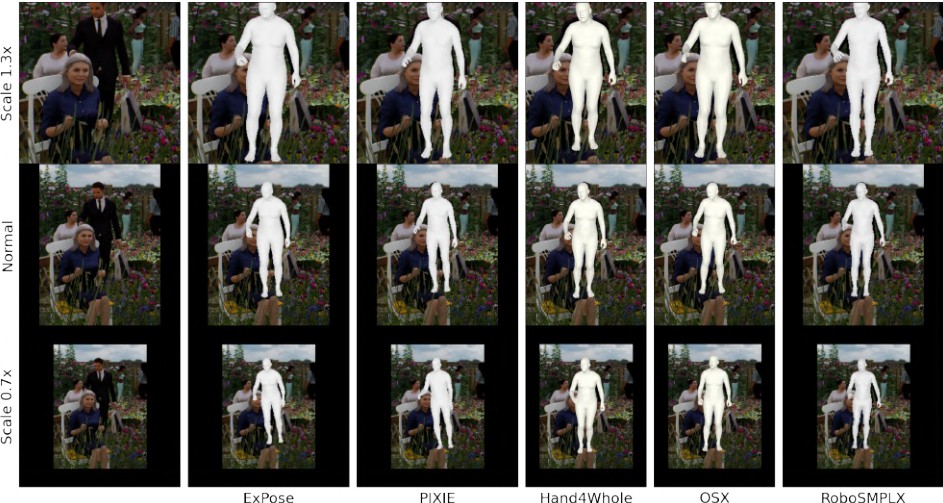

Figure 14: **Visualisation of Expose [41], PIXIE [11], Hand4Whole [35], OS-X [27] and** RoboSMPLX **under different scales and alignment on AGORA validation set.**

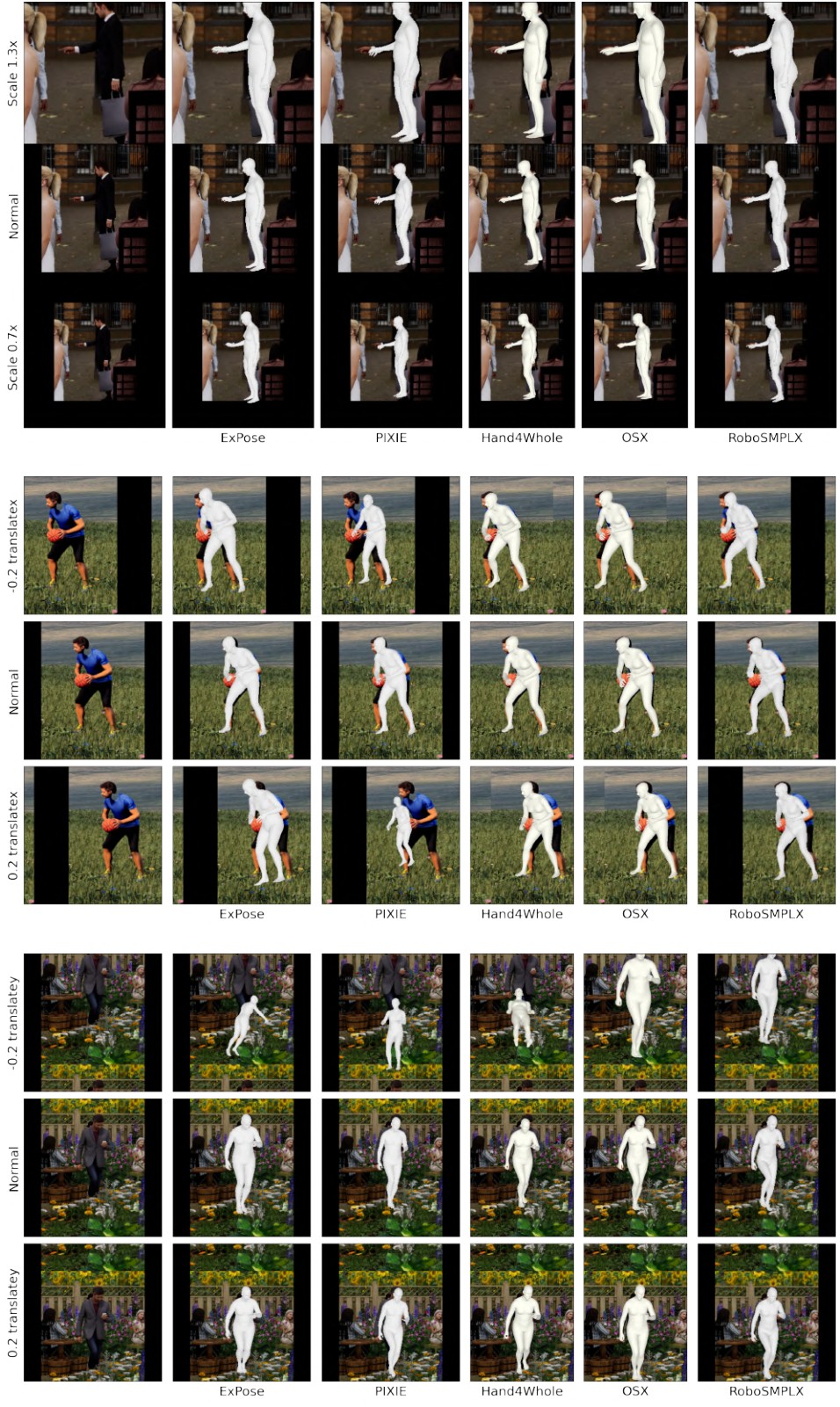

Figure 14: **Visualisation of Expose [41], PIXIE [11], Hand4Whole [35], OS-X [27] and** RoboSMPLX **under different scales and alignment on AGORA validation set (cont.).**

# E    Ablation on Body Subnetwork

Table 12 shows the ablation of different modules on the Body subnetwork. The conclusions derived from the Hand ablation study (Table 11) extends to the Body subnetwork as well.

Table 12: **Ablation of different modules on Body subnetwork. Results are trained on EFT-COCO and tested on 3DPW test set.**

|  | loss | representation | PA- | MPJPE |
|---|---|---|---|---|
| Baseline (HMR) | - | - | 60.8 | 96.2 |
| LF (all) | - | - | 56.7 | 105.7 |
| LF (all), $L_{con}$ | L1 | pose | 55.9 | 90.9 |
| LF (all), $L_{con}$ | MSE | pose | 58.5 | 93.9 |
| LF (all), $L_{con}$ | SmoothL1 | pose | 56.6 | 92.5 |
| LF (all), $L_{con}$ | L1 | pose(rot6d) | 58.9 | 95.0 |
| LF (all), $L_{con}$ | L1 | pose + go | 76.8 | 118.9 |
| **LF (all), $L_{con}$, +ve** | L1 | keypoints | **55.4** | **90.56** |

# F    Qualitative and quantitative comparisons for pixel alignment

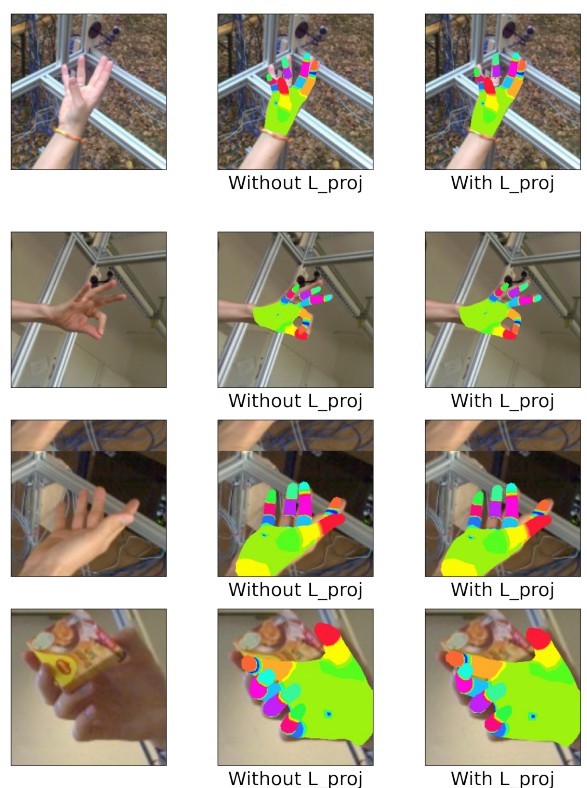

Figure 15: **(C) Visualisation from training with and without $L_{proj}$.**

Prevailing metrics such as Per Vertex Error (PVE) and Mean Per Joint Position Error (MPJPE) do not incorporate alignment measurement in their evaluation. Before these metrics are computed, the mesh undergoes root alignment, but this process does not necessarily reflect the level of alignment accuracy when the mesh is reprojected back into the image space.

Moreover, for pose and shape estimation methods, the absence of ground-truth camera parameters implies that there is no direct supervision for these parameters. Camera parameters are, instead, often weakly supervised through the supervision of projected keypoints (derived from regressed joints of the mesh and predicted camera parameters) and the ground-truth 2D joints by ensuring their alignment. This only provides a sparse supervision. To enhance better learning of camera, pose and shape parameters, pixel alignment strategy is introduced, which ensures denser supervision.

Presently, there's an absence of a metric tailored to gauge the degree of pixel alignment of a mesh in this context. We included qualitative examples of training with and without $L_{proj}$, and demonstrate that the projection of vertices results in better pixel alignment (Figure 15).

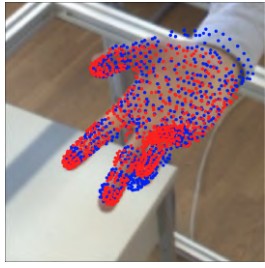

Figure 16: **Projected Vertex Errors is measured as distance between projected ground-truth (red) and predicted (blue) vertices in image space.**

| Method | Projected Vertex Errors ↓ |
|---|---|
| HMR (no PA) | 11.796 |
| HMR + PA (vertex) | 11.211 |
| HMR + PA (part-seg) | **10.298** |

Table 13: **Results of Projected Vertex Errors under different Part Alignment (PA)**

To provide quantitative analysis, we measure errors between the projected 2D vertices of ground-truth and projected meshes (Figure 16). From Table 13, it is evident that omitting the pixel alignment module leads to suboptimal outcomes. In contrast, our pixel alignment strategy, leveraging rendered segmentation maps, showcases better performance than using vertex loss as supervision.

## G  Failure cases

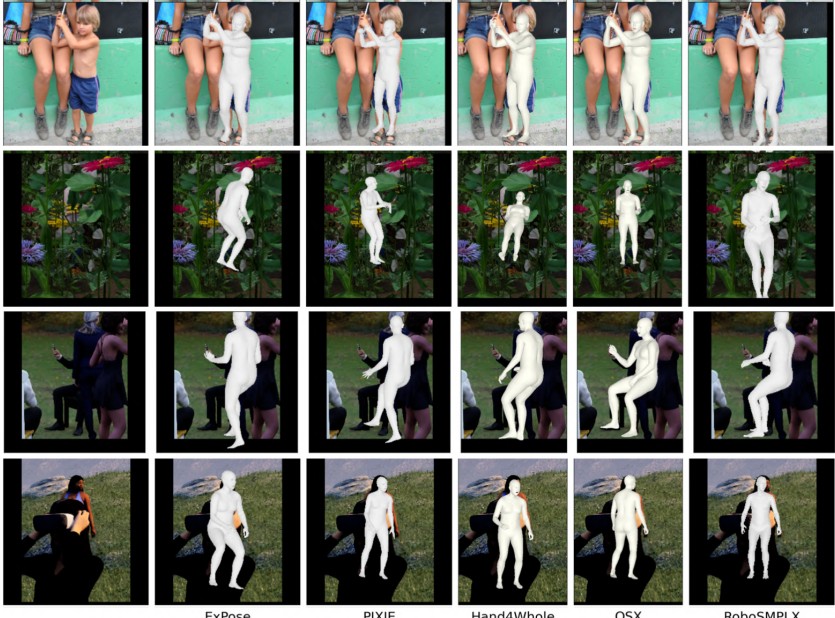

Figure 17: **Examples of failure cases. (1) Inaccurate beta estimation due to out-of-distribution data (children) (2) Severe object-occlusion (3) Person-person occlusion (4) Prediction for wrong person in multi-person scenarios.**

## H  Embedding similarity

Our use of the contrastive module is motivated by the need to constrain/maintain the same pose feature for different augmentations, to avoid domain shift caused by strong augmentation alone. The experiments show that the use of strong augmentation alone for training can lead to performance deterioration, while combining it with the contrastive loss consistently results in minimal errors (Table 11).

Table 14: **Ablation of CFE module on Hand Subnetwork. (This excludes Localization and Pixel Alignment Module). Results are trained and evaluated on FreiHAND.**

| Method | PA-MPJPE ↓ | MPJPE ↓ | PA-PVE ↓ | PVE ↓ | Pose embedding distance ↓ |
|---|---|---|---|---|---|
| Model 0: HMR | 8.06 | 16.78 | 7.85 | 16.71 | 0.132 |
| Model 1: HMR + Strongaug | 8.47 | 17.01 | 8.11 | 16.17 | 0.138 |
| Model 2: HMR + Strongaug + CL | **7.79** | **15.68** | **7.41** | **15.27** | 0.101 |

To illustrate this further, we delved into a visualization of the pose similarity for augmented samples. The findings reveal that augmented samples are perceived as dissimilar in both Model 0 and Model 1 (Table 14). Yet, when examining Model 2, a marked increase in embedding similarity is evident, underscoring the advantage of the contrastive approach.

# I   Discussion of pose versus keypoint representation

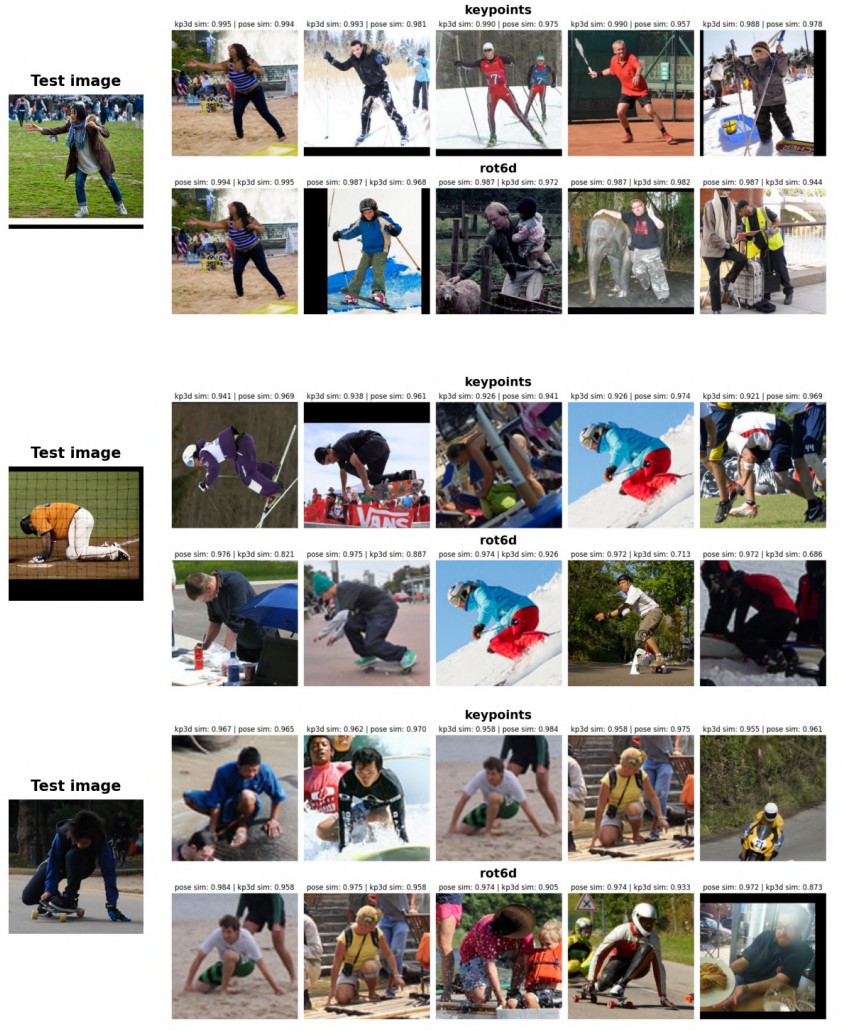

Figure 18: **Comparison of keypoints and pose representations.**

Figure 18 compares two distinct methods for image retrieval: one based on pose similarity (rot6d representation) and the other based on keypoint similarity. Samples with high keypoint similarity tends to have comparatively high pose similarity. On the contrary, similar pose representation might have considerably lower joint representation. This could occur due to the accumulation of minor discrepancies in joint rotations which, over time, may result in significant disparities in the keypoints.

This results in instances where the overall pose of the retrieved sample is very high to the query sample, but the keypoints may not coincide as accurately. Meanwhile, using keypoint representation would result in samples that have demonstrate improved alignment with the query image, presenting a more accurate correspondence.

This could explain why using regressed keypoints as representation have better performance (Table 9). Clustering based on keypoint similarity is more effective than pose similarity, as pose representation might be susceptible to minor shifts in joint rotations.

## J   Extra comparisons against SOTA body networks

There are many factors affecting training,including, but not limited to, the choice of backbone, datasets employed, and specific protocols executed during evaluation. Specifically, with regards to 3DPW, various protocols—ranging from fine-tuning (3DPW Protocol 1), collective training, to omission during training (3DPW Protocol 2)—have a large influence on 3DPW results in the evaluation process.

In Table 15, we outperform HybrIK when using the same backbone (HRNet-W48) and not fine-tuning on 3DPW (3DPW Protocol 2). Notably, CLIFF incorporated 3DPW within its training datasets. Given that our approach and that of both HybrIK and CLIFF do not utilize identical dataset combinations, a direct comparison becomes inherently challenging.

Table 15: Evaluation of HybrIK, CLIFF and our network on 3DPW. Our results are also available in Table 2.

| Method | Backbone | F-T on 3DPW | PA-MPJPE (3DPW) | MPJPE (3DPW) |
|--------|----------|-------------|-----------------|--------------|
| HybrIK | HRNet-W48 | No | 48.6 | 88.0 |
| HybrIK | HRNet-W48 | Yes | 41.8 | 71.3 |
| CLIFF | Res-50 | Trained with 3DPW | 45.7 | 72.0 |
| CLIFF | HRNet-W48 | Trained with 3DPW | 43.0 | 69.0 |
| Ours | Resnet-50 | No | 49.8 | 80.8 |
| Ours | HRNet-W48 | No | 48.5 | 80.1 |

We have provided qualitative comparisons of body-only methods under different scale and alignment in Figure 21. Below, we provide quantitative evaluations of our method with HMR, SPIN and PARE (Table 16). Our method is able to achieve better performance under different scales and alignment.

Table 16: Evaluated on 3DPW (PA-MPJPE/MPJPE) under different scales and alignment. * denote the same dataset combination

| | Normal | Transx +0.2x | Transx -0.2x | Transy +0.2y | Transy -0.2y | Scale 1.3x | Scale 0.7x |
|--|--------|--------------|--------------|--------------|--------------|-----------|-----------|
| HMR | 67.53/112.34 | 77.31/141.70 | 77.06/ 138.51 | 86.57/ 151.15 | 77.26/148.33 | 68.46/ 117.1 | 75.38/ 124.79 |
| SPIN | 57.54/94.11 | 70.14/122.56 | 68.67/ 120.04 | 73.08/ 111.33 | 70.64/133.2 | 61.08/ 103.60 | 61.63/ 99.6 |
| PARE (HR32) * | **49.3/81.8** | 74.9/139.2 | 77.1/ 141.7 | 59.1/92.3 | 64.2/ 109.7 | 54.7/86.9 | **50.5/ 83.9** |
| Ours (R50) * | 49.8/80.8 | **67.2/117.2** | **67.72/111.5** | **56.4/90.0** | **62.8/105.6** | **50.2/84.6** | 50.8/ 82.4 |

## K   Training and inference time

Our model was trained utilizing a cluster of 8xTesla V100-SXM2-32GB GPUs. Specific to the training duration, the hand models required approximately one day, whereas the body and face models necessitated two days. The joint training process was completed within a day.

We measure the model size, computation complexity and inference time for different models including ours, as shown in Table 17. Although our framework has sophisticated design, it has comparable inference speed as others, validating its efficacy.

## L   Quantitative evaluation of predicted bounding box accuracy

To assess the precision of predicted part bounding boxes on the EHF test set, we utilized Intersection over Union (IoU) as our evaluation metric (see Figure 19). Our method achieved the highest IoU

Table 17: These results are tested on RTX3090. FLOP refers to the total number of floating point operations required for a single forward pass. The higher the FLOPs, the slower the model and hence low throughput. Inference Time is obtained by averaging across 100 runs.

| | Total parameters (M) | GFLOPs | Inference time (s) |
|---|---|---|---|
| ExPose | 26.06 | 21.04 | $0.1330 \pm 0.0050$ |
| PIXIE | 109.67 | 24.23 | $0.1670 \pm 0.0065$ |
| Hand4Whole | 77.84 | 17.98 | $0.0709 \pm 0.0022$ |
| OSX | 422.52 | 83.77 | $0.1998 \pm 0.0028$ |
| PyMAF-X (gt H/F bbox) | 205.93 | 33.41 | $0.2194 \pm 0.0027$ |
| PyMAF-X + OpenPipaf | 205.93 + 115.0 | 33.41 + 120.52 | $0.2727 \pm 0.0136$ |
| RoboSMPLX | 120.68 | 29.66 | $0.2008 \pm 0.0220$ |

scores, as demonstrated in Table 18. It is important to note that in the OSX implementation, the hand and face features are cropped from the body features rather than directly from the image.

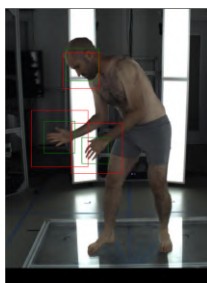

Figure 19: **Calculation for the Face, LHand and RHand IoU scores for ground-truth (green) and predicted (red) part bounding boxes.**

| Method | Face IoU | LHand IoU | RHand IoU |
|---|---|---|---|
| ExPose | 0.61 | 0.23 | 0.31 |
| PIXIE | 0.66 | 0.34 | 0.36 |
| Hand4Whole | 0.75 | 0.41 | 0.45 |
| OSX | 0.70 | 0.38 | 0.41 |
| RoboSMPLX | 0.86 | 0.52 | 0.55 |

Table 18: **Results for IoU of the predicted part bounding boxes on the EHF test set.**

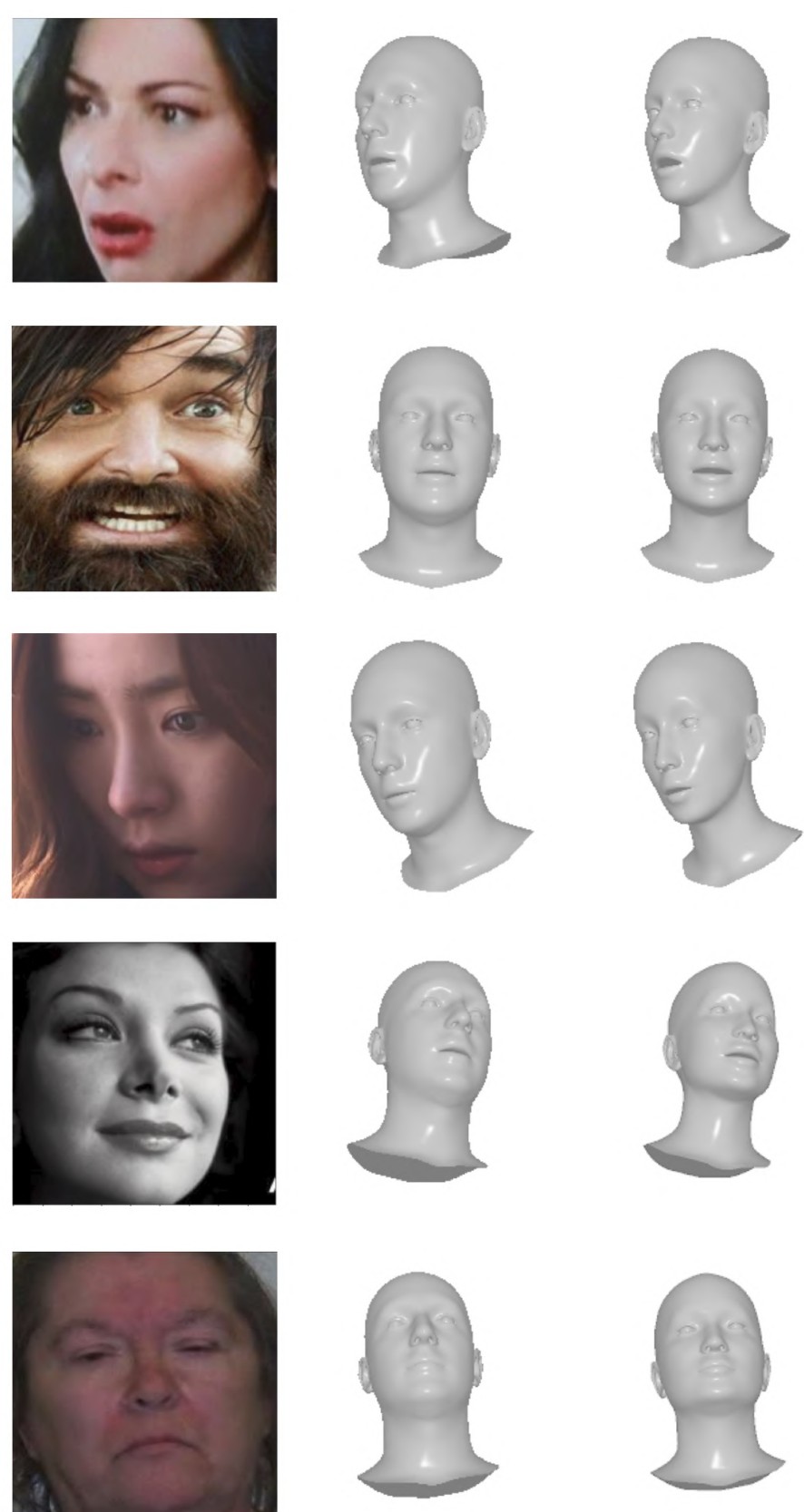

Figure 8: **Inference on AffectNet validation images using Expose [6] and** RoboSMPLX**'s Face subnetwork.**

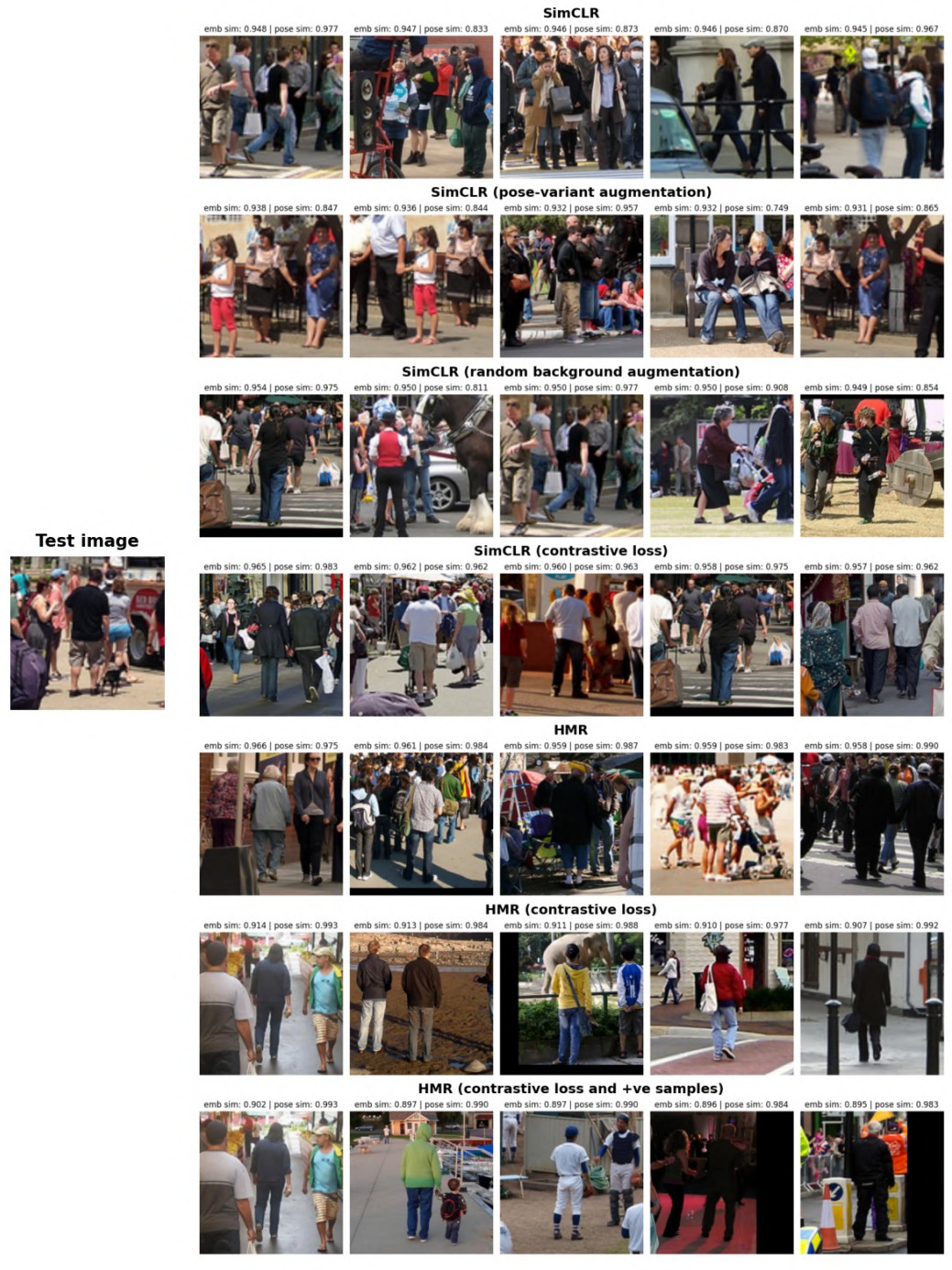

Figure 9: **Left: Query image from the EFT-COCO-Test set, Right: Retrieved image from the EFT-COCO-Train set ordered in descending embedding similarity.**

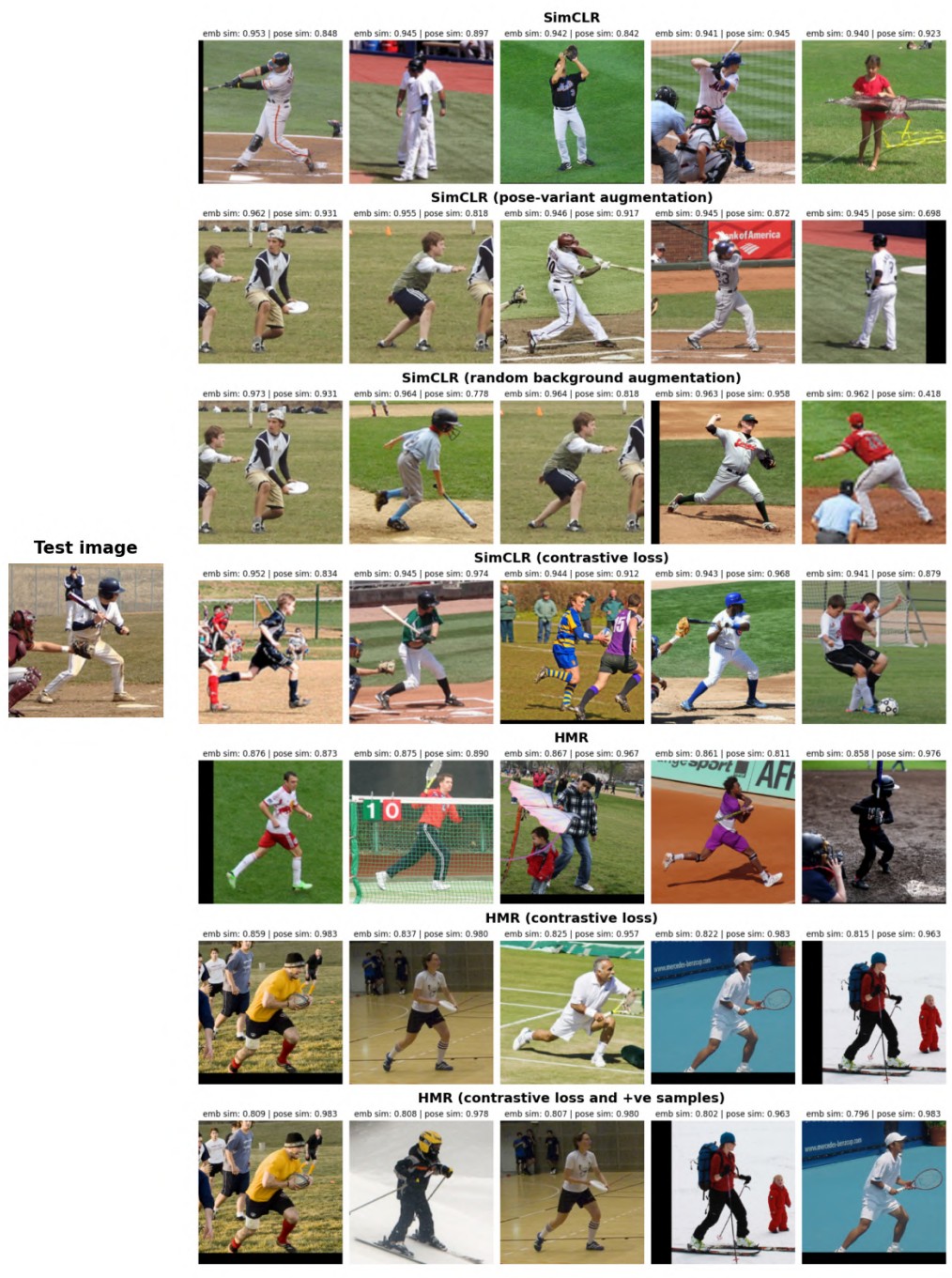

Figure 10: **Left: Query image from the EFT-COCO-Test set, Right: Retrieved image from the EFT-COCO-Train set ordered in descending embedding similarity.**

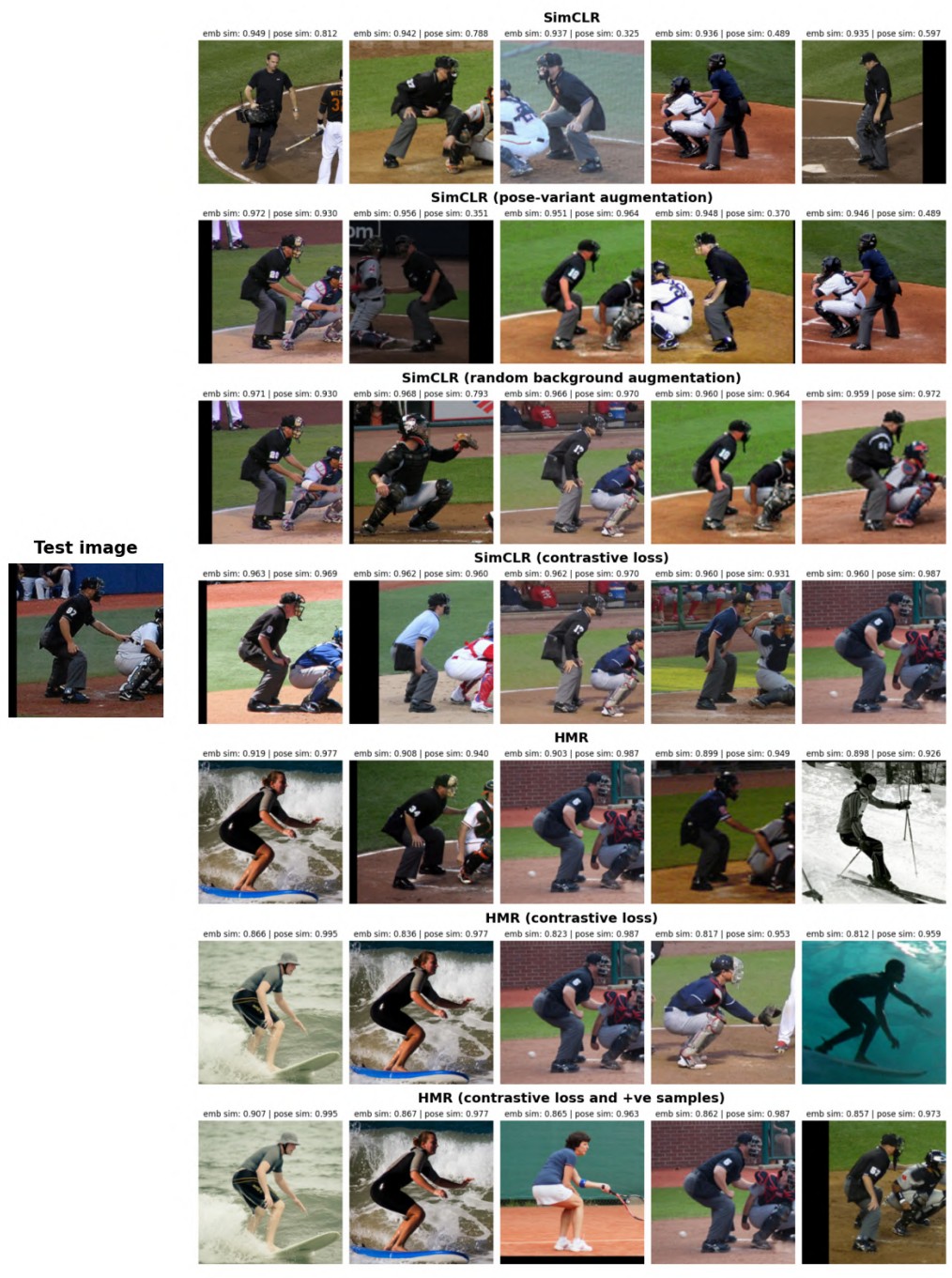

Figure 11: **Left: Query image from the EFT-COCO-Test set, Right: Retrieved image from the EFT-COCO-Train set ordered in descending embedding similarity.**

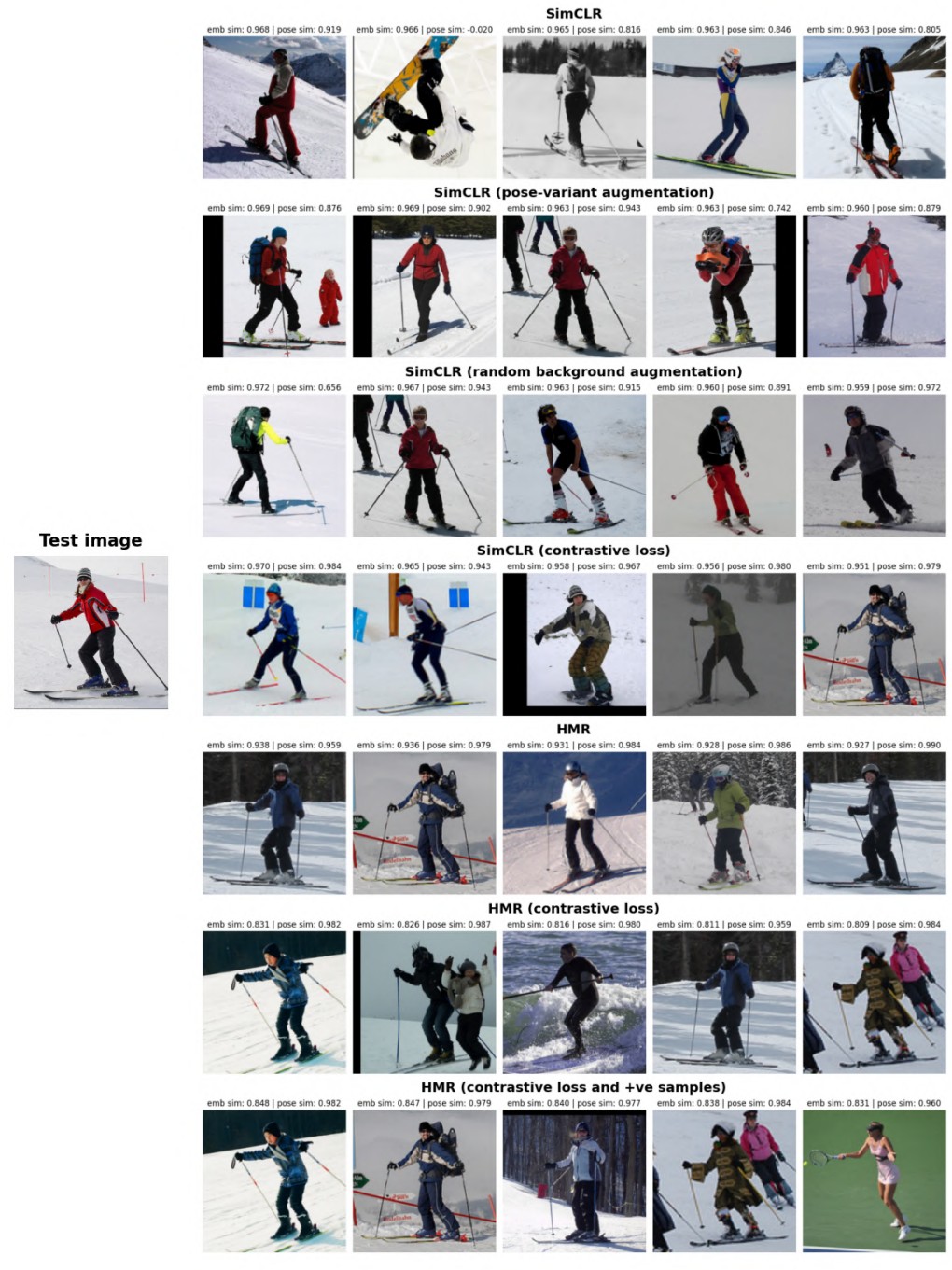

Figure 12: **Left: Query image from the EFT-COCO-Test set, Right: Retrieved image from the EFT-COCO-Train set ordered in descending embedding similarity.**

# M  Qualitative comparisons for different models under augmentation

We show qualitative comparisons of RoboSMPLX's Hand (Figure 20), Face (Figure 21) and Body (Figure 21) subnetwork to existing models under different positional augmentations.

In general, RoboSMPLX s' subnetworks demonstrate better pixel alignment and are less sensitive to changes in scale and alignment.

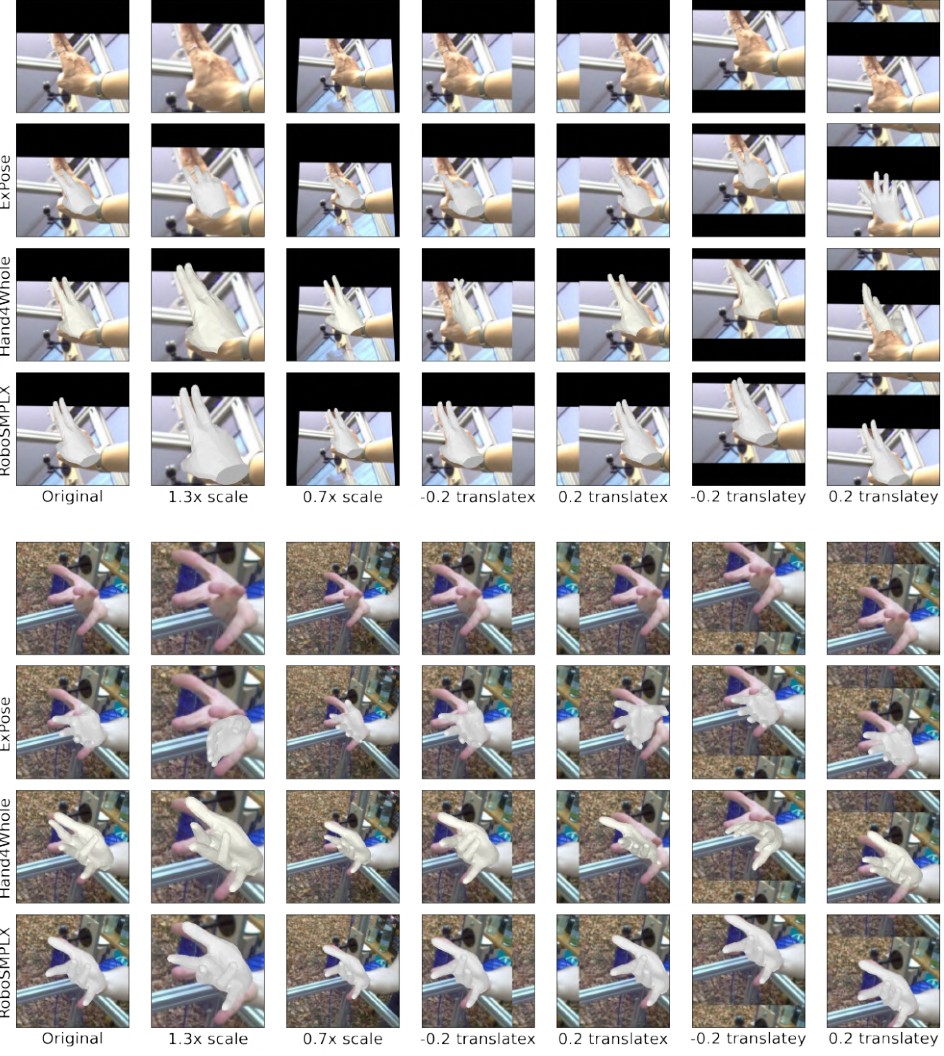

Figure 20: **Comparison of ExPose [6], Hand4Whole [35] and RoboSMPLX's Hand subnetwork under various augmentations on FreiHAND test set.**

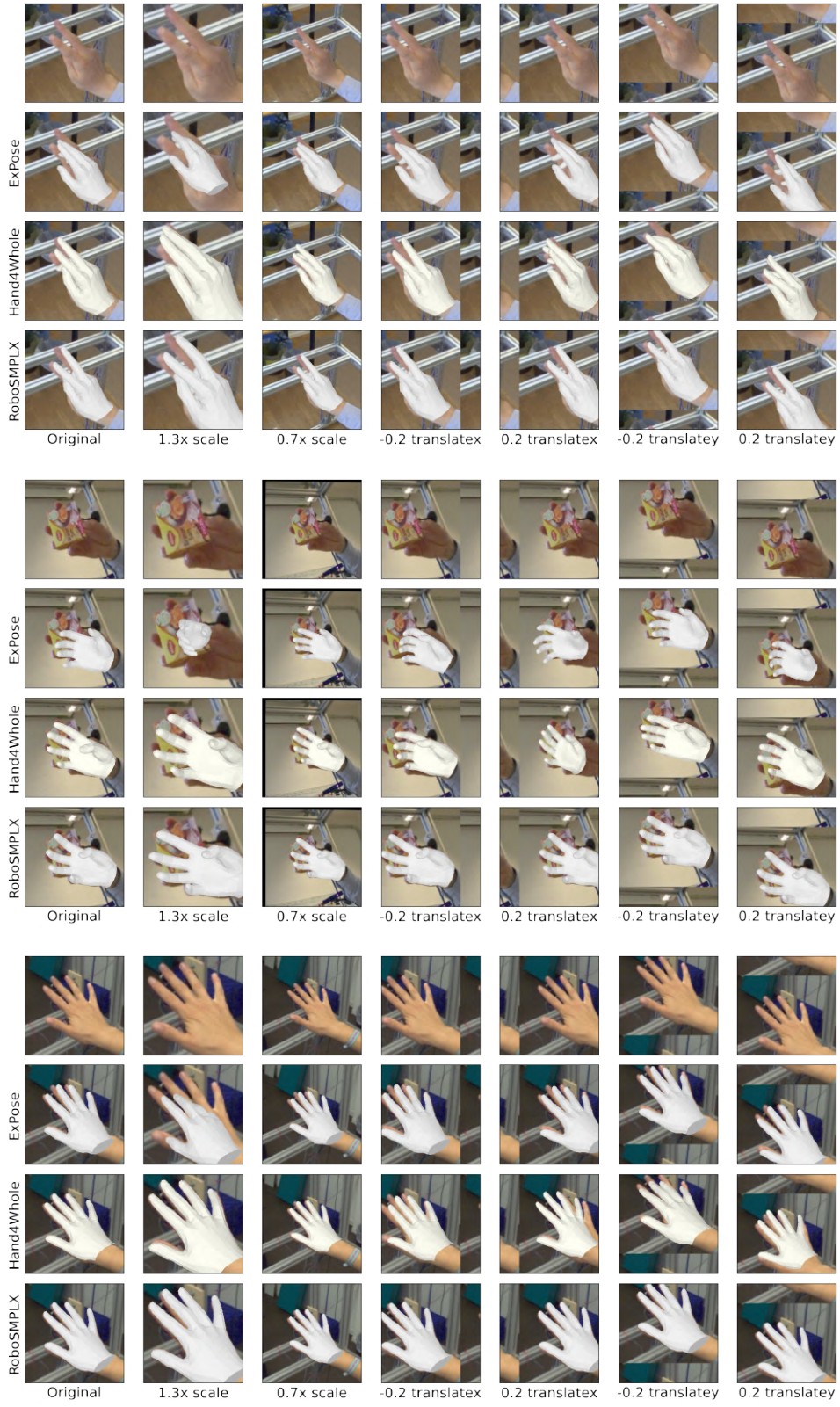

Figure 20: **Comparison of of ExPose [6], Hand4Whole [35] and** RoboSMPLX**'s Hand subnetwork under various augmentations on FreiHAND test set (cont.)**

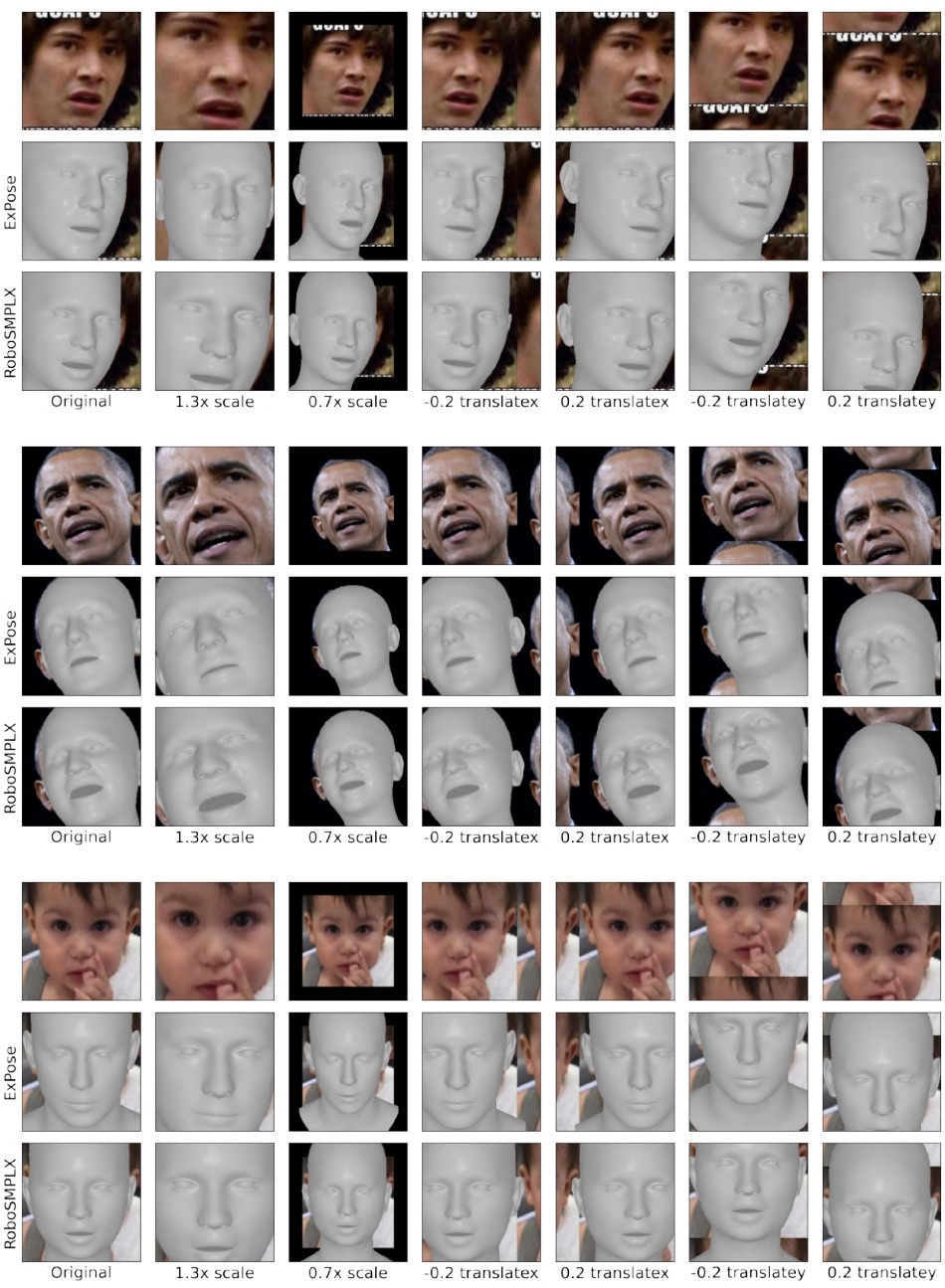

Figure 21: **Comparison of ExPose [6] and** RoboSMPLX**'s Face subnetwork under various augmentations on AffectNet val set.**

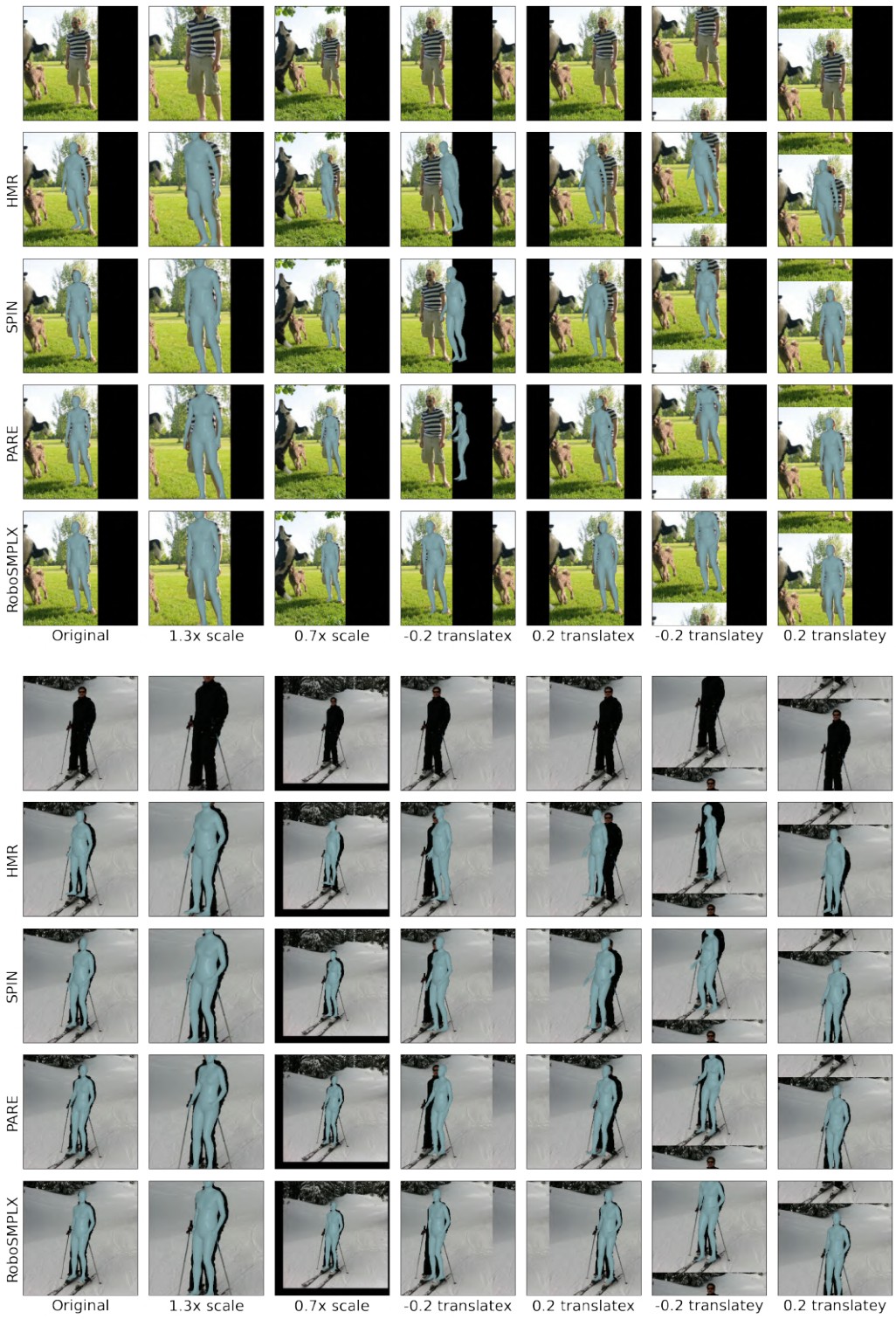

Figure 21: **Comparison of HMR [17], SPIN [22], PARE[20] and** `RoboSMPLX`**'s Body subnetwork under various augmentations on COCO validation set.**

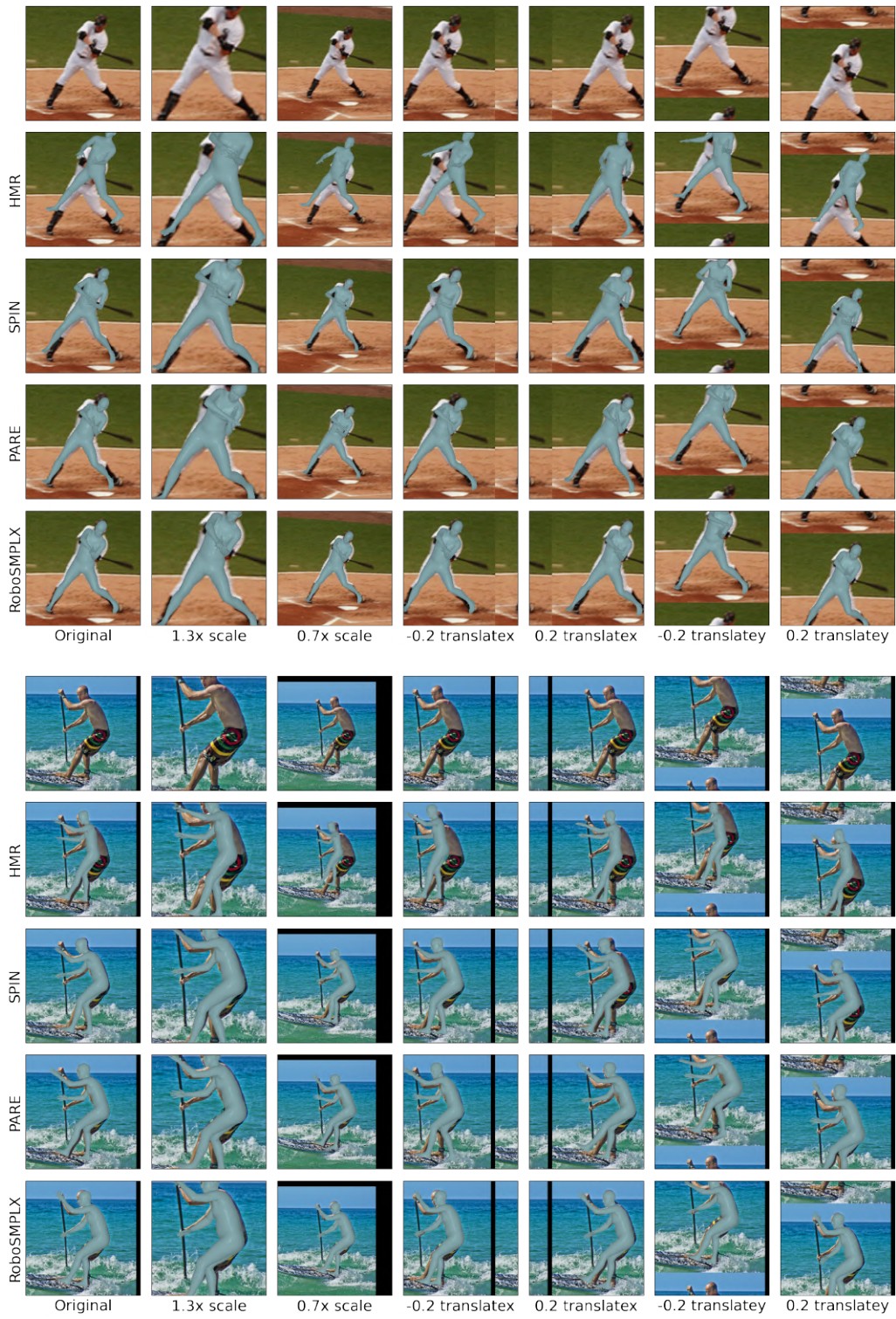

Figure 21: **Comparison of HMR [17], SPIN [22], PARE[20] and** `RoboSMPLX`**'s Body subnetwork under various augmentations on COCO validation set (cont.)**

# N  Quantitative and qualitative comparisons for wholebody models

We provide quantitative comparisons of wholebody models under different augmentations on EHF test set in Figures 22 and 23. We also added qualitative comparisons under different scale and alignment on EHF test set in Figures 24 to 26. We demonstrate that RoboSMPLX produces better pixel alignment of the body, and more accurate hand and face predictions. In addition, we inference on in-the-wild examples on COCO-validation set in Figures 27 and 28.

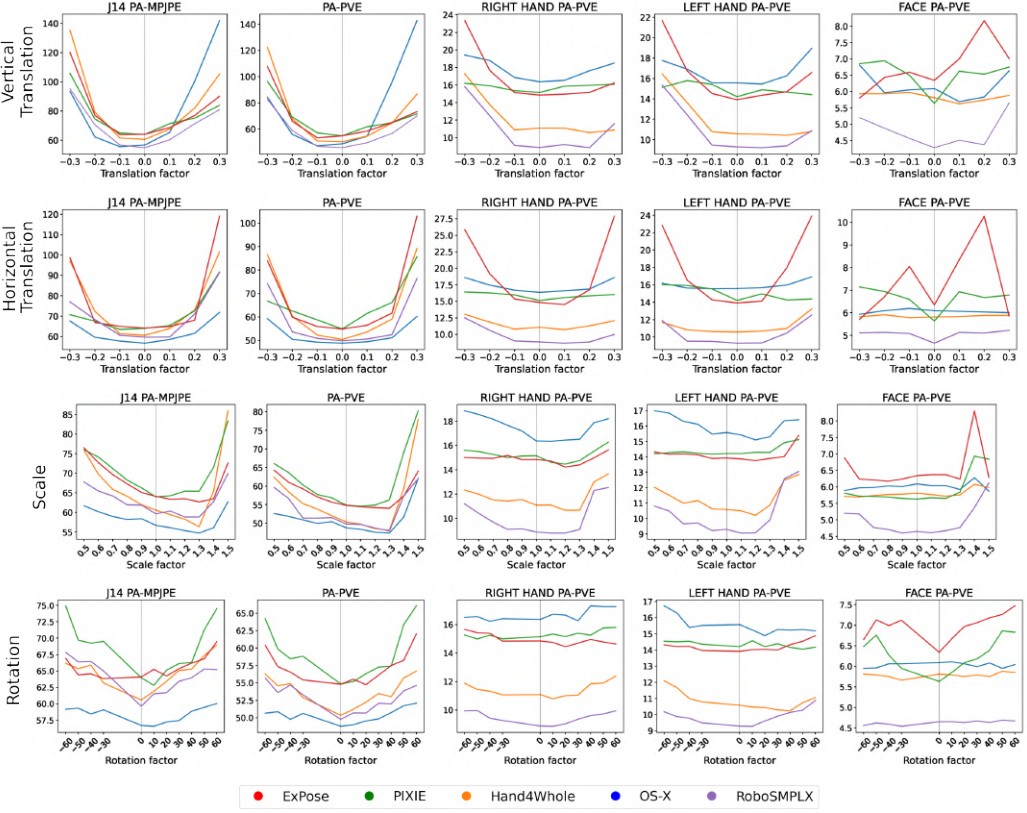

Figure 22: **Wholebody errors under different amounts of augmentation on EHF test set. The gray line indicates baseline performance without augmentation.**

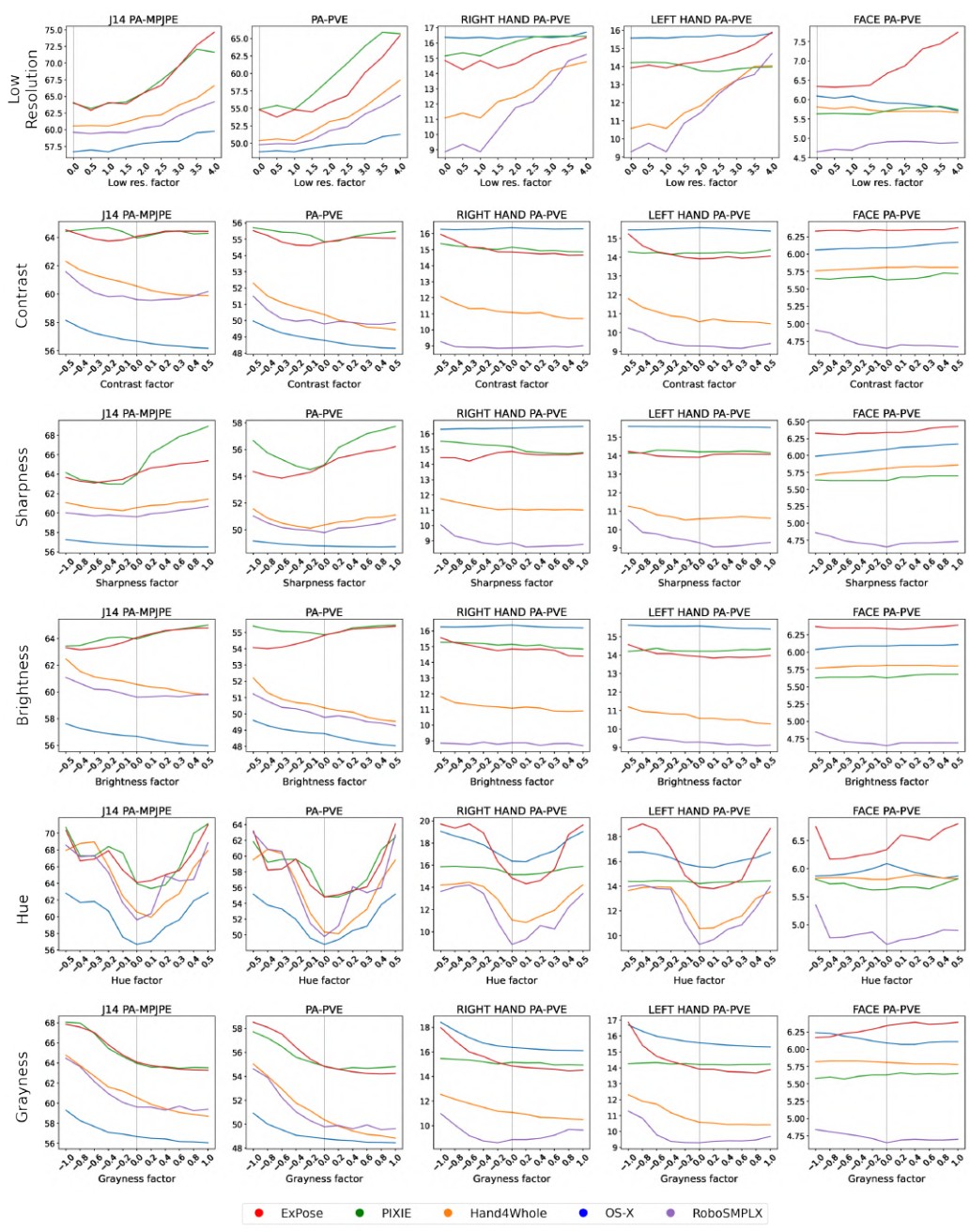

Figure 23: **Wholebody errors under different amounts of augmentation on EHF test set (cont.) The gray line indicates baseline performance without augmentation.**

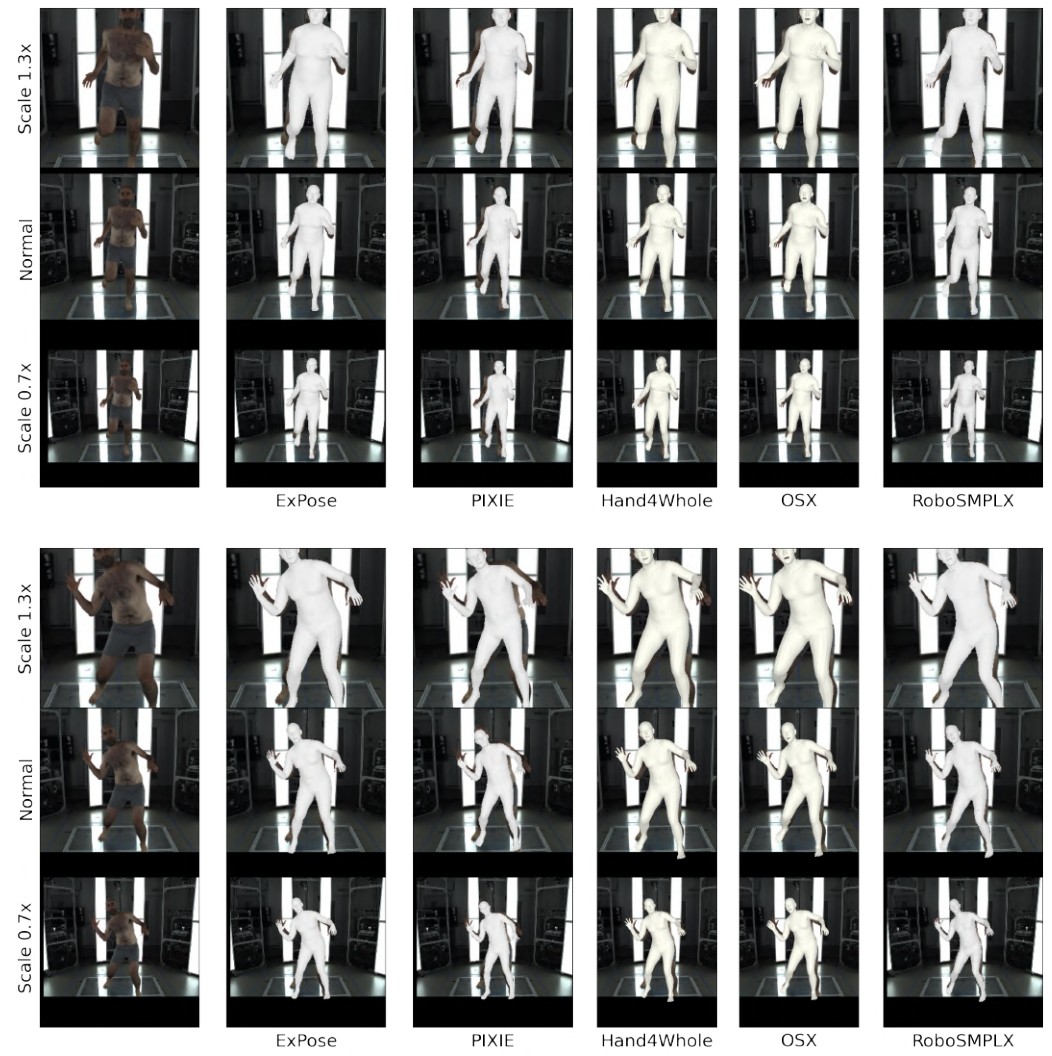

Figure 24: **Visualisation of Expose [41], PIXIE [11], Hand4Whole [35], OS-X [27] and** RoboSMPLX **under different scales on EHF test set.**

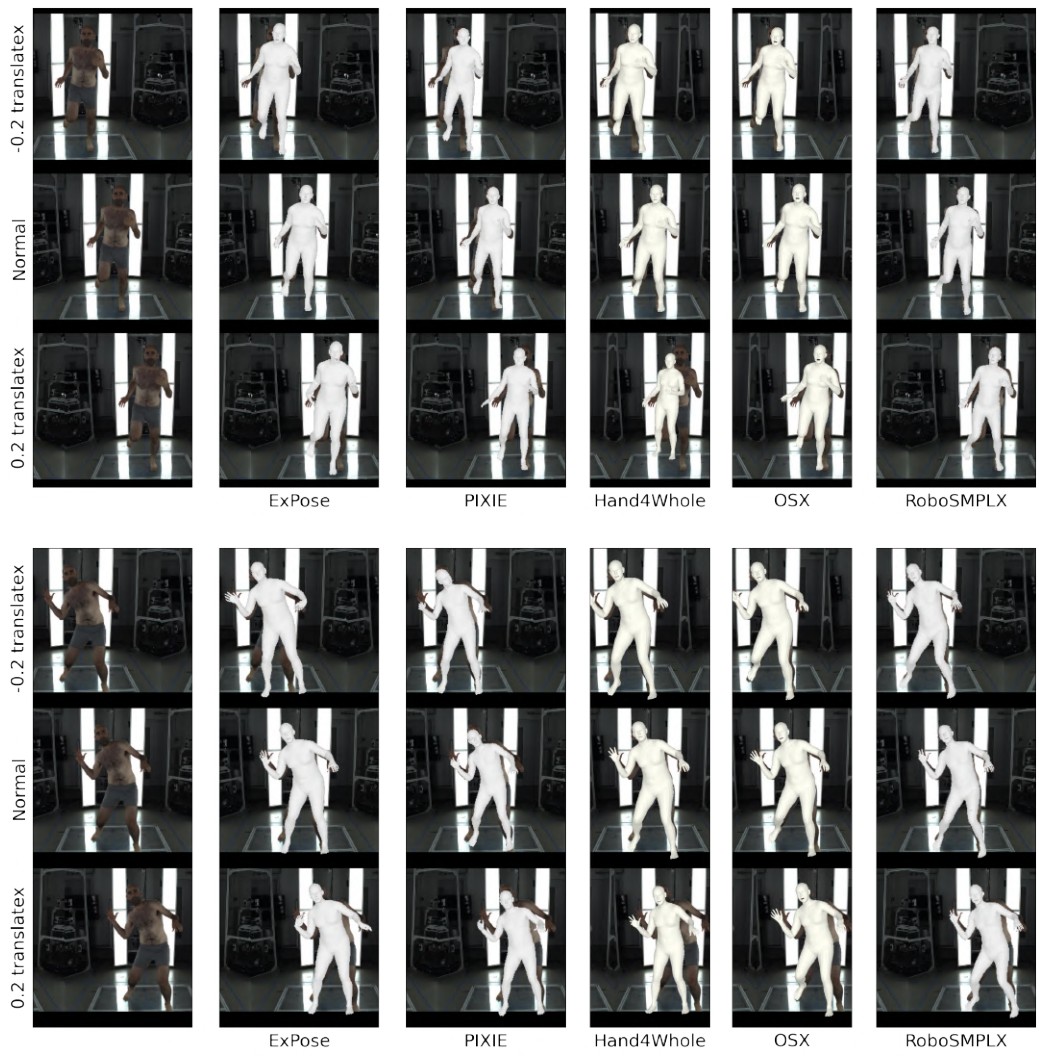

Figure 25: **Visualisation of Expose [41], PIXIE [11], Hand4Whole [35], OS-X [27] and** RoboSMPLX **under different levels of horizontal translation on EHF test set.**

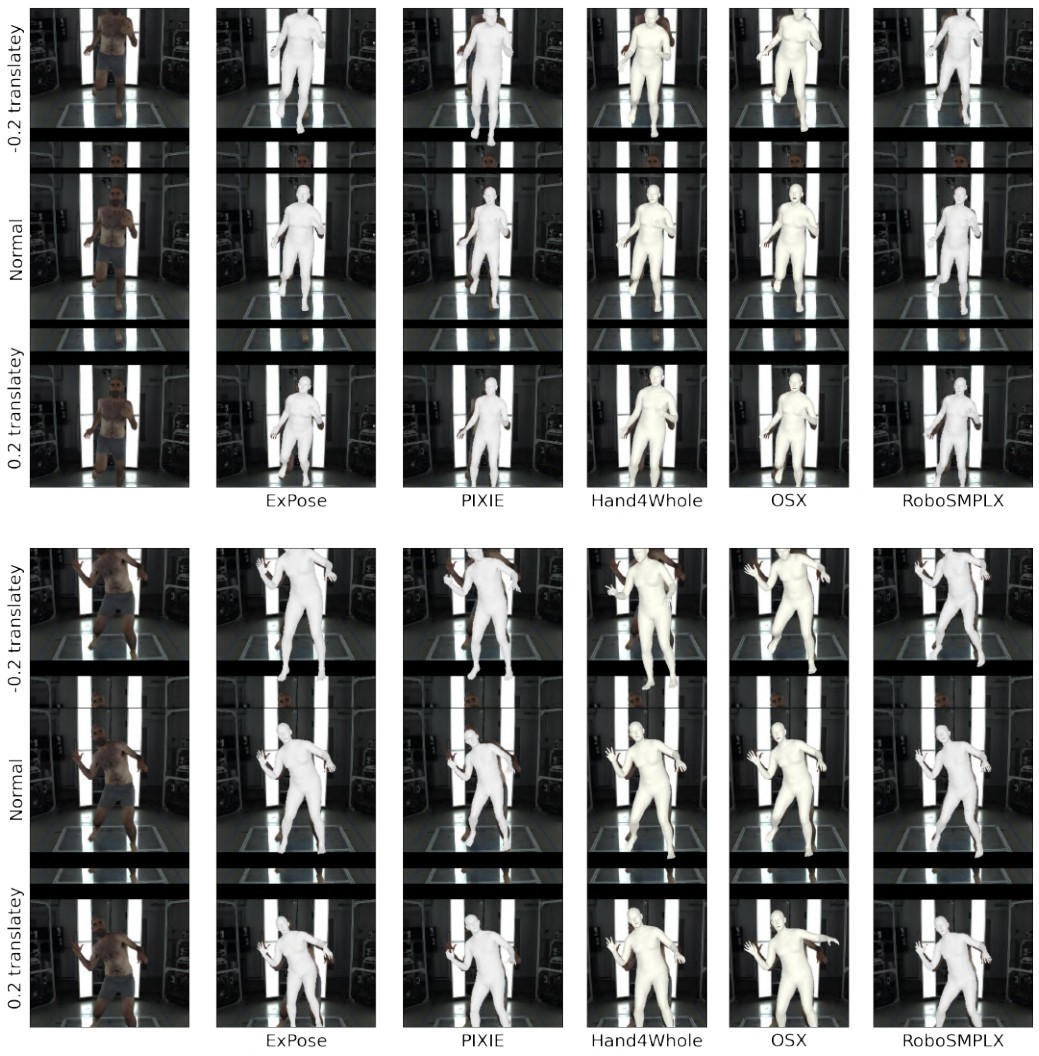

Figure 26: **Visualisation of Expose [41], PIXIE [11], Hand4Whole [35], OS-X [27] and** `RoboSMPLX` **under different levels of vertical translation on EHF test set.**

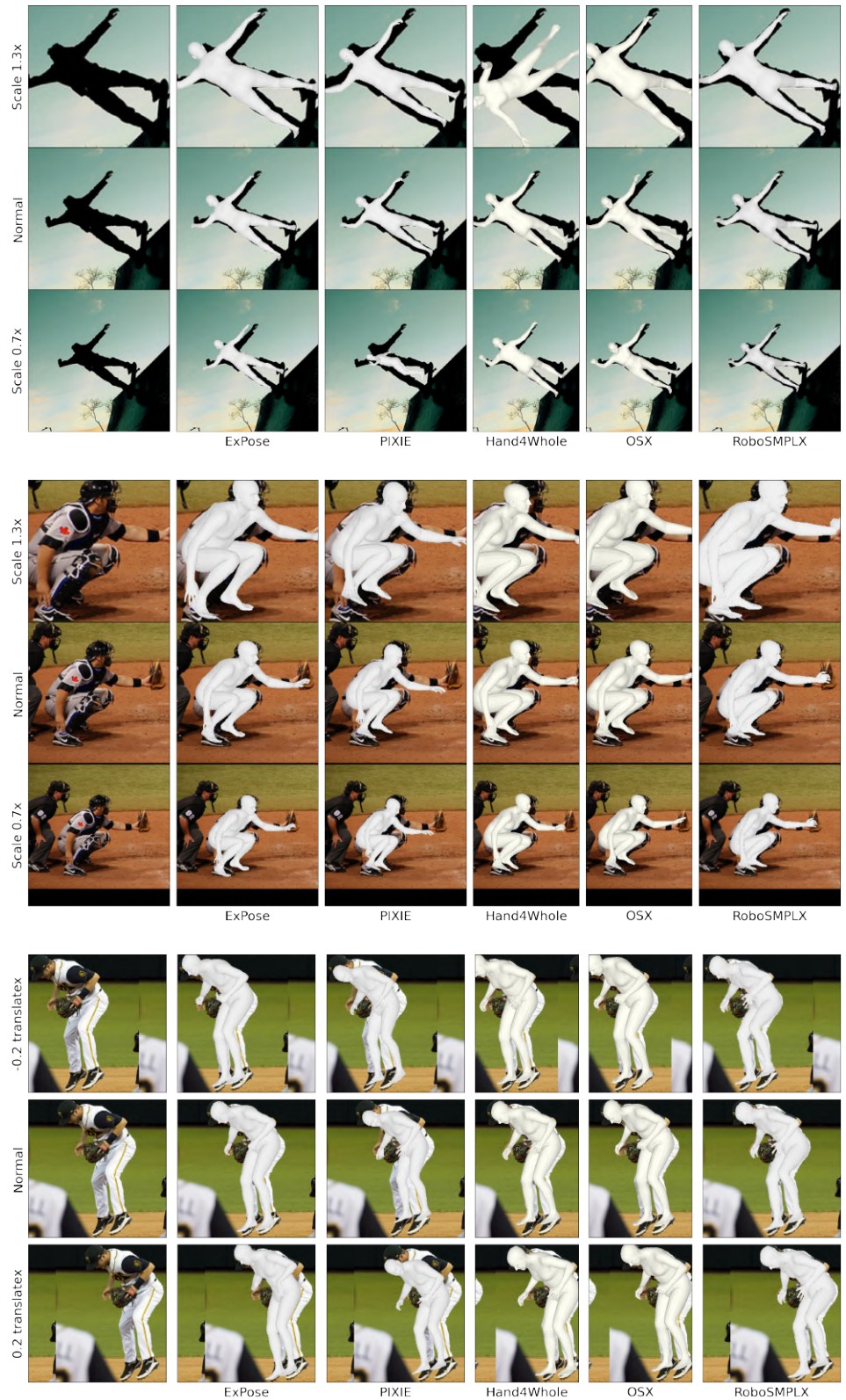

Figure 27: **Visualisation of Expose [41], PIXIE [11], Hand4Whole [35], OS-X [27] and** `RoboSMPLX` **under different scales and alignment on COCO validation set.**

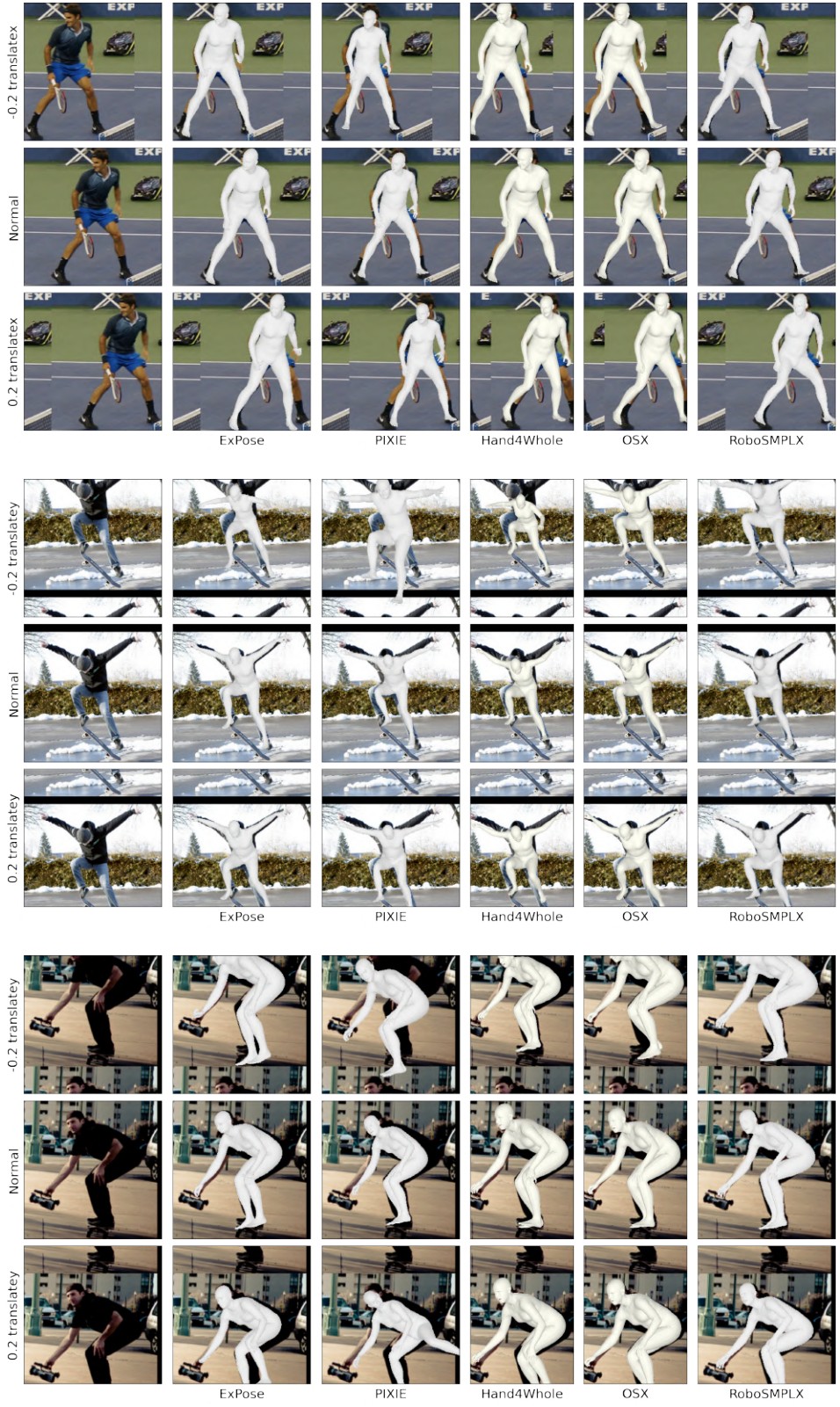

Figure 28: **Visualisation of Expose [41], PIXIE [11], Hand4Whole [35], OS-X [27] and RoboSMPLX under different scales and alignment on COCO validation set.**

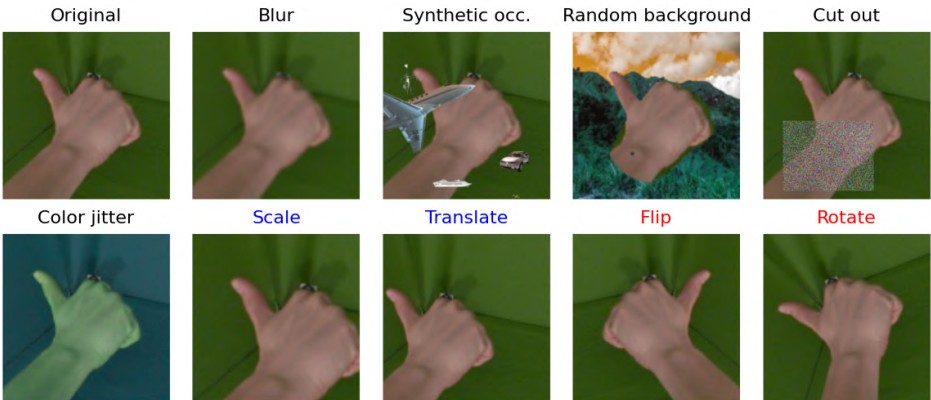

Figure 29: **Augmentations for Hand sub-networks. Blue and red labels represent location-variant and pose-variant augmentations respectively.**