# OpenReview forum: "Towards Robust and Expressive Whole-body Human Pose and Shape Estimation"
_NeurIPS.cc/2023/Conference — NeurIPS 2023 poster_

### Official Review · Reviewer_Uf3X · 2023-06-25

**Soundness:** 3 good
**Presentation:** 4 excellent
**Contribution:** 3 good
**Rating:** 6
**Confidence:** 4

**Summary:**

The paper focuses on improving whole-body pose and shape estimation from monocular images, a task that often struggles with complex, real-world scenarios. The authors argue that the performance of these models is significantly impacted by the quality of the predicted bounding box, such as the scale and alignment of body parts. The difference between ideal bounding box annotations and model detection results poses a substantial challenge to whole-body pose and shape estimation. To address this, the authors introduce RoboSMPLX, a framework to enhance the robustness of whole-body pose and shape estimation. RoboSMPLX integrates three new modules: a Localization Module to improve model awareness of the subject's location and semantics within the image space, a Contrastive Feature Extraction Module that uses contrastive loss with dedicated positive samples to help the model be invariant to robust augmentations, and a Pixel Alignment Module to ensure the reprojected mesh from predicted camera and body model parameters are accurate and pixel-aligned. The effectiveness of the method is showcased through comprehensive experiments on body, hands, face, and whole-body benchmarks.

**Strengths:**

The paper is well-structured and reader-friendly.

It begins with a compelling motivation that efficiently highlights the issue of the current state-of-the-art methods' low robustness against varying crops and misalignments. The authors also convincingly justify potential solutions to these problems.

The proposed method by the authors is sound even if it lacks complete novelty.

Impressively, the authors have performed extensive experiments and provided an in-depth ablation study, substantiating their proposed approach effectively.

Although their method doesn't top all the benchmarks, its efficacy is undeniable, particularly in its demonstrated robustness to variations in crop size and alignment - a challenge that other methods fail to meet. The authors provide a thorough justification with robust qualitative and quantitative results.

**Weaknesses:**

The claim made in lines 47-51 and 76-77, suggesting that contrastive learning is not used in the parametric estimation of human meshes, is inaccurate. Previous works, such as [1], have already incorporated a form of contrastive and triplet loss for 3D face shape and pose estimation from monocular face images. Thus, the application of contrastive loss for mesh recovery from monocular images isn't entirely novel. The authors should moderate their claims and include a thorough discussion concerning related work in this field.

Furthermore, the pixel alignment loss achieved by projecting the mesh as a mask, a technique used by ICON [2] for improved SMPL pose parameter estimation, is not entirely new.

The authors should also include instances of the method's failure during testing within the main body of the paper.

[1] Learning to Regress 3D Face Shape and Expression from an Image without 3D Supervision. CVPR 2019
[2] ICON: Implicit Clothed humans Obtained from Normals. CVPR  2022

**Questions:**

The paper seems sound to me and I do not have much questions.

**Limitations:**

Although some limitations have been discussed it might not be adequate. Please add some visual limitations of the model during the inference time.

---

> ### Author Rebuttal · Authors · 2023-08-09
>
> We sincerely thank the reviewer for acknowledging our extensive experiments and in-depth ablation study, and recognising the efficacy of our methods. We will polish the paper and add the clarifications below in the revised version.
>
> Below we would like to provide point-to-point responses to address all the raised questions:
>
> **Q1: "application of contrastive loss for mesh recovery from monocular images isn't entirely novel. The authors should moderate their claims and include a thorough discussion concerning related work in this field."**
>
> **A1:** Thank you for your suggestion. We have added [1] in the discussion for related works in this field.
>
> As stated in Sec.4.3, our novel design lies in:
> 1.  **The scheme of supervised contrastive learning for high-dimensional regression.** Prior SSL methods [43, 58, 4] are not particularly adept at learning useful embeddings for human pose and shape estimation. Without labels, the model primarily extracts features based on background information instead of pose information.
> 2. **The design of human pose representation.** Sanyal et al. [1] use shape as the representation to disentangle expression, head pose and camera parameters. To learn accurate poses, we find that using the normalized keypoints representation in 3D is more useful than using pose parameters (joint rotations).
> 3. **The sampling strategy for selecting positive samples.** Prior works [43, 58] employed pose-variant augmentations (e.g., rotation and flipping), which can adversely affect the learning by altering the global orientation.
>
> Below is the discussion and comparisons with existing methods, which will be added to the related works in the revision:
>
> “Contrastive Learning. Recently contrastive learning has demonstrated state-of-the-art performance among self-supervised learning (SSL) approaches. This strategy has been applied to 3D hand pose and shape estimation [43, 58]. **Sanyal et al [1] incorporates a novel shape consistency loss for 3D face shape and pose estimation that encourages the face shape parameters to be similar when the identity is the same and different for different people.** Choi et al. [4] were the first to apply contrastive learning for 3D human pose and shape estimation. They found that SSL is not useful for this task, as the learned representations could be challenging to embed with high-level human-related information. Khosla et al. [18] proposed supervised contrastive learning for image classification tasks, which incorporates label information during training. **Currently there is no attempt to apply this strategy to human pose and shape estimation, where the definition of positive samples is unclear, and data lie in a continuous space. We are the first to overcome these challenges and integrate supervised contrastive learning with whole-body pose and shape estimation.**”
>
> **Q2: "pixel alignment loss achieved by projecting the mesh as a mask, a technique used by ICON [2] for improved SMPL pose parameter estimation, is not entirely new."**
>
> **A2:** We acknowledge the methodological parallels between our work and that of ICON. However, it's crucial to delineate the distinctions in implementation and objectives.
>
> ICON [2] relies on an existing segmentation model to retrieve a silhouette mask of a clothed human. Pixel alignment loss is employed during inference time to refine the SMPL parameters by fitting the projected SMPL silhouette to the “ground-truth” clothed silhouette mask.
>
> Our method differs from ICON is several ways:
> - Our methodology adopts a regression approach to directly predict pixel-aligned mesh. Pixel alignment loss is exclusively employed during the training phase to supervise the ground-truth and predicted part segmentation map. This offers a notable speed-up over ICON's optimization procedure.
> - Additionally, our approach circumvents the need for an external pretrained network dedicated to clothed silhouette mask prediction. The fitting results of ICON [2] depend on the accuracy of the segmentation model, which remains limited in occlusion scenarios.
> - We have found that a differentiable part segmentation map holds a higher efficacy compared to normal silhouette supervision. This encourages learning of correct prediction of body part and silhouette, even in instances of object or self-occlusion.
>
>
> **Q3: "instances of the method's failure during testing"**
>
> **A3:** Thank you for the suggestion. Please refer to Figure 3 in Rebuttal document.
>
> [1] Learning to Regress 3D Face Shape and Expression from an Image without 3D Supervision. CVPR 2019
> [2] ICON: Implicit Clothed humans Obtained from Normals. CVPR 2022
>
> Please don’t hesitate to let us know if there are any additional clarifications or experiments that we can offer!

---

> > ### Author Response · Authors · 2023-08-16
> > **Follow-up**
> >
> > Dear reviewer,
> >
> > We would like to follow up to check if your concerns have been addressed. In the previous response, we have made the following updates/clarification:
> >
> > - Following your precious advice to include more literature on use of contrastive setting (Q1), we have added extra discussions to the Related Works sections and provided clarification on how our module differs from previous work.
> > - Regarding the difference in pixel alignment compared to ICON (Q2), we have outlined the key distinctions.
> > - We have also added visual examples of the method’s failure cases (Q3) in Figure 3 of `Rebuttal.pdf`.
> >
> > We are happy to answer further questions.

---

> > > ### Comment · Reviewer_Uf3X · 2023-08-19
> > >
> > > I have read the rebuttal from the authors and my queries were properly answered. Therefore I will keep my ratings positive for this paper.

---

> > > > ### Author Response · Authors · 2023-08-20
> > > >
> > > > Dear Reviewer,
> > > >
> > > > We sincerely appreciate your acknowledgement of our work, and we are grateful for the emphasis you placed on our method's motivation, execution, and supporting experiments.

---

### Official Review · Reviewer_4HYU · 2023-06-27

**Soundness:** 3 good
**Presentation:** 3 good
**Contribution:** 3 good
**Rating:** 7
**Confidence:** 5

**Summary:**

This paper addresses the task of whole-body pose and shape estimation, including human mesh, hand gestures, and facial expressions, from monocular images. The author identifies the impact of predicted bounding box quality on the accuracy and reliability of existing methods. Based on this observation, a novel framework called RoboSMPLX is proposed to enhance robustness through three modules: a localization module, a contrastive feature extraction module, and a pixel alignment module. Comprehensive experiments are conducted to demonstrate the effectiveness of the framework.

**Strengths:**

1. Overall, the paper is of high quality, with clear motivation and well-organized sections. It is easy to follow and presents novel modules that address existing limitations. The paper includes comprehensive experiments and provides sufficient visualizations of the generated mesh. The author's contributions are highly appreciated.

2. The empirical study on the impact of subject localization, feature extraction, and pixel alignment, as stated in lines 91-100, is crucial in breaking through barriers. The paper provides extensive qualitative and quantitative results, both in the main paper and supplementary material, making it solid and convincing.

3. The three modules work together effectively, aligning intermediate representations such as pose, dense landmarks, segmentation, and silhouette masks. This enhances robustness through proper data augmentation techniques.

4. The proposed framework consistently achieves impressive results for various tasks across multiple datasets, demonstrating superior performance in whole-body pose and shape estimation.

**Weaknesses:**

1. Since the author does not mention code release, it would be beneficial to release the code to contribute to the research community.

2. The paper lacks a discussion on computational complexity and inference speed, which are important evaluation metrics, especially for practical deployment. I suggest reporting total parameters, FLOPs, and fps with a detailed discussion to strengthen the paper.


**Questions:**

See weakness

**Limitations:**

1.  Although the proposed framework achieves impressive performance, the overall framework and training strategy is somewhat complex. Simplifying the framework without sacrificing performance would be beneficial.

2. Currently, the proposed framework only allows for whole-body estimation from a single image. Future work could focus on video-based estimation to further improve robustness, alleviate depth ambiguity, and enhance temporal smoothness.

---

> ### Author Rebuttal · Authors · 2023-08-09
>
> We sincerely thank the reviewer for acknowledging that we present novel modules, and recognising that our extensive qualitative and quantitative results are solid and convincing. We will polish the paper and add the clarifications below in the revised version.
>
>
> Below we would like to provide point-to-point responses to address all the raised questions:
>
> **Q1: "it would be beneficial to release the code to contribute to the research community."**
>
> **A1:** Thanks for your suggestion. We plan to release code here at https://github.com/robosmplx/RoboSMPLX/ when the paper is accepted.
>
> **Q2: "reporting total parameters, FLOPs, and fps with a detailed discussion to strengthen the paper"**
>
> **A2:** Please refer to our response to General Concerns 2.
>
> **Q3: "the overall framework and training strategy is somewhat complex. Simplifying the framework without sacrificing performance would be beneficial"**
>
> **A3:** Thanks for this great suggestion. Our current goal is to enhance the model robustness. How to simplify the framework without performance drop will be our future work.
>
> **Q4: " Future work could focus on video-based estimation to further improve robustness, alleviate depth ambiguity, and enhance temporal smoothness"**
>
> **A4:**  Thanks for this great suggestion. This is indeed an interesting and important direction, and we will explore this in future work.
>
>
> Please don’t hesitate to let us know if there are any additional clarifications or experiments that we can offer!

---

> > ### Author Response · Authors · 2023-08-16
> > **Follow-up**
> >
> > Dear reviewer,
> >
> > We would like to follow up to check if your concerns have been addressed. In the previous response, we have made the following updates/clarification:
> > - Following your advice on code release (Q1), we have created a GitHub for the project and plan to release the code there when the paper is accepted..
> > - Regarding the total parameters, FLOPs, and fps of the various methods, we have added it to General Concerns 2.
> > - For your valuable suggestions to improve upon this project (Q3 and Q4), we intend to explore it in the future.
> >
> > We are happy to answer further questions.

---

> > > ### Comment · Reviewer_4HYU · 2023-08-19
> > >
> > > I appreciate this nice rebuttal, I believe many concerns have been solved effectively. Please further refine the paper based on all valuable feedback from reviewers. I will keep my rating positive as I explained in the general response.

---

> > > > ### Author Response · Authors · 2023-08-20
> > > > **Follow-up 2**
> > > >
> > > > Dear Reviewer,
> > > >
> > > > We genuinely appreciate your acknowledgement and support of our work. We will continue to refine the paper based on the feedback provided by you and other reviewers.

---

### Official Review · Reviewer_bkwo · 2023-06-30

**Soundness:** 3 good
**Presentation:** 2 fair
**Contribution:** 2 fair
**Rating:** 5
**Confidence:** 5

**Summary:**

The paper introduces RoboSMPLX, a method for whole-body 3d human pose and shape estimation from a monocular image. Motivated by the poor robustness of existing methods, especially w.r.t. the quality of bounding boxes, three components are proposed: 1) a localization module, 2) a contrastive feature extraction module, and 3) a pixel alignment module. The robustness of existing methods is evaluated by applying different types of image, location and pose augmentations. By using these augmentations in a contrastive setting during training of RoboSMPLX, RoboSMPLX achieves higher robustness when evaluated on these augmentations. Additionally, the performance of Hand, Body, Face and Wholebody reconstructions are evaluated on typical benchmark datasets.

**Strengths:**

Increasing the robustness of whole-body human pose and shape estimation methods is an important topic. The problem is well motivated, and the evaluation of the performance of existing methods under different augmentations is interesting.

**Weaknesses:**

I have the following main concerns about the paper:

1) Evaluation seems inconsistent and incomplete.
It is unclear why some competitors are omitted in different experiments. For example, the performance of PyMAF[54] is reported in Table 1 and 2, but not in Table 3 and 6, even though [54] reports the relevant numbers. OSX[25] is also missing in Table 2. The competitors outperform RoboSMPLX in the experiments where they are omitted.

2) Robustness of RoboSMPLX.
The robustness evaluation seems not realistic. It is not surprising that RoboSMPLX shows better performance under the different augmentations the method is trained on. However, most relevant is the performance on in-the-wild scenarios. This is not properly evaluated for the whole-body task. On the contrary, RoboSMPLX performs worse on AGORA than OSX. Although AGORA is a synthetic dataset, it contains realistic and diverse scenes with multiple persons and occlusions. On the other hand, EHF only consists of 100 images of a single subject recorded in a mocap studio. It is therefore questionable, if RoboSMPLX really succeeds in being more robust than its competitors. To better assess the in-the-wild performance, the methods could for example be evaluated on RICH (CVPR'22) (https://rich.is.tue.mpg.de/) or on BEDLAM (CVPR'23) (https://bedlam.is.tue.mpg.de/). Additionally, to show the effectiveness of the localization module, the accuracy of the predicted bounding boxes could be evaluated.

Futher concerns:
- The contrastive module is not well motivated. Why use the contrastive setting with all its overhead, instead of simply using the different transformations as data augmentation? Especially since adding positive samples only has small influence.

- Why are the ablation studies in Table 11 not conducted with the whole body model? Mean+std of multiple runs should be reported since the differences are so small.

- Why is the pixel alignment not evaluated quantitatively? This would be more meaningful instead of only showing some examples e.g. in Figure 13.

- The experiments section should be revised. It is difficult to read due to the many experiments and referrals to the appendix.

- The novelty is limited since the augmentations and the pixel alignment module already have been used extensively in the literature for estimating human pose and shape, e.g. [35] and [45].

- IIn Table 10 7.18 PA-PVE should be in bold, and 14.38 MPJPE in Table 11.

**Questions:**

The following requires clarification:
- What is the estimation error reported in Table8?
- Table 11: what does DR54 and KS stand for? What is the row named "joints"?
- Regarding line 204: why should the model produce consistent representations for the same subject under different poses?
- For how long is the model trained on what hardware? What's the inference time?
- How are the joints for the body, hand and face determined? How does the 137 joints body skeleton look like?

**Limitations:**

More failure cases could be shown in the paper. Also in comparisons to the competitors.

The potential societal impact is not discussed.

---

> ### Author Rebuttal · Authors · 2023-08-10
>
> We sincerely thank the reviewer for your insightful and constructive feedback. We will polish the paper, add the experiments and make the clarifications in the revised version.
>
> **Q1: "Evaluation seems inconsistent and incomplete. It is unclear why some competitors are omitted in different experiments."**
>
> **A1:** **Our evaluation methodology has been systematically designed based on the underlying models of the compared methods.** Specifically, the body network was trained on datasets tailored for SMPL and hence was benchmarked against other methods utilizing the SMPL framework, including PyMAF [54]. Conversely, OS-X, which is rooted in the SMPL-X framework, was compared solely with its SMPL-X counterparts, such as Expose, Hand4Whole and PIXIE in Table 6 and not Table 2. For comparisons with PyMAF-X, please refer to  **Response 1 A2**
>
> **Q2: "Additionally, to show the effectiveness of the localization module, the accuracy of the predicted bounding boxes could be evaluated**
>
> **A2:** Thank you for the suggestion. We have evaluated the accuracy of predicted part bounding boxes on the EHF test set. We have employed the Intersection over Union (IoU) as our metric (Please refer to Figure and Table 1 in the Rebuttal.pdf). Our method obtains the highest IoU scores in Table 1.
>
> **Q3: "Why are the ablation studies in Table 11 not conducted with the whole body model?"**
>
> **A3:** **The primary objective of the ablation study was to demonstrate the efficacy of individual modules.** Given that wholebody network contains three subnetworks (body, hand, face), and each subnetwork encompasses all three modules, **it was deemed more methodologically sound to isolate and assess the impact of each module within a distinct subnetwork.** This approach ensures a clearer understanding of the contribution of each module without the potential confounding effects of evaluating them within the entirety of the whole body model.
>
> **Q4: The contrastive module is not well motivated. Why use the contrastive setting with all its overhead, instead of simply using the different transformations as data augmentation? "**
>
> **A4:** Our use of the contrastive module is motivated by the need to constrain/maintain the same pose feature for different augmentations, to avoid domain shift caused by strong augmentation alone. The experiments show that the use of strong augmentation alone for training can lead to performance deterioration, while combining it with the contrastive loss consistently results in minimal errors (Table 2 in Rebuttal.pdf)).
>
> The contrastive features extraction module is not used in the inference, and only used during training. Therefore, during inference time it will not pose any computation overhead. Contrastive learning is a successful strategy in other CV areas [43, 58], and we are the first to introduce it on pose for Human Pose and Shape Estimation.
>
> To illustrate this further, we delved into a visualization of the pose similarity for augmented samples. The findings reveal that augmented samples are perceived as dissimilar in both Model 0 and Model 1 (Table 3 in Rebuttal.pdf)). Yet, when examining Model 2, a marked increase in embedding similarity is evident, underscoring the advantage of the contrastive approach.
>
> **Q5: Why is the pixel alignment not evaluated quantitatively? "**
>
> **A5:** In assessing the pixel alignment, it's crucial to recognize that standard metrics like PVE and MPJPE errors are computed post root alignment. They also do not measure the accuracy of mesh projection within the image space. Presently, there's an absence of a metric tailored to gauge the degree of pixel alignment of a mesh in this context. As such, our study primarily offers a qualitative analysis.
>
> Nevertheless, in an effort to address your concern and introduce some quantitative results, we measure the errors between the projected 2D vertices of ground-truth and projected mesh (Please refer to Figure 2 and Table 3 in Rebuttal.pdf). From Table 3, it is evident that omitting the pixel alignment module leads to suboptimal outcomes. In contrast, our pixel alignment strategy, leveraging rendered segmentation maps, showcases better performance than using vertex loss as supervision.
>
> **Q6: The experiments section should be revised. "**
>
> **A6:** Thanks for the suggestion. If we have the opportunity for revision, we will include important experiments in the main text as much as possible, while maintaining the completeness of information without frequent reference to supplementary materials.
>
> **Q7: In Table 10 7.18 PA-PVE should be in bold, and 14.38 MPJPE in Table 11"**
>
> **A7:**  Thanks for pointing that out, we will fix the bolding in Table 11.
>
> **Q8: What is the estimation error reported in Table8?"**
>
> **A8:** The estimation error for Table 8 refers to the difference between the top-1 retrieved pose (COCO-train) and query pose (COCO-test). Thanks for pointing this out, we will edit the caption to make it clearer.
>
> **Q9: Regarding line 204: why should the model produce consistent representations for the same subject under different poses?**
>
> **A9:** L204:  “By minimizing this loss, the model can produce consistent representations for the same subject, even when presented with different augmentations”. To clarify, this means that the model should produce consistent representation for the same subject with the same pose under different augmentations.
>
> **Q10: For how long is the model trained on what hardware? What's the inference time?**
>
> **A10:** Our model was trained utilizing a cluster of 8xTesla V100-SXM2-32GB GPUs. Specific to the training duration, the hand models required approximately one day, whereas the body and face models necessitated two days. The joint training process was completed within a day. For details on inference time, please refer to the section labeled "General Concerns 2."
>
> Please don’t hesitate to let us know if there are any additional clarifications or experiments that we can offer!

---

> > ### Author Response · Authors · 2023-08-16
> >
> > **Q11: How are the joints for the body, hand and face determined? How does the 137 joints body skeleton look like?"**
> >
> > **A11:**
> > We follow the same setting used by H4W and OSX. The whole-body skeleton comprises 137 joints, broken down as follows: 25 body joints, 40 hand joints, and 72 face joints. The 25 body joints adhere to the SMPL body convention. The joints determined by the MANO model consist of 21 for each hand. However, we omitted the wrist joint as it is already accounted for in the body, resulting in a total of 40 hand joints (calculated as (21-1) x 2). The 72 face joints are derived from the FLAME convention, with the exclusion of the neck joint, which is encompassed within the body convention, adjusting the count from 73 to 72.
> >
> > **Q12: More failure cases could be shown in the paper. Also in comparisons to the competitors.**
> >
> > **A12:** Thank you for the suggestion. Please refer to Figure 3 in Rebuttal document.

---

> > > ### Author Response · Authors · 2023-08-16
> > > **Follow-up**
> > >
> > > Dear reviewer,
> > >
> > > We would like to follow up to check if your concerns have been addressed. In the previous response, we have made the following updates/clarification:
> > > - Regarding the evaluation inconsistency and completeness (Q1), we provided explanations of why certain methods are compared against others, given the underlying frameworks.
> > > Regarding your advice to show the effectiveness of the localization module (Q2), we have evaluated the accuracy of the predicted bounding boxes in `Rebuttal.pdf`.
> > > - Regarding the use of data augmentation instead of contrastive setting (Q4), we demonstrated that the Contrastive Module can help to avoid domain shift, results in improved performance and is only used during training.
> > > - Regarding your concern for the quantitative evaluation of pixel alignment (Q5), we have added projected vertex error as an additional metric.
> > > - Following your advice, we plan to refine the experiment section (Q6), correct table formatting errors (Q7), clarify the meaning of estimation error (Q8) in the main text.
> > > - We have also provided extra clarification regarding why the ablation study was not conducted with the wholebody (Q2), the consistency for representations across varied augmentations (Q9), and have detailed the training and inference specifics (Q10).
> > >
> > > We are happy to answer further questions.

---

> > > > ### Comment · Reviewer_bkwo · 2023-08-17
> > > >
> > > > Dear authors,
> > > >
> > > > thanks for the extensive rebuttal! I especially appreciate the additional experiments regarding the bounding box prediction accuracy and pixel alignment. My questions were also addressed satisfactorily.
> > > >
> > > > However, my main concern remains the wholebody performance on in-the-wild scenarios. Experiments on AGORA (Table 6) show that the proposed method is quite heavily outperformed by OS-X (and PyMAF-X). Additionally, with the release of more diverse and accurate SMPL-X datasets, e.g. RICH (CVPR'22) and BEDLAM (CVPR'23), SMPL-X prediction methods that focus on in-the-wild robustness should use such datasets for evaluation, instead of the extremely limited mocap dataset EHF.
> > > >
> > > > Furthermore, I disagree that OS-X cannot be compared with your method on Body-only accuracy (Table 2). In essence, the goal of both methods is to reconstruct whole-body meshes. Both methods utilize the SMPL-X body model and can be evaluated, among others, on body-only 3D reconstruction (as is done in both papers). It's rather a selling point that OS-X is a one-stage pipeline instead of a multi-stage pipeline with different expert models for face, hands and body, and that OS-X doesn't need or use hand/face/body-specific training data.
> > > >
> > > > Since I appreciate the rebuttal but some of my main concerns couldn't be resolved, I will only slightly raise my score.

---

> > > > > ### Author Response · Authors · 2023-08-20
> > > > > **Follow-up 2**
> > > > >
> > > > > Dear reviewer,
> > > > >
> > > > > We deeply appreciate your thoughtful feedback and the adjustment in your scoring. We are glad to know that our additional experiments and explanations addressed many of your questions.
> > > > >
> > > > > **Q1: Additionally, with the release of more diverse and accurate SMPL-X datasets, e.g. RICH (CVPR'22) and BEDLAM (CVPR'23), SMPL-X prediction methods that focus on in-the-wild robustness should use such datasets for evaluation, instead of the extremely limited mocap dataset EHF.**
> > > > >
> > > > > **A1:** We chose EHF as it served as the primary benchmark for SMPL-X methodologies, which has been commonly used in previous relevant works [5, 9, 32, 35]. However, we acknowledge your valid observation regarding the increased availability of diverse SMPL-X datasets, specifically RICH and BEDLAM. Thanks for your suggestion, we are working on the experiments. Due to the time constraint, we are not sure if we can get the evaluation results before the discussion period closes. If so, we will update you with the results.
> > > > >
> > > > > **Q2: Furthermore, I disagree that OS-X cannot be compared with your method on Body-only accuracy (Table 2). In essence, the goal of both methods is to reconstruct whole-body meshes. Both methods utilize the SMPL-X body model and can be evaluated, among others, on body-only 3D reconstruction (as is done in both papers). It's rather a selling point that OS-X is a one-stage pipeline instead of a multi-stage pipeline with different expert models for face, hands and body, and that OS-X doesn't need or use hand/face/body-specific training data.**
> > > > >
> > > > > Table 2. 3D reconstruction error trained without additional hand-only and face-only training datasets
> > > > > |                  |        |   MPVPE  |          |          | PA-MPVPE |          |         |
> > > > > |------------------|--------|:--------:|:--------:|:--------:|:--------:|:--------:|:-------:|
> > > > > |                  | Params | All      | Hands    | Face     | All      | Hands    | Face    |
> > > > > | H4W (1st stage)  | 77.84  | 79.2     | 43.2     | 25.0     | 53.1     | 12.1     | 5.8     |
> > > > > | OSX              | 422.52 | **70.8** | 53.7     | 26.4     | **48.7** | 15.9     | 6.0     |
> > > > > | Ours (1st stage) | 120.68 | 76.7    | **39.9** | **21.2** | 51.79    | **11.2** | **5.5** |
> > > > >
> > > > > **A2:** We recognize the merit of OS-X as a streamlined, one-stage pipeline, and acknowledge its superior performance in certain body metrics. However, the hands and face results are not as competitive. Table 2 displays the results of Hand4Whole, OSX and our approach when trained in a one-stage pipeline using the same SMPL-X datasets (COCO, MPII, Human36M from NeuralAnnot). Both our method and Hand4Whole outperforms OSX in terms of hand and face metrics. We would also like to point out that OS-X uses a much larger backbone (Vit-L), which might have some advantage in the performance.
> > > > >
> > > > > Moreover, the limitations of a one-stage pipeline like OS-X become evident when one considers the vast potential of specialized datasets. By relying solely on SMPL-X data, the approach misses the opportunity to leverage on widely available hand and face-specific datasets. These datasets encapsulate nuanced scenarios, such as interacting hands (InterHand2.6M), hands with objects (GRAB, 100DOH), emotional facial expressions (AffectNet), and racially diverse faces (BUPT). Such rich datasets can significantly enhance the robustness of hand and face subnetworks. In addition, our multi-stage pipeline allows for modularity. Each subnetwork can be independently upgraded, leveraging new, more specialized datasets as they become available.
> > > > >
> > > > > We are happy to answer further questions.

---

### Official Review · Reviewer_go2c · 2023-07-01

**Soundness:** 3 good
**Presentation:** 3 good
**Contribution:** 2 fair
**Rating:** 4
**Confidence:** 4

**Summary:**

This paper proposes a method to improve the robustness of whole-body pose and shape estimation, which mainly contains three components: 1) localization module to give the network awareness of location and semantic part; 2) contrastive feature extraction module to predict consistent representations under different augmentations; 3) pixel-alignment module to ensure alignment between projected mesh and 2d evidence.


**Strengths:**

1) This paper is well-written and easy to understand

2) The topic of robustness of pose and shape estimation is meaningful and  the proposed method is effective against augmentations

3) The experiments are comprehensive and visualizations are very nice to help understand the method


**Weaknesses:**

1) Since the main topic is the robustness of whole body pose and shape estimation, the literature review of robustness in vision tasks, especially in pose estimation tasks should be included. For example, [1][2].

[1] Bai, Yutong, et al. "CoKe: Contrastive Learning for Robust Keypoint Detection."
[2] Zhang, Yumeng, et al. "Improving robustness for pose estimation via stable heatmap regression."


2) I wonder why the target task is whole-body pose and shape estimation rather than body/hand pose estimation?
3) Why does ‘baseline’ only appear in table2 but not in table1 and 3? Also, the definition of baseline is not clear. The definition of baseline is not clear either in table 9
4) Where is the result of robustness of body subnetwork again augmentations?
5) Where is the details of table 7?
6) Eq,2 typo


**Questions:**

see weaknesses

**Limitations:**

see weaknesses

---

> ### Author Rebuttal · Authors · 2023-08-10
>
> **Q1: "literature review of robustness in vision tasks, especially in pose estimation tasks should be included"**
>
> **A1:** Thank you for your suggestion of a comprehensive literature review. We would like to emphasize that **prior works on 2D pose estimation are different from our 3D pose and shape estimation tasks. They focus on 2D/3D joint locations while we target the prediction of pose, shape and camera parameters for image subjects.** Below is our detailed discussion, which will be added to the revision.
>
> “Tackling robustness in vision tasks have motivated extensive works. Within the domain of human pose estimation, strategies including data augmentation, architectural innovations, and diverse training strategies have been actively explored. Specifically, (1) "AdvMix" [3] stands out by enhancing the robustness of pose estimation models using data augmentation. This method combines adversarial augmentation with knowledge distillation, where a generative framework mixes various corrupted images to confuse a pose estimator. Through such adversarial training, the estimator is conditioned to learn from harder samples, making it more robust. (2) Architectural modifications have been explored as a means to increase robustness. For instance, the work in [2] delineates a unique heatmap regression approach, encompassing three core components: a row-column correlation layer, a highly differentiated heatmap regression, and a maximum stability training protocol. This strategy is devised to buffer the network against minute perturbations. (3) Contrastive learning has also been applied to improve robustness.  Bai et al. [1] introduced “CoKe”, a contrastive learning framework tailored for keypoint detection. By detecting each keypoint independently, the method demonstrates robustness, especially against occlusions, compared to more conventional approaches.
>
> **Q2: "I wonder why the target task is whole-body pose and shape estimation rather than body/hand pose estimation?"**
>
> **A2:** 3D pose and shape estimation and 2D/3D pose estimation are fundamentally two different tasks. **The former involves determining the "pose" and "shape" parameters of a statistical human body model, whereas the latter concentrates on predicting the locations of 2D/3D keypoints within an image.** We follow existing works [16, 21, 19, 23] in the task definition of human pose and shape estimation. 3D whole-body pose and shape estimation allows us to retrieve a human mesh from the predicted pose and shape parameters, opening a myriad of applications across diverse domains such as computer graphics and augmented/virtual reality. It has been a popular topic and attracts lots of attention [16, 21, 3, 15, 23, 8, 20, 19, 9, 5, 56, 42, 54]
>
> **Q3: "Why does ‘baseline’ only appear in table2 but not in table1 and 3? Also, the definition of baseline is not clear. The definition of baseline is not clear either in table 9"**
>
> **A3:** Sorry for the confusion. Baseline refers to training with the same datasets and backbone on HMR, but with the addition of the extra modules (Localization, Contrastive FE and Pixel Alignment Modules). For Table 1, the Baseline is HMR which is included. For Table 3, we have updated the baseline below.
>
> Updated Table 3. Evaluation of the Face subnetwork.
> | Method      | LQ Mean(mm) &darr; | HQ Mean(mm) &darr; |
> | ----------- | ------------- | ------------- |
> | ExPose [5]  | 2.27          | 2.42          |
> | ExPose †    | 2.46          | 2.38          |
> | HMR         | 2.18          | 2.11          |
> | HMR†        | 2.31          | 2.27          |
> | RoboSMPLX   | 2.12          | 2.08          |
> | RoboSMPLX † | 2.12          | 2.1           |
>
> **Q4: Where is the result of robustness of body subnetwork again augmentations?"**
>
> **A4:**
> Thanks for the great suggestion. Please find the result in the table below. Similar to the findings in Table 11, we conclude that training with strong augmentations can cause domain shift. This was also found in prior works [15, 19, 35]. In Table 4 of PARE [19], and Table 4 of HMR-EFT [15], Table 8 of [35], they also showed that adding crop augmentation can harm performance on existing benchmarks.
>
> Ablation of different modules on Body subnetwork. Results are trained on EFT-COCO and tested on 3DPW test set.
> |                            | PA-MPJPE &darr; | MPJPE &darr;|
> | -------------------------- | -------- | ----- |
> | Baseline (HMR)             | 60.8     | 96.2  |
> | Baseline (HMR) + strongaug | 63.2     | 101.5 |
>
> **Q5: Where is the details of table 7?"**
>
> **A5:** Thanks for pointing this out. It was previously wrongly referenced to Table 18 in Line 288. We have fixed it, it is now referenced to Table 7 instead. We hope this helps to clarify.
>
> **Q6: Eq,2 typo"**
>
> **A6:** We would like to clarify what does “Eq,2 typo” refer to?
>
> Please don’t hesitate to let us know if there are any additional clarifications or experiments that we can offer!

---

> > ### Comment · Reviewer_go2c · 2023-08-15
> > **Clarification of Q2**
> >
> > Dear author,
> >
> > I want to have further clarification on my second question. My question is why the target task is whole-body pose and shape estimation rather than body/hand pose and shape estimation. Let's call this task mesh recovery. I understand that mesh recovery tasks are very different from 2D/3D keypoint detection. My question is usually whole-body is an advanced task compared with body or hand mesh recovery and there are many more existing works focused on body or hand mesh recovery than whole-body. Also, the challenges of whole-body mesh recovery are usually low resolution of hands or the rotation of the wrist. When coming to robustness, whole-body tasks seems the same as purely body or hand tasks for me. Therefore, it would be more meaningful to first study the robustness of body mesh recovery since there are many more competitive works to compare to show the superiority of your method. The proposed method which should be applicable to all mesh recovery tasks also seems less convincing in this way.

---

> > > ### Author Response · Authors · 2023-08-16
> > > **Response to Clarification of Q2**
> > >
> > > Dear reviewer,
> > >
> > > Thank you for raising valid concerns about our choice of focusing on whole-body mesh recovery. Below we provide explanations that address your queries.
> > >
> > > **Q1: why the target task is whole-body pose and shape estimation rather than body/hand pose and shape estimation**
> > >
> > > **A1:** Compared to body mesh recovery, **whole-body mesh recovery is a more prevalent and important task which offers broader applications**. It allows for the accurate modeling of hand gestures and facial expressions, making it particularly advantageous for intricate tasks such as clothed human reconstruction, editing, and animations that utilize SMPL-X predictions.
> > >
> > > **Q2:  the challenges of whole-body mesh recovery are usually low resolution of hands or the rotation of the wrist. When coming to robustness, whole-body tasks seems the same as purely body or hand tasks for me**
> > >
> > > **A2:** While we acknowledge that whole-body mesh recovery inherently encapsulates the challenges of individual body parts like hands, face, and the body itself, **whole-body mesh recovery presents unique robustness problems, different from body/hand mesh recovery.** Beyond the common challenges like the “low resolution of hands [25] or rotation of wrist [32]”, **the accuracy of face and hand part crops influencing their respective subnetworks is a significant factor.** Two main concerns are the inaccurate localization of part crops by the wholebody network and the robustness of hand and face subnetworks when handling inaccurate part crops. Our approach addresses both these challenges.
> > >
> > > It's worth noting that in dedicated hand or face-mesh recovery, images primarily showcase the hand or face at the center. In contrast,  this is often not the case in whole body estimation (part crops of existing methods are illustrated in Figure 2). The robustness issue is more pronounced and relevant in whole-body mesh recovery. This motivates our study to improve the robustness in every subnetwork of the wholebody pipeline.
> > >
> > > **Q3: it would be more meaningful to first study the robustness of body mesh recovery since there are many more competitive works to compare to show the superiority of your method**
> > >
> > > **A3: **
> > >
> > > **We demonstrate the efficacy of our proposed modules, namely Localization, Contrastive Feature Extraction, and Pixel Alignment on body mesh recovery** (Table 2, Figure 15). In addition, the modules are also effective for  face (Table 3, Table 5, Figure 15)  and hand mesh recovery (Table 1, Table 4, Figure 14). Their performance outpaces existing methods (as seen in Tables 1, 2, 3), and they exhibit robustness under various positional augmentations (evidenced by Table 4 and 5). Qualitative demonstrations of this can be found in Figures 14 and 15.
> > >
> > > **Existing solutions on robustness for body/hand mesh recovery do not work as well.** Previous efforts for robustness in body/hand mesh recovery often relied heavily on data augmentation, leading to domain shifts as seen in [15, 19, 35]. Notably, some found that excessive crop augmentation can degrade benchmark performance (e.g., Tables 4 of PARE [19] and HMR-EFT [15], and Table 8 of [35]). Contrarily, our method leverages supervised contrastive learning, proven to effectively mitigate errors while maintaining robustness.
> > >
> > > We hope this helps to clarify why we choose whole body mesh recovery instead of body/hand mesh recovery. We are happy to answer further questions.

---

> > > > ### Comment · Reviewer_go2c · 2023-08-16
> > > >
> > > > Dear authors,
> > > >
> > > > Thank you for your explanation. Tables are evaluating the performance of body, hand, and face separately against sota whole-body methods. However, the performance on body subnetwork is not very competitive against state-of-the-art body networks, e.g. HybrIK, CLIFF. This makes me doubt whether the gain mainly comes from the more powerful network or robustness strategies. To evaluate the robustness of body-only methods, one can still input inaccurate bounding boxes.

---

> > > > > ### Author Response · Authors · 2023-08-20
> > > > > **Follow-up 2**
> > > > >
> > > > > Dear reviewer,
> > > > >
> > > > > We will polish the paper and add the clarifications below in the revised version.
> > > > >
> > > > > **Q1: However, the performance on body subnetwork is not very competitive against state-of-the-art body networks, e.g. HybrIK, CLIFF.**
> > > > >
> > > > > Table 1. Evaluation of HybrIK, CLIFF and our network on 3DPW. Our results are also available in Table 2 in main paper.
> > > > > | Method | Backbone  | F-T on 3DPW       | PA-MPJPE (3DPW) | MPJPE (3DPW) |
> > > > > | ------ | --------- | ----------------- | --------------- | ------------ |
> > > > > | HybrIK | HRNet-W48 | No                | 48.6            | 88.0         |
> > > > > | HybrIK | HRNet-W48 | Yes               | 41.8            | 71.3         |
> > > > > | CLIFF  | Res-50    | Trained with 3DPW | 45.7            | 72.0         |
> > > > > | CLIFF  | HRNet-W48 | Trained with 3DPW | 43.0            | 69.0         |
> > > > > | Ours   | Resnet-50 | No                | 49.8            | 80.8         |
> > > > > | Ours   | HRNet-W48 | No                | 48.5            | 80.1         |
> > > > >
> > > > > **A1:** There are many factors affecting training,including, but not limited to, the choice of backbone, datasets employed, and specific protocols executed during evaluation. Specifically, with regards to 3DPW, various protocols—ranging from fine-tuning (3DPW Protocol 1), collective training, to omission during training (3DPW Protocol 2)—have a large influence on 3DPW results in the evaluation process.
> > > > >
> > > > > We outperform HybrIK when using the same backbone (HRNet-W48) and not fine-tuning on 3DPW (3DPW Protocol 2). Notably, CLIFF incorporated 3DPW within its training datasets. Given that our approach and that of both HybrIK and CLIFF do not utilize identical dataset combinations, a direct comparison becomes inherently challenging.
> > > > >
> > > > >
> > > > > **Q2: This makes me doubt whether the gain mainly comes from the more powerful network or robustness strategies.**
> > > > >
> > > > > **A2:** For thorough ablation, we presented baselines in Table 2 in the main paper. The baseline is training on the same dataset combination, backbone, and employing consistent training strategies. It's evident that the introduction of our modules improves performance while keeping all other factors (backbone, dataset) the same. Additionally, we provide ablation studies on the hand subnetwork for different modules in Table 11 to showcase their effectiveness when the same backbone and dataset choice is maintained.
> > > > >
> > > > >
> > > > > **Q3: To evaluate the robustness of body-only methods, one can still input inaccurate bounding boxes.**
> > > > >
> > > > > **A3:** We have provided qualitative comparisons of body-only methods under different scale and alignment in Figure 15.
> > > > >
> > > > > Below, we provide quantitative evaluations of our method with HMR, SPIN and PARE. Our method is able to achieve better performance under different scales and alignment. In the future, we will make our robustness evaluation pipeline publicly available for other methods to evaluate on. We believe this can serve as a valuable benchmark for future studies in this domain.
> > > > >
> > > > > Table 2. Evaluated on 3DPW (PA-MPJPE/MPJPE) under different scales and alignment. * denote the same dataset combination
> > > > >
> > > > > |                 | Normal       | Transx +0.2x | Transx -0.2x  | Transy +0.2y  | Transy -0.2y | Scale 1.3x    | Scale 0.7x    |
> > > > > | --------------- | ------------ | ------------ | ------------- | ------------- | ------------ | ------------- | ------------- |
> > > > > | HMR             | 67.53/112.34 | 77.31/141.70 | 77.06/ 138.51 | 86.57/ 151.15 | 77.26/148.33 | 68.46/ 117.1  | 75.38/ 124.79 |
> > > > > | SPIN            | 57.54/94.11  | 70.14/122.56 | 68.67/ 120.04 | 73.08/ 111.33 | 70.64/133.2  | 61.08/ 103.60 | 61.63/ 99.6   |
> > > > > | PARE  (HR32) \* | **49.3/81.8**    | 74.9/139.2   | 77.1/ 141.7   | 59.1/92.3     | 64.2/ 109.7  | 54.7/86.9     | **50.5/ 83.9**    |
> > > > > | Ours (R50) \*   | 49.8/80.8    | **67.2/117.2**   | **67.7/111.5**   | **56.4/90.0**     | **62.8/105.6**   | **50.2/84.6**     | 50.8/ 82.4    |
> > > > >
> > > > >
> > > > > We are happy to answer further questions.

---

### Official Review · Reviewer_p9Xq · 2023-07-05

**Soundness:** 2 fair
**Presentation:** 3 good
**Contribution:** 2 fair
**Rating:** 3
**Confidence:** 5

**Summary:**

This submission proposes RoboSMPLX for whole-body pose and shape estimation. RoboSMPLX incorporates three modules, including a localization module, a contrastive feature extraction module, and a pixel alignment module. The localization module is aware of the location and semantics of body parts so that cropping could be more accurate. The contrastive feature extraction module incorporates a pose- and shape-aware contrastive loss, along with positive samples, for better feature extraction under robust augmentations. The pixel alignment  module applies differentiable rendering to inherence of the re-projection alignment of the mesh.

**Strengths:**

The motivation of the proposed method is clear: using a localization module to improve the robustness of the cropping. Overall, the proposed method technically makes sense and can be easily reproduced. The experiments also show comparable or better performances with previous methods.

**Weaknesses:**

The most severe weakness is the novelty of the proposed method and the lack of comparisons with recent state-of-the-art solutions.

To be more specific, there are several major issues:

- The proposed localization and pixel alignment modules have very limited contributions to the community, as these operations are commonly used in this field. For instance, similar localization strategies are used in [39,56], and differentiable modules are used in [i].
[i] SK Dwivedi, N Athanasiou, M Kocabas, MJ Black, Learning to regress bodies from images using differentiable semantic rendering, ICCV 2021.

- There is a lack of discussion and comparison with the recent state-of-the-art method PyMAF-X [54]. An in-depth discussion and comparison with [54] is necessary to support the claim of the proposed method. It is recommended to include [54] in the Related Work section, compare results with [54] in Tables 3,4,5, and Table 7, and show qualitative results of the proposed method and [54] for compressive comparisons.

- This paper claims robust performances of whole-body pose and shape estimation, but no video result is provided in the supplementary materials. To convincingly demonstrate the robustness of the proposed method, it is also recommended to include a side-by-side comparison video [54].

**Questions:**

How about the run time of the proposed method?

**Limitations:**

The limitations of this paper are mainly the novelty and the lack of comprehensive comparisons with recent state-of-the-art solutions. Given such clear defects in the experimental results, I rate this paper as the one below the acceptance bar.

---

> ### Author Rebuttal · Authors · 2023-08-10
>
> We sincerely thank the reviewer for acknowledging the proposed method is clear. We will polish the paper, add the experiments and clarify below points in the revised version.
>
> **Q1: "The proposed localization and pixel alignment modules have very limited contributions to the community, as these operations are commonly used in this field."**
>
> **A1:** Our method is significantly different from the mentioned works [39, 56, i]. Specifically, Expose [39] directly regresses pose and shape parameters from the image **without taking into account any location information.**
>
> For [56], its IK modules (BodyIKNet and HandIKNet) focus **solely on keypoint data** for deriving pose parameters, and the FaceNet operates on a direct regression from the image. In contrast, our Localization module effectively captures **both sparse (through a 2.5 heatmap for keypoint location) and dense (through part segmentation) predictions for each human body part (body, face and hand).** The encoded location information is then used for prediction of pose, shape and camera parameters. The ablation study in Table 7 shows the superiority of our design over [56].
>
> Our method is technically different from [i] in several primary aspects:
> 1. [i] relies on an existing clothing segmentation model to retrieve a clothing segmentation mask, while we capitalize on the projected ground-truth mesh to source the part-segmentation mask, which is more cost-efficient as many SMPL-X datasets contain ground-truth wholebody parameters.
> 2. We have found that a differentiable part segmentation map holds a higher efficacy compared to normal silhouette supervision. This encourages learning of correct prediction of body part and silhouette,  even in instances of object or self-occlusion.
> 3. The module we introduce aims to bridge the disparity in pixel alignment observed in numerous SMPL-X models. Moreover, the integration of this module predominantly facilitates the accurate determination of camera parameters and enhances pixel alignment. Even marginal variations in camera parameters can induce significant alignment shifts.
>
>
> **Q2a: "There is a lack of discussion and comparison with the recent state-of-the-art method PyMAF-X [54]"**
>
> **A2a: 1)** Thanks for the suggestion. Below we provide detailed discussions and comparisons with PyMAF-X. We will also include [54] in the Related Work section.
> 1. **Acquisition of part bounding boxes**: PyMAF-X relies on an off-the-shelf whole-body pose estimation model (OpenPifpaf) to obtain whole body 2D keypoints of the person in the image, from which part crops are derived. During the EHF evaluation, PyMAF-X employs ground-truth hand and face bounding boxes. In contrast, our method and other works (ExPose [5], PIXIE [9], Hand4Whole [32], OS-X [25]) encompass a self-integrated module designed to extract hand and face bounding boxes directly from the image. So it is unfair to directly compare these works with PyMAF-X.
> 2. **Operational efficiency**: Openpifpaf imposes extra computation during inference, making PyMAF-X less efficient than our method. Please refer to Table 1 in General Comments 2.
> 3. **Network architecture**: Due to the diverse backbone and dataset combinations utilized, it is challenging for us to make whole-body network comparisons. In our paper (Table 1), we focus on contrasting RoboSMPLX’s Hand subnetwork with PyMAF’s Hand subnetwork. Both networks are trained and evaluated on the same backbone and dataset, FreiHAND. In this context, our method surpasses PyMAF.
> 4. **Performance**: On the EHF metrics, our performance lags behind PyMAF-X. This could potentially arise from variations in the training datasets employed. While the training pipeline of the body network for PyMAF-X has been disclosed, the training specifics for hands and face and the methodology to integrate hand, face, and body module PyMAF-X, remains undisclosed. We intend to replicate with similar training datasets in the future.
>
> **Q2b: "recommended to compare results with [54] in Tables 3,4,5, and Table 7, and show qualitative results of the proposed method and [54] for compressive comparisons.""**
>
> For comparisons on Tables 3, 4, 5, PyMAF-X did not provide the [pre-trained hand and face models](https://cloud.tsinghua.edu.cn/d/3bc20811a93b488b99a9/?p=%2Fdata%2Fpretrained_model&mode=list) to be evaluated on the respective FreiHand and Stirling benchmarks. In addition, the face model of PyMAF-X was trained on [VGGFace2](https://www.robots.ox.ac.uk/~vgg/data/vgg_face2/) which is no longer publicly available. Therefore, we were unable to reproduce a result with similar training configurations.
>
> For Table 7 when evaluating the whole body, the hand and face evaluations are affected by the accuracy of the part bounding box detected. In PyMAF-X’s evaluation on EHF dataset, the ground-truth part- bounding boxes are fed in for evaluation. Therefore this poses an unfair comparison. If we were to obtain the part-bounding box from OpenPifPaf, we will be indirectly evaluating OpenPifPaf, rather than PyMAF-X as they do not have any module for locating the part-bounding boxes.
>
> **Q3: "include a side-by-side comparison video "**
>
> We plan to include additional visualizations here https://github.com/robosmplx/RoboSMPLX/.
>
> **Q4: "run time of the proposed method"**
>
> Please refer to our response to General Concerns 2.
>
> Please don’t hesitate to let us know if there are any additional clarifications or experiments that we can offer!
>
> [1] SK Dwivedi, N Athanasiou, M Kocabas, MJ Black, Learning to regress bodies from images using differentiable semantic rendering, ICCV 2021.

---

> > ### Author Response · Authors · 2023-08-16
> > **Follow-up**
> >
> > Dear reviewer,
> >
> > We would like to follow up to check if your concerns have been addressed. In the previous response, we have made the following updates/clarification:
> >
> > - Regarding your concern on the novelty of the proposed localization and pixel alignment modules (Q1), we clarified the uniqueness of our modules in contrast to other works.
> > - Regarding your precious advice to compare to PyMAF-X (Q2), we discussed and compared our approach with PyMAF-X in detail and plan to add it to the "Related Work" section.
> > - Regarding your advice to include a side-by-side comparison video (Q3), we've uploaded comparison videos per your advice (Q3). Notably, PyMAF-X's performance is influenced by OpenPifpaf's predictions, evident in `pymafx_openpifpaf.mp4`. We observed improvements with a more robust pose estimator in `pymafx_mmpose.mp4`. Our model's results are in `robosmplx.mp4`.
> > - Regarding the run time of the proposed methods (Q4), we have addressed it in General Concerns 2.
> >
> > We are happy to answer further questions.

---

> > > ### Comment · Reviewer_p9Xq · 2023-08-18
> > > **Follow-up Comments**
> > >
> > > Dear authors,
> > >
> > > Thanks for the responses. I have read the rebuttal and comments from other reviewers. The additional comparisons indeed strengthen the experiments. However, my major concern about the novelty remains. The authors clarified the differences between the proposed method and previous works. But from my point of view, the novelty of these modifications is limited, making this paper hard to meet the bar of a NeurIPS paper.
> > >
> > > From the additional results provided by the authors, it seems that the hand and body box detection has a great impact on the final results. The robustness of previous whole-body solutions can be also improved with a better hand/body detector. As for the proposed method, the major improvements come from the body subnetwork, which is trained additionally and provides better hand/face detection for the re-cropping of hand/face. As pointed out in the first-round review, similar localization modules have been explored in previous work. These  localization modules are indeed improved by this work but the contribution is not significant.
> > >
> > > Given the additional results provided in the rebuttal, I would only slightly raise my score but keep it negative.

---

> > > > ### Author Response · Authors · 2023-08-20
> > > > **Follow-up 2**
> > > >
> > > > Dear reviewer,
> > > >
> > > > We appreciate the adjustment in your scoring. Regarding your concerns about the limited novelty of our approach, we provide additional clarifications below.
> > > >
> > > > While the body subnetwork indeed provides better hand/face detection, it isn’t the primary advancement. Our hand and face subnetworks are more robust even when the inputs are imprecise. This is demonstrated in Tables 4 and 5, and in Figures 14 and 15, where in instances of inaccurate crops, our method consistently outperforms existing techniques.
> > > >
> > > > Furthermore, It's crucial to highlight that the localization module is not the sole contribution for the improved robustness of each network. We have two other modules - contrastive feature extraction module and pixel alignment module, both of which significantly contribute to the overall robustness of the network. We hope that you can also consider the merits of the other two modules when evaluating our approach.
> > > >
> > > > We are happy to answer further questions.

---

> > > > > ### Comment · Reviewer_p9Xq · 2023-08-20
> > > > > **Follow-up Comments**
> > > > >
> > > > > Dear authors,
> > > > >
> > > > > Thank you for providing more clarifications. Despite the novelty, the evaluation of the proposed method remains insufficient from my point of view. Actually, I have pointed out the insufficient evaluation of Tables 3,4,5, and Table 7 in the first-round review. These tables are not updated in the rebuttal due to the lack of the pre-trained hand and face models of previous methods or the unfair settings. I do not agree with that as the subnetwork is typically contained in the whole-body network for whole-body methods. One can still evaluate the subnetworks and report results by feeding face/hand images to the subnetworks (please correct me if it is impossible). As for the unfair settings of whole-body evaluation, I would recommend reporting comprehensive results to Tables as possible as we can and explaining the performance gap and in-depth reasons in the paper. In this way, it could be clearer for readers to have more comprehensive comparisons regarding the performances of the proposed and recent solutions. Insufficient evaluations in benchmark Tables would indicate that the results are selective and not comprehensive/state-of-the-art in my opinion, especially when there is a lack of recent state-of-the-art methods.

---

> > > > > > ### Author Response · Authors · 2023-08-20
> > > > > > **Follow-up 3**
> > > > > >
> > > > > > Dear Reviewer,
> > > > > >
> > > > > > We appreciate the valuable feedback. Regarding your concerns, we provide additional clarifications below.
> > > > > >
> > > > > > **Q1: I have pointed out the insufficient evaluation of Tables 3,4,5, and Table 7 in the first-round review. These tables are not updated in the rebuttal due to the lack of the pre-trained hand and face models of previous methods or the unfair settings.”**
> > > > > >
> > > > > > **A1:** Regarding Tables 3, 4, 5, and 7 Evaluation:
> > > > > >
> > > > > > 1. **Table 3:** As previously highlighted, the VGG Face2 training dataset for face is no longer available to the public. While we can incorporate PyMAF’s face results into Table 3, we have to make a caveat that these results cannot be replicated.
> > > > > >
> > > > > > 2. **Tables 4 & 5**: These tables specifically evaluate the hand and face subnetwork under different positional augmentation on hand and face specific datasets. As we rely on pre-trained subnetworks, we will update the results once the hand and face subnetworks of PyMAF-X, or their respective training codes become publicly available.
> > > > > >
> > > > > > 3. **Table 7**: As mentioned in Follow-up 1, PyMAF-X was excluded from the whole body evaluation under different scale and alignment in Table 7 **as they used ground-truth hand and face bounding boxes from the EHF test set**. Beyond the fairness concern, we believe it is not meaningful to include them as the errors will be invariant across positional alignments.
> > > > > >
> > > > > > **Q2: I do not agree with that as the subnetwork is typically contained in the whole-body network for whole-body methods. One can still evaluate the subnetworks and report results by feeding face/hand images to the subnetworks (please correct me if it is impossible).**
> > > > > >
> > > > > > **A2:** Indeed, extracting the hand and face subnetworks directly for comparison would be ideal. However, there are some practical challenges:
> > > > > >
> > > > > > 1. In methods including ExPose, PIXIE, Hand4Whole, OS-X, and our proposed model, **crucial layers for shape and global orientation are discarded in the hand and face subnetworks during the whole-body training**. This absence means that without all required parameters—pose, shape, global orientation, and facial expression—we cannot generate a hand or face mesh using MANO or FLAME layers.
> > > > > >
> > > > > > 2. Feeding hand or face images directly into the whole-body pipeline and then trying to crop the hand and face mesh from the SMPL-X mesh is also ambiguous. The output mesh relies on parameters such as body shape, wrist rotation, and neck rotation, all of which are predicted by the main body subnetwork, and dependent on the input body image. It is not so straightforward to disentangle the hand and face subnetworks from the body subnetwork. For example, in PyMAF-X, the wrist's rotation (global orientation of the hand subnetwork) is tied to the body's overall pose, as shown [here](https://github.com/HongwenZhang/PyMAF-X/blob/95e60745c68445cac4ba7c6a1d02071493297ddd/models/pymaf_net.py#L453-L454).
> > > > > >
> > > > > > **Q3: As for the unfair settings of whole-body evaluation, I would recommend reporting comprehensive results to Tables as possible as we can and explaining the performance gap and in-depth reasons in the paper. In this way, it could be clearer for readers to have more comprehensive comparisons regarding the performances of the proposed and recent solutions. Insufficient evaluations in benchmark Tables would indicate that the results are selective and not comprehensive/state-of-the-art in my opinion, especially when there is a lack of recent state-of-the-art methods**
> > > > > >
> > > > > > **A3:** We understand and are open to the inclusion of PyMAF-X in Tables 3 and 6, accompanied by an in-depth comparative analysis. For Tables 4 and 5, updates will be made once PyMAF-X's respective subnetworks or training codes are publicly available.
> > > > > >
> > > > > > We are happy to answer further questions.

---

> > > > > > > ### Comment · Reviewer_p9Xq · 2023-08-21
> > > > > > >
> > > > > > > Dear authors,
> > > > > > >
> > > > > > > Thank you for more clarifications. After reading the follow-up responses, I acknowledge the challenges of reporting the results of other methods in Tables 4 and 5. However, my primary concerns regarding the limited contributions and insufficient evaluation remain. I would like to summarize my thoughts as follows:
> > > > > > >
> > > > > > > - The contributions of two (of three) modules proposed in this paper are limited. I have pointed out the incremental modification of the localization module in previous comments. As for the Pixel Alignment Module, the differentiable rendering used in the proposed method is very similar to [DSR](https://dsr.is.tue.mpg.de). The authors highlight the usage of the part segmentation obtained from the projected ground-truth mesh. However, such operations can be dated back to [SURREAL](https://github.com/gulvarol/surreal) and are common practices in the task of 3D HPS Estimation. The code of [SPIN](https://github.com/nkolot/SPIN/blob/master/eval.py#L183) also contains similar operations .
> > > > > > >
> > > > > > > - As pointed out in my previous comments, the benchmark tables reported in the main paper are misleading due to the lack of comprehensive comparisons. For instance, the results shown in Table 6 indicate the performance of the proposed method is top-two among existing methods. But it is actually not true when considering the results of the recent method PyMAF-X. As the EHF and AGORA datasets are the only two benchmark datasets used in whole-body methods. A comprehensive comparison with recent methods is crucial as potential readers only read the paper but not the rebuttal. The authors blame this issue on unfairness. However, most of the benchmark comparisons in recent papers are unfair due to the usage of different training data and backbones. Including recent state-of-the-art methods is required for more in-depth discussions of the performance gaps and reasons.
> > > > > > >
> > > > > > > - Some claims/presentations/statements in the paper and rebuttal are overclaimed, misleading, and even wrong.
> > > > > > >
> > > > > > >     - Line 52: "This module applies differentiable rendering to ensure precise pixel alignment", I do not agree that the differentiable rendering used in this module can ensure alignment as it is not an optimization-based method. The qualitative results shown in the paper are also far from precise alignment.
> > > > > > >
> > > > > > >     - Line 88: "dense predictions and differentiable rendering have not been employed in whole-body pose and shape estimation", dense predictions are also used in PyMAF-X.
> > > > > > >
> > > > > > >     - As I am familiar with the EHF dataset, I know that this dataset does not provide ground-truth hand and face bounding boxes. When downloading the EHF dataset, we can obtain the images, SMPL-X models, and 2D whole-body joints. As explained in the file EHF_README.txt, the 2D joints are estimated with OpenPose from monocular RGB images, which are not considered as ground-truth annotations. Hence, saying PyMAF-X used ground-truth hand and face bounding boxes from the EHF test set is wrong. Actually, the comparisons in Table 7 basically show the performances of different solutions under different detection results. It is confusing to me why the authors refuse to include PyMAF-X in Table 7 as it can be viewed as a two-stage solution, which relies on off-the-shelf face/hand detection modules. The two-stage solutions are common in practices as the detection modules have been investigated extensively for decades. It remains unclear to readers regarding the performance gaps between the proposed method and the combination of strong detection modules and whole-body pose and shape estimators.
> > > > > > >
> > > > > > > Overall, after in-depth reading of the original submissions and the rebuttal, I hold the opinion that the current paper is not suitable to be accepted as a NeurIPS paper. I have rollbacked my rating to 3: Reject.

---

> > > > > > > > ### Author Response · Authors · 2023-08-21
> > > > > > > > **Follow-up 4 (1/2)**
> > > > > > > >
> > > > > > > > Dear Reviewer,
> > > > > > > >
> > > > > > > > Regarding your concerns, we provide additional clarifications below.
> > > > > > > >
> > > > > > > > **Q1: The contributions of two (of three) modules proposed in this paper are limited. I have pointed out the incremental modification of the localization module in previous comments. As for the Pixel Alignment Module, the differentiable rendering used in the proposed method is very similar to DSR. The authors highlight the usage of the part segmentation obtained from the projected ground-truth mesh. However, such operations can be dated back to SURREAL and are common practices in the task of 3D HPS Estimation. The code of SPIN also contains similar operations.**
> > > > > > > >
> > > > > > > > **A1:** In Follow-up 1, we have delineated how our method differs from DSR. For SURREAL, they tackle the part segmentation task. This is fundamentally different from the use of differentiable rendering for part segmentation masks to facilitate mesh recovery. As for SPIN, it utilizes an optimization technique to iteratively refine its predictions. Our method is a direct regression approach.
> > > > > > > >
> > > > > > > > **Q2a: As pointed out in my previous comments, the benchmark tables reported in the main paper are misleading due to the lack of comprehensive comparisons. For instance, the results shown in Table 6 indicate the performance of the proposed method is top-two among existing methods. But it is actually not true when considering the results of the recent method PyMAF-X. As the EHF and AGORA datasets are the only two benchmark datasets used in whole-body methods. A comprehensive comparison with recent methods is crucial as potential readers only read the paper but not the rebuttal. The authors blame this issue on unfairness. However, most of the benchmark comparisons in recent papers are unfair due to the usage of different training data and backbones. Including recent state-of-the-art methods is required for more in-depth discussions of the performance gaps and reasons.**
> > > > > > > >
> > > > > > > > **A2a:** Our intent was never to strategically exclude any method, let alone because of its performance. When raised in the rebuttal, our initial reasoning for not comparing with PyMAF-X is due to its distinct paradigm. While our approach and many existing methods operate on a one-stage mechanism, PyMAF-X employs a two-stage method where the hand and face bounding box localizations are dependent on a separate pose estimation network. By evaluating PyMAF-X, we would also be indirectly evaluating the off-the-shelf pose estimation network. Our apprehension, as voiced earlier, was rooted in the perceived 'unfairness' of comparing the detection ability inherent in one-stage methods to that of the borrowed pose estimation network.
> > > > > > > >
> > > > > > > > Upon your strong recommendation, we recognize the value of including PyMAF-X in our comparisons. As mentioned in Follow-up 3, and emphasized here again, **we are entirely open to integrating PyMAF-X results in Table 3, 4, 5, 6, 7 along with a thorough comparative analysis.**
> > > > > > > >
> > > > > > > > **Q2b: A comprehensive comparison with recent methods is crucial as potential readers only read the paper but not the rebuttal.**
> > > > > > > >
> > > > > > > > **A2b:** We would like to highlight that, during this current discussion period, **we are not allowed to make edits to the main paper**. However, if given an opportunity in future revisions or updates, we will include the necessary comparisons for a complete representation.
> > > > > > > >
> > > > > > > > **Q3: Line 52: "This module applies differentiable rendering to ensure precise pixel alignment", I do not agree that the differentiable rendering used in this module can ensure alignment as it is not an optimization-based method. The qualitative results shown in the paper are also far from precise alignment.**
> > > > > > > >
> > > > > > > > **A3:** We understand that the phrasing "ensure precise pixel alignment" might be seen as an absolute claim. Our intention was to highlight the improved pixel alignment provided by the differentiable rendering, not to suggest it guarantees perfect alignment. Based on your feedback, we will adopt a more nuanced wording that it “improves pixel alignment” or “ensures a more precise pixel alignment”.
> > > > > > > >
> > > > > > > > **Q4: Line 88: "dense predictions and differentiable rendering have not been employed in whole-body pose and shape estimation", dense predictions are also used in PyMAF-X.**
> > > > > > > >
> > > > > > > > **A4:** If the term "dense predictions" is a point of contention, we will rephrase Line 88 to: "The integration of differentiable rendering with dense part segmentation has not been previously employed in whole-body pose and shape estimation."

---

> > > > > > > > > ### Author Response · Authors · 2023-08-21
> > > > > > > > > **Follow-up 4 (2/2)**
> > > > > > > > >
> > > > > > > > > **Q5: As I am familiar with the EHF dataset, I know that this dataset does not provide ground-truth hand and face bounding boxes. When downloading the EHF dataset, we can obtain the images, SMPL-X models, and 2D whole-body joints. As explained in the file EHF_README.txt, the 2D joints are estimated with OpenPose from monocular RGB images, which are not considered as ground-truth annotations. Hence, saying PyMAF-X used ground-truth hand and face bounding boxes from the EHF test set is wrong.**
> > > > > > > > >
> > > > > > > > > **A5:** The OpenPose keypoints are provided alongside the EHF test set upon its release, and therefore we consider it as ground-truth. We'll ensure to make this distinction clear that PyMAF-X employs part-bounding boxes derived from ground-truth keypoints provided by the EHF test set, and not ground-truth part bounding boxes provided by the EHF test set.
> > > > > > > > >
> > > > > > > > > **Q6: Actually, the comparisons in Table 7 basically show the performances of different solutions under different detection results. It is confusing to me why the authors refuse to include PyMAF-X in Table 7 as it can be viewed as a two-stage solution, which relies on off-the-shelf face/hand detection modules. The two-stage solutions are common in practices as the detection modules have been investigated extensively for decades. It remains unclear to readers regarding the performance gaps between the proposed method and the combination of strong detection modules and whole-body pose and shape estimators.**
> > > > > > > > >
> > > > > > > > > **A6:** The primary purpose of Table 7 was to evaluate the robustness of different whole-body methods under varying scale and alignment. Our method and other works (ExPose [5], PIXIE [9], Hand4Whole [32], OS-X [25]) encompass a self-integrated module designed to extract hand and face bounding boxes directly from the image. Nevertheless, we are open to accommodating it. By adding PyMAF-X, we could highlight the performance gaps between existing one-stage methods and two-stage methods that combine whole-body pose and shape estimators and strong whole-body keypoint estimators.
> > > > > > > > >
> > > > > > > > > **In summary**, we have no intentions to exclude the results of any method, including PyMAF-X, whether its performance was superior or otherwise. At the time of our paper's submission, we were not apprised of PyMAF-X's recent acceptance into a peer-reviewed journal, TPAMI 2023. As mentioned in Follow-up 3, we recognize the value of including PyMAF-X in our comparisons. We are fully prepared to incorporate PyMAF-X outcomes in Tables 3 through 7, complemented by a detailed comparative review. It's also important to note that, during this ongoing discussion phase, we are not permitted to modify the main manuscript.
> > > > > > > > >
> > > > > > > > > Do let us know if you have further concerns.

---

> > > > > > > > > > ### Comment · Reviewer_p9Xq · 2023-08-22
> > > > > > > > > > **Follow-up Comments**
> > > > > > > > > >
> > > > > > > > > > Dear authors,
> > > > > > > > > >
> > > > > > > > > > I acknowledge the efforts the authors devoted to the rebuttal. As the authors are open to discussions, I would like to post additional comments regarding the follow-up responses.
> > > > > > > > > >
> > > > > > > > > > - "our approach and many existing methods operate on a one-stage mechanism". I disagree with this statement. i) Most existing methods are two-stage solutions, including PIXIE, H4W, and PyMAF-X. All these methods have separate detection networks and SMPL-X regression networks. PIXIE and H4W integrate the detection networks in their code, while PyMAF-X resorts to the usage of another detection module. ii) The detection networks may not be considered as a part of the whole-body methods as they are not claimed as contributions or are not even mentioned in previous papers.  iii) Whole-body methods focus on the final performances of the whole-body pose and shape estimation, and most of the previous methods can be integrated with a stronger off-the-shelf whole-body detector. Excluding the solutions using stronger whole-body detectors from comparisons makes the experiments less convincing. iv) Besides, when evaluating a method on AGORA, most recent methods rely on strong off-the-shelf detectors to crop person images. I do not see there is any reason to exclude the solutions using off-the-shelf detectors.
> > > > > > > > > >
> > > > > > > > > > - "The OpenPose keypoints are provided alongside the EHF test set upon its release, and therefore we consider it as ground-truth." I disagree that the joints estimated by OpenPose can be considered as the ground truth. The EHF dataset has never claimed the 2D joints as the ground truth. Actually, assuming that the EHF did not provide 2D joints, one can use OpenPose to generate 2D joints and crop body/hand/face images accordingly. The bounding boxes used in PyMAF-X can be viewed as the outputs of a strong detector based on OpenPose but are not related to ground-truth joints or bounding boxes.
> > > > > > > > > >
> > > > > > > > > > - From my point of view, it is necessary to include the comparisons of the proposed method and the combined solutions of strong whole-body detectors and whole-body pose and pose estimators. As the authors are willing to revise the main paper, I would like to see the comparisons of the proposed method and PyMAF-X in Tables 3, 6, and 7. The revision of Tables 4 and 5 is optional as I acknowledge the challenges of reporting the results of other methods in these two tables. Please post the updated Tables (3, 6, and 7) in the following days, and I will adjust the rating accordingly.

---

### Author Rebuttal · Authors · 2023-08-10

We sincerely thank all the reviewers for your constructive feedback and recognitions of this work, especially for acknowledging that the problem is well motivated and meaningful [p9Xq, bkwo, go2c, 4HYU, Uf3X], the experiments are comprehensive [go2c, 4HYU, Uf3X], and the proposed modules are effective [Uf3X, 4HYU, go2c].

We will follow your suggestions to polish the paper, add the experiments and make the clarifications in the revised version.

**General concerns:**

**1. Clarification about the novelty of the proposed method**

Below, we summarize the novelty of our approach and differences from previous works:

Our research aims to enhance the robustness of whole-body pose and shape estimation. Notably, many current methods face challenges in maintaining performance under the augmentations commonly observed in complex in-the-wild scenarios. We posit that the accuracy and reliability of such models are influenced by the quality of the predicted bounding box, especially concerning the scale and alignment of individual body components. In addressing these issues, we introduce three novel modules and empirically validate their efficacy across body, hand, face and whole body models.
1. **Localization Module**: This module incorporates both sparse and dense prediction branches, ensuring the model is aware of the location and semantics of the subject’s parts in the image. The learned location features that encode information about the sparse and dense representations are helpful in recovering relative rotations, shape and camera parameters. Our work is distinct from previous SMPL-X methods [5, 9] that do not use any location information. We also demonstrate that our method is more effective than previous methods that only use joint features [32, 25] or keypoints [56] for recovering pose, but neglect the effectiveness to recover shape and camera parameters. In the whole-body estimation pipeline, these location information play a pivotal role in localizing bounding boxes for hand and face subnetworks, which previous methods have a separate bounding box predictor for.
2. **Contrastive Module**: Merely leveraging strong data augmentation can introduce domain shifts. This was also found in prior works [15, 19, 35]. In Table 4 of PARE [19], and Table 4 of HMR-EFT [15], Table 8 of [35], it shows that adding crop augmentation can harm performance on existing benchmarks.  Our approach integrates supervised contrastive learning, utilizing the regressed keypoints from the mesh as the representation. This encourages the network to learn to produce similar embeddings for samples with the same pose under different augmentations.
3. **Pixel Alignment Module**: Minor deviations in scale or positional translations often result in visible misalignments in the projected mesh, indicative of errors in camera parameter estimations. While prior work relied on the supervision of projected keypoint for learning camera parameters, we introduce dense supervision of projected mesh using part-segmentation maps through differentiable rendering. This helps to ensure accurate pixel alignment of outputs. Notably, the combination of part-segmentation supervision and differentiable rendering has not been applied in whole-body pose and shape estimation.

**2. Run-time/ inference speed**

We measure the model size, computation complexity and inference time for different models including ours, as shown in the table below. Although our framework has sophisticated design, it has comparable inference speed as others, validating its efficacy.

Table 1: These results are tested on RTX3090. FLOP refers to the total number of floating point operations required for a single forward pass. The higher the FLOPs, the slower the model and hence low throughput. Inference Time is obtained by averaging across 100 runs.
|                       | Total parameters (M) | GFLOPs         | Inference time (s) |
| --------------------- | -------------------- | -------------- | ------------------ |
| ExPose                | 26.06                | 21.04          | 0.1330 &pm; 0.0050      |
| PIXIE                 | 109.67               | 24.23          | 0.1670 &pm; 0.0065      |
| Hand4Whole            | 77.84                | 17.98          | 0.0709 &pm; 0.0022      |
| OSX                   | 422.52               | 83.77          | 0.1998 &pm; 0.0028      |
| PyMAF-X (gt H/F bbox) | 205.93               | 33.41          | 0.2194 &pm; 0.0027      |
| PyMAF-X + OpenPifpaf   | 205.93 + 115.0       | 33.41 + 120.52 | 0.2727 &pm; 0.0136      |
| RoboSMPLX             | 120.68               | 29.66          | 0.2008 &pm; 0.0220      |

---

> ### Comment · Area_Chair_uZ43 · 2023-08-18
> **Reviewers: please read authors' responses and share your thoughts and additional questions**
>
> Dear reviewers,
>
> This paper got quite mixed ratings, and we should have more in-depth discussions to get an agreement.
>
> Reviewer 4HYU and Reviewer Uf3X: you originally gave positive ratings and we are curious about your current thoughts after you read other reviewers' reviews and authors' responses.
>
> Reviewer p9Xq: We are also curious about your opinion.

---

> > ### Comment · Reviewer_4HYU · 2023-08-18
> >
> > Dear AC,
> >
> > I appreciate the authors' effort in this whole-body human pose and shape estimation task, which is more challenging than any single human/face/hand task. I stand by my previous comments:
> > 1. The paper demonstrates high quality, featuring clear motivation and well-organized sections.
> > 2. The empirical study on the impact of three modules is crucial in breaking through barriers.
> >
> > The essence of this paper's contribution, as reiterated by the authors in their general response, is closely tied to the task of whole-body human pose and shape estimation. Assessing the contributions separately might undervalue their impact.
> >
> > While certain reviewers have raised concerns about insufficient comparisons, I agree with the authors that variations in settings across methods (e.g. training data combinations, strategies, and bounding box acquisition…) make fair comparisons challenging. I'm pleased to see substantial experiments in the rebuttal that showcase the proposed method's advantages over some mentioned methods.
> >
> > Indeed, the paper does have some issues as identified by reviewers. However, considering the paper's quality and contribution, my vote remains positive. I believe the authors can further enhance this paper based on the valuable feedback.

---

> > ### Comment · Reviewer_Uf3X · 2023-08-19
> >
> > I concur with reviewer 4HYU. Having reviewed the feedback from other reviewers and the authors' responses, I find no compelling reason to reject the manuscript. My assessment hinges on three pivotal points: a) the method's clear motivation, b) the sound execution of the method, and c) comprehensive experiments supporting the proposal. Although there were initial reservations regarding the novelty of the implementation, as shared by several reviewers, including myself, these concerns were adequately addressed in the authors' rebuttal. Even though parts of it might appear incremental, the effectiveness of the method, especially considering the intricate nature of the task, is evident. Moreover, it doesn't perturb me that the presented method hasn't topped every benchmark in the pose estimation arena. Such occurrences can be attributed to varying training conditions, like data sets, methodologies, training duration, backbone, and so on. A prevailing issue in the human pose estimation domain is that many studies solely present a singular accuracy figure in the MPJPE table, possibly from their best-performing network iteration. Ideally, it would be more scientifically rigorous to also disclose accuracy variances, rather than only highlighting the top-performing run. Hence, my lack of concern regarding the method not securing the top spot in every table persists. I maintain my positive stance on the paper.

---

### Decision · Program_Chairs · 2023-09-21

**Decision:**

Accept (poster)

**Comment:**

This paper got mixed ratings (A, WA, BA, R, BR) from reviewers.

The major strength recognized by reviewers is their high performance and practical efficacy on whole-body human pose and shape estimation, achieved by the proposed three modules.

The major concerns raised by the reviewers on the negative sides are regarding the level of novelty and some missing evaluations.

The authors' rebuttal and answers mostly resolved other concerns of reviewers.

The AC is inclined to the positive side. While the AC agreed that the technical innovation of each module would not be very strong, the contribution of each component is well supported by the experiments and their combinations demonstrate their practical advantages on the well-motivated research problem. The AC also believes the authors convincingly answered the questions and concerns raised by reviewers. Overall, the paper meets the bar of NeurIPS.
Authors should very carefully check the reviewer’s feedback and improve the quality of the paper for the camera-ready version.